# LLM-Powered GUI Agents in Phone Automation: Surveying Progress and Prospects

**Guangyi Liu**[1*]**, Pengxiang Zhao**[1*]**, Yaozhen Liang**[1*]**, Liang Liu**[2†]**, Yaxuan Guo**[2]**, Han Xiao**[3]**,
**Weifeng Lin**[3]**, Yuxiang Chai**[3]**, Yue Han**[1]**, Shuai Ren**[2]**, Hao Wang**[1]**, Xiaoyu Liang**[1]**, WenHao
**Wang**[1]**, Tianze Wu**[1]**, Zhengxi Lu**[1]**, Siheng Chen**[4]**, LiLinghao**[1]**, Hao Wang**[2]**, Guanjing Xiong**[2]**,
**Yong Liu**[1‡]**, Hongsheng Li**[3‡]
[1]*Zhejiang University*   [2]*vivo AI Lab*   [3]*CUHK MMLab*   [4]*Shanghai Jiao Tong University*

**Reviewed on OpenReview:** *https://openreview.net/forum?id=yWQqoi1G1K*

## Abstract

With the rapid rise of large language models (LLMs), phone automation has undergone transformative changes. This paper systematically reviews LLM-driven phone GUI agents, highlighting their evolution from script-based automation to intelligent, adaptive systems. We first contextualize key challenges, (i) limited generality, (ii) high maintenance overhead, and (iii) weak intent comprehension, and show how LLMs address these issues through advanced language understanding, multimodal perception, and robust decision-making. We then propose a taxonomy covering fundamental agent frameworks (single-agent, multi-agent, plan-then-act), modeling approaches (prompt engineering, training-based), and essential datasets and benchmarks. Furthermore, we detail task-specific architectures, supervised fine-tuning, and reinforcement learning strategies that bridge user intent and GUI operations. Finally, we discuss open challenges such as dataset diversity, on-device deployment efficiency, user-centric adaptation, and security concerns, offering forward-looking insights into this rapidly evolving field. By providing a structured overview and identifying pressing research gaps, this paper serves as a definitive reference for researchers and practitioners seeking to harness LLMs in designing scalable, user-friendly phone GUI agents. The collection of papers reviewed in this survey will be hosted and regularly updated on the GitHub repository: https://github.com/PhoneLLM/Awesome-LLM-Powered-Phone-GUI-Agents

## 1 Introduction

The core of phone GUI automation involves programmatically simulating human interactions with mobile interfaces to accomplish complex tasks. This technology has wide applications in testing and shortcut creation, enhancing efficiency and reducing manual effort Azim & Neamtiu (2013); Pan et al. (2020); Koroglu et al. (2018); Li et al. (2019); Degott et al. (2019). Traditional approaches rely on predefined scripts and templates which, while functional, lack flexibility when confronting variable interfaces and dynamic environments Arnatovich et al. (2018); Deshmukh & Phalke (2023); Nass (2024); Nass et al. (2021).

In computer science, an agent perceives its environment through sensors and acts via actuators to achieve goals Li et al. (2024d); Guo et al. (2024); Wang et al. (2024d); Jin et al. (2024); Bubeck et al. (2023). These range from simple scripts to complex systems capable of learning and adaptation Wang et al. (2024d); Jin et al. (2024); Huang et al. (2024b). Traditional phone automation agents are constrained by static scripts and limited adaptability, making them ill-suited for modern mobile interfaces' dynamic nature.

Building intelligent autonomous agents with planning, decision-making, and execution capabilities remains a long-term AI goal Albrecht & Stone (2018). As technologies advanced, agents evolved from traditional

---

*Equal Contribution; †Project Lead; ‡Corresponding Authors: yongliu@iipc.zju.edu.cn, hsli@ee.cuhk.edu.hk

forms Anscombe (2000); Dennett (1988); Shoham (1993) to AI agents Poole & Mackworth (2010); Inkster et al. (2018); Gao et al. (2018) incorporating machine learning and probabilistic decision-making. However, these still struggle with complex instructions Luger & Sellen (2016); Amershi et al. (2014) and dynamic environments Christiano et al. (2017); Köhl et al. (2019).

With the rapid development of Large Language Models (LLMs) like the GPT series Radford (2018); Radford et al. (2019); Brown (2020); Achiam et al. (2023) and specialized models such as Fuyu-8B Bavishi et al. (2023), LLM-based agents have demonstrated powerful capabilities across numerous domains Wang et al. (2023c); Hong et al. (2023); Li et al. (2023a); Park et al. (2023); Boiko et al. (2023); Qian et al. (2023); Xia et al. (2023); Dasgupta et al. (2023); Qian et al. (2024a); Dong et al. (2024); Goertzel (2014). As Figure 1 illustrates, conversational LLMs primarily focus on language understanding and generation, while LLM-based agents extend these capabilities by integrating perception and action components. This integration enables interaction with external environments through multimodal inputs and operational outputs Wang et al. (2023c); Hong et al. (2023); Qian et al. (2024a), bridging language understanding and real-world interactions Xi et al. (2023b); Li et al. (2024d); Guo et al. (2024); Furuta et al. (2024).

Applying LLM-based agents to phone automation has created a new paradigm, making mobile interface operations more intelligent Hong et al. (2024); Zheng et al. (2024a); Zhang et al. (2023a); Song et al. (2023b). ***LLM-powered phone GUI agents are intelligent systems that leverage large language models to understand, plan, and execute tasks on mobile devices by integrating natural language processing, multimodal perception, and action execution capabilities.*** These agents can recognize interfaces, understand instructions, perceive changes in real time, and respond dynamically. Unlike script-based automation, they can autonomously plan complex sequences through multimodal processing of instructions and interface information. Their adaptability and flexibility improve user experience through intent understanding, planning, and automated task execution, enhancing efficiency across scenarios from app testing to complex operations like configuring settings Wen et al. (2024), navigating maps Wang et al. (2024b;a), and shopping Zhang et al. (2023a).

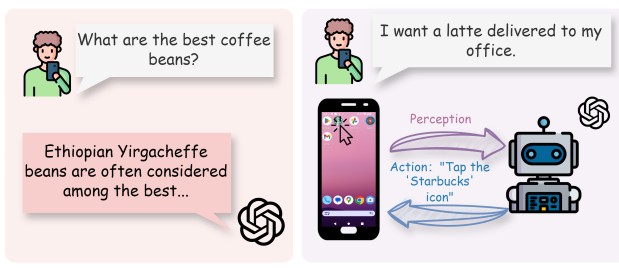

Figure 1: Comparison between conversational LLMs and phone GUI agents. While a conversational LLM can understand queries and provide informative responses (*e.g.*, recommending coffee beans), a Phone GUI agent can go beyond text generation to perceive the device's interface, decide on an appropriate action (like tapping an app icon), and execute it in the real environment, thus enabling tasks like ordering a latte directly on the user's phone.

Clarifying the development trajectory of phone GUI agents is crucial. On one hand, with the support of large language models Radford (2018); Radford et al. (2019); Brown (2020); Achiam et al. (2023), phone GUI agents can significantly enhance the efficiency of phone automation scenarios, making operations more intelligent and no longer limited to coding fixed operation paths. This enhancement not only optimizes phone automation processes but also expands the application scope of automation. On the other hand, phone GUI agents can understand and execute complex natural language instructions, transforming human intentions into specific operations such as automatically scheduling appointments, booking restaurants, summoning transportation, and even achieving functionalities similar to autonomous driving in advanced automation. These capabilities demonstrate the potential of phone GUI agents in executing complex tasks, providing convenience to users and laying practical foundations for AI development.

With the increasing research on large language models in phone automation Wen et al. (2023; 2024); Wang et al. (2024b;a); Liu et al. (2024d); Zhang et al. (2024b); Lu et al. (2024b), the research community's attention to this field has grown rapidly. However, there is still a lack of dedicated systematic surveys in this area, especially comprehensive explorations of phone automation from the perspective of large language models. Given the importance of phone GUI agents, the purpose of this paper is to fill this gap by systematically

summarizing current research achievements, reviewing relevant literature, analyzing the application status of large language models in phone automation, and pointing out directions for future research.

To provide a comprehensive overview of the current state and future prospects of LLM-Powered GUI Agents in Phone Automation, we present a taxonomy that categorizes the field into three main areas: Frameworks of LLM-powered phone GUI agents, Large Language Models for Phone Automation, and Datasets and Evaluation Methods Figure 2. This taxonomy highlights the diversity and complexity of the field, as well as the interdisciplinary nature of the research involved.

While the field of GUI automation is broad, this survey centers on a specific and critical domain: **LLM-powered agents for phone environments**. Our focus on mobile is deliberate, as it presents a unique convergence of challenges: a distinct interaction paradigm (touchscreens, gestures), constrained on-device resources, and a highly diverse and dynamic app ecosystem. However, we also recognize that many foundational techniques and milestone advancements in GUI automation are demonstrated across different platforms, including desktop and web. Therefore, while our primary lens is the mobile phone, we also discuss relevant cross-platform agents and technologies where they offer crucial insights into the principles, challenges, and future trajectory of phone automation. This approach allows us to provide a comprehensive and contextually rich overview of the field.

Our main contributions can be summarized as follows:

- **A Comprehensive and Systematic Survey of LLM-Powered Phone GUI Agents.** We provide an in-depth and structured overview of recent literature on LLM-powered phone automation, examining its developmental trajectory, core technologies, and real-world application scenarios. By comparing LLM-driven methods to traditional phone automation approaches, this survey clarifies how large models transform GUI-based tasks and enable more intelligent, adaptive interaction paradigms.

- **Methodological Framework from Multiple Perspectives.** Leveraging insights from existing studies, we propose a unified methodology for designing LLM-driven phone GUI agents. This encompasses framework design (e.g., single-agent vs. multi-agent vs. plan-then-act frameworks), LLM model selection and training (prompt engineering vs. training-based methods), data collection and preparation strategies (GUI-specific datasets and annotations), and evaluation protocols (benchmarks and metrics). Our systematic taxonomy and method-oriented discussion serve as practical guidelines for both academic and industrial practitioners.

- **In-Depth Analysis of Why LLMs Empower Phone Automation.** We delve into the fundamental reasons behind LLMs' capacity to enhance phone automation. By detailing their advancements in natural language comprehension, multimodal grounding, reasoning, and decision-making, we illustrate how LLMs bridge the gap between user intent and GUI actions. This analysis elucidates the critical role of large models in tackling issues of scalability, adaptability, and human-like interaction in real-world mobile environment.

- **Insights into Latest Developments, Datasets, and Benchmarks.** We introduce and evaluate the most recent progress in the field, highlighting innovative datasets that capture the complexity of modern GUIs and benchmarks that allow reliable performance assessment. These resources form the backbone of LLM-based phone automation, enabling systematic training, fair evaluation, and transparent comparisons across different agent designs.

- **Identification of Key Challenges and Novel Perspectives for Future Research.** Beyond discussing mainstream hurdles (e.g., dataset coverage, on-device constraints, reliability), we propose forward-looking viewpoints on user-centric adaptations, security and privacy considerations, long-horizon planning, and multi-agent coordination. These novel perspectives shed light on how researchers and developers might advance the current state of the art toward more robust, secure, and personalized phone GUI agents.

By addressing these aspects, our survey not only provides an up-to-date map of LLM-powered phone GUI automation but also offers a clear roadmap for future exploration. We hope this work will guide researchers

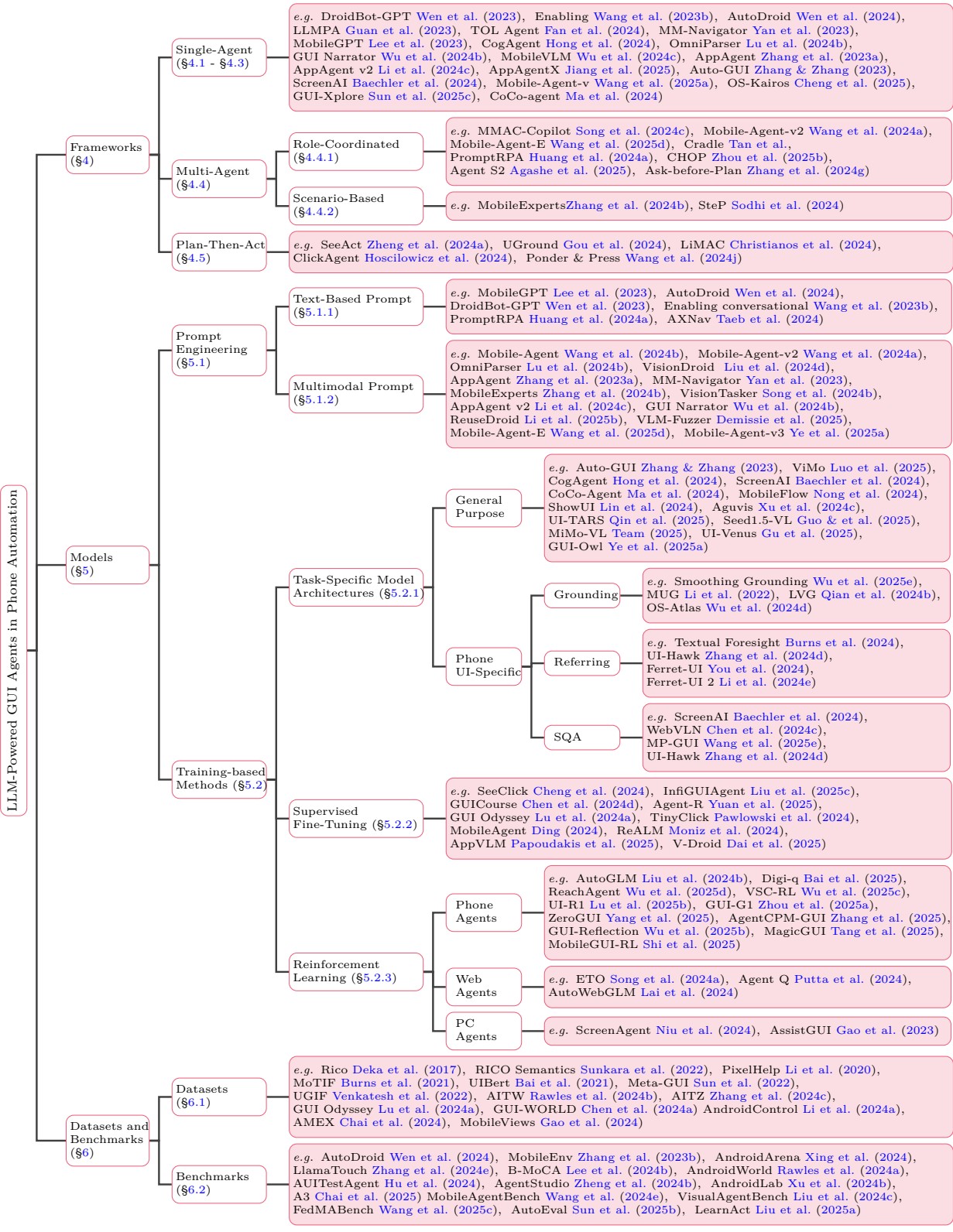

Figure 2: A comprehensive taxonomy of LLM-powered phone GUI agents in phone automation. Note that only a selection of representative works is included in this categorization.

in identifying pressing open problems and inform practitioners about promising directions to harness LLMs in designing efficient, adaptive, and user-friendly phone GUI agents.

## 2 Related Work

Our survey is situated at the intersection of two major research areas: traditional GUI automation and the emerging field of LLM-powered agents. In this section, we review representative surveys from both domains to contextualize our contribution and highlight its unique focus on the mobile phone platform.

### 2.1 Surveys on GUI Automation and Robotic Process Automation

Research in GUI automation has a long history, primarily rooted in software testing. Foundational books like Li & Wu (2006) laid the groundwork for developing automated testing tools. Comprehensive mapping studies, such as the 30-year review by Rodríguez-Pérez et al. (2021) and the work by Nass et al. (2021), have systematically analyzed the evolution and persistent challenges of GUI testing. Concurrently, a sub-field focused on mobile GUI testing has emerged, with surveys by Arnatovich et al. (2018) and Said et al. (2020) specifically addressing the objectives and challenges unique to mobile platforms, such as handling diverse screen sizes and touch-based interactions. More recent work by Deshmukh & Phalke (2023) continues to explore GUI testing as a means to enhance user experience.

Closely related to GUI automation is Robotic Process Automation (RPA), which also focuses on automating rule-based tasks on graphical interfaces. A significant body of literature has reviewed the landscape of RPA, including general literature reviews Ivančić et al. (2019), analyses of contemporary challenges Syed et al. (2020), and systematic mapping studies Enriquez et al. (2020). As RPA matured, researchers began exploring its synergy with AI, often termed "Intelligent Automation" Chakraborti et al. (2020); Agostinelli et al. (2019); Ribeiro et al. (2021). While both traditional GUI testing and RPA provide a solid foundation for task automation, they often lack the semantic understanding and reasoning capabilities needed to handle complex, non-deterministic tasks, a gap that LLM-powered agents aim to fill.

### 2.2 Surveys on LLM-Powered Agents

The rapid advancement of LLMs has catalyzed the development of autonomous agents, leading to a proliferation of surveys on this topic. Broad surveys by Xi et al. (2023a) and Wang et al. (2024c) provide a high-level overview of LLM-based agents, while Chang et al. (2024) focuses on the critical aspect of their evaluation. Other reviews, like Wali et al. (2023), concentrate specifically on intelligent agents for task automation.

Recently, several surveys have begun to chronicle the application of LLMs to GUI automation, as summarized in Table 1. These works, including Wang et al. (2024g); Zhang et al. (2024a); Li & Huang (2025); Du et al. (2025), have provided the first systematic overviews of (M)LLM-powered GUI agents. The survey by Wu et al. (2025a) is particularly relevant, as it narrows its focus to multimodal agents on mobile platforms.

However, as our comparative analysis in Table 1 illustrates, our work provides a unique **Mobile-Specific Analysis** that sets it apart. While other surveys are either platform-agnostic or offer a high-level mobile focus, our contribution lies in applying a rigorous analytical structure specifically to the phone ecosystem. This phone-centric viewpoint is not just an added topic, but the core lens through which we structure our entire analysis. Our primary contributions are articulated through three pillars, each deeply rooted in the mobile context:

- **A Unified Methodological Framework Tailored for Mobile:** We propose a comprehensive taxonomy organized into *Frameworks* (§4), *Models* (§5), and *Resources* (§6), each analyzed through a uniquely mobile-centric lens absent in prior work. For **Frameworks**, our analysis is grounded in the specific I/O modalities of smartphones, providing a detailed breakdown of mobile-specific perceptual information (e.g., view hierarchies, §4.1) and the distinct touch-based action space (e.g., swipes, drags, §4.3). For **Models**, we assess technical approaches (§5) not in a vacuum, but in the context of their suitability for the severe constraints of **on-device deployment**, a critical challenge unique to the mobile ecosystem. Finally,

Table 1: Comparison of this survey with existing work in the domains of GUI Automation and LLM Agents. A ✔symbol indicates explicit focus, while a ∘indicates related discussion. Our work is the first to provide an in-depth, focused analysis of LLM-powered agents specifically for the phone platform, including a quantitative comparison of mobile-specific benchmarks.

| Survey | Primary Focus | Platform | Traditional GUI Automation | LLM Agents | Mobile-Specific Analysis |
|---|---|---|---|---|---|
| Rodríguez-Pérez et al. (2021) | GUI Testing | General | ✔ | | ∘ |
| Nass et al. (2021) | GUI Testing Challenges | General | ✔ | | ∘ |
| Arnatovich et al. (2018) | Mobile GUI Testing | Phone | ✔ | | ✔ |
| Said et al. (2020) | Mobile GUI Challenges | Phone | ✔ | | ✔ |
| Syed et al. (2020) | RPA | General | ✔ | | |
| Xi et al. (2023a) | General LLM Agents | General | ∘ | ✔ | |
| Wang et al. (2024c) | General LLM Agents | General | ∘ | ✔ | |
| Wali et al. (2023) | Task Automation Agents | General | ∘ | ✔ | |
| Wu et al. (2025a) | Mobile Multimodal Agents | Phone | ∘ | ✔ | ✔ |
| Wang et al. (2024g) | (M)LLM-powered GUI Agents | General | ✔ | ✔ | ∘ |
| Zhang et al. (2024a) | (M)LLM-powered GUI Agents | General | ✔ | ✔ | ∘ |
| Li & Huang (2025) | (M)LLM-powered GUI Agents | General | ✔ | ✔ | ∘ |
| Du et al. (2025) | (M)LLM Agent Optimization | General | ✔ | ✔ | ∘ |
| **Our Work** | **LLM-powered Phone GUI Agents** | **Phone** | ✔ | ✔ | ✔ |

for **Resources**, our survey provides a targeted review and critical analysis of datasets and benchmarks built specifically for **phone GUI automation** (§6). This multi-faceted, mobile-specific methodological breakdown represents a core contribution that other surveys do not offer.

- **An In-depth Analysis of the Mobile Automation Trajectory:** Our survey provides a crucial historical narrative focused specifically on phone automation. We first dissect the pre-LLM era, identifying four central, long-standing challenges that hindered traditional methods like RPA and script-based testing: **Limited Generality**, **High Maintenance Costs**, **Poor Intent Comprehension**, and **Weak Screen GUI Perception** (§3). We then demonstrate how LLMs directly address these mobile-specific bottlenecks, establishing *why* they represent a paradigm shift *for mobile automation.* Specifically, we analyze how LLMs provide sophisticated **Contextual Semantic Understanding** to overcome intent ambiguity, leverage **Multi-Modal Perception** to interpret complex GUIs, and employ advanced **Reasoning and Decision-Making** to handle dynamic, cross-app workflows. This historical and problem-solution analysis provides a clear trajectory of the field's development.
- **A Forward-Looking Perspective on Phone-Centric Challenges:** Our analysis of future directions is also grounded in the mobile ecosystem (§7). We prioritize challenges that are most acute on phones, such as the technical hurdles of **on-device deployment**, the heightened need for **privacy and security** on personal devices, and the complexity of ensuring robust **long-horizon planning** across a multitude of apps. This pillar reflects our contribution to identifying key challenges and guiding future exploration specifically for phone GUI agents.

By focusing on these structural and analytical contributions, our survey provides a targeted, in-depth, and practical guide to the field.

## 3 Development of Phone Automation

The evolution of phone automation has been marked by significant technological advancements Kong et al. (2018), particularly with the emergence of LLMs Radford (2018); Radford et al. (2019); Brown (2020); Achiam et al. (2023). This section explores the historical development of phone automation, the challenges faced by traditional methods, and how LLMs have revolutionized the field.

### 3.1 Phone Automation Before the LLM Era

Before the advent of LLMs, phone automation was predominantly achieved through *traditional technical methods* Kirubakaran & Karthikeyani (2013); Azim & Neamtiu (2013); Amalfitano et al. (2014); Linares-

Vásquez et al. (2017); Kong et al. (2018); Zhao et al. (2024). This subsection delves into the primary areas of research and application during that period, including automation testing, shortcuts, and Robotic Process Automation (RPA), highlighting their methodologies and limitations.

### 3.1.1 Automation Testing

Phone applications (apps) have become extremely popular, with approximately 1.68 million apps in the Google Play Store[1]. The increasing complexity of apps Hecht et al. (2015) has raised significant concerns about app quality. Moreover, due to rapid release cycles and limited human resources, developers find it challenging to manually construct test cases. Therefore, various automated phone app testing techniques have been developed and applied, making phone automation testing the main application of phone automation before the era of large models Kirubakaran & Karthikeyani (2013); Kong et al. (2018); Linares-Vásquez et al. (2017); Zein et al. (2016). Test cases for phone apps are typically represented by a sequence of GUI events Jensen et al. (2013) to simulate user interactions with the app. The goal of automated test generators is to produce such event sequences to achieve high code coverage or detect bugs Zhao et al. (2024).

In the development history of phone automation testing, we have witnessed several key breakthroughs and advancements. Initially, random testing (e.g., Monkey Testing Machiry et al. (2013)) was used as a simple and fundamental testing method, detecting application stability and robustness by randomly generating user actions. Although this method could cover a wide range of operational scenarios, its testing process lacked focus and was difficult to reproduce and pinpoint specific issues Kong et al. (2018).

Subsequently, *model-based testing* Amalfitano et al. (2012; 2014); Azim & Neamtiu (2013) became a more systematic testing approach. It establishes a user interface model of the application, using predefined states and transition rules to generate test cases. This method improved testing coverage and efficiency, but the construction and maintenance of the model required substantial manual involvement, and updating the model became a challenge for highly dynamic applications.

With the development of machine learning techniques, *learning-based* testing methods began to emerge Koroglu et al. (2018); Pan et al. (2020); Li et al. (2019); Degott et al. (2019). These methods generate test cases by analyzing historical data to learn user behavior patterns. For example, Humanoid Li et al. (2019) uses deep learning to mimic human tester interaction behavior and uses the learned model to guide test generation like a human tester. However, this method relies on human-generated datasets to train the model and needs to combine the model with a set of predefined rules to guide testing.

Recently, *reinforcement learning* Ladosz et al. (2022) has shown great potential in the field of automated testing. DinoDroid Zhao et al. (2024) is an example that uses Deep Q-Network (DQN) Fan et al. (2020) to automate testing of Android applications. By learning behavior models of existing applications, it automatically explores and generates test cases, not only improving code coverage but also enhancing bug detection capabilities. Deep reinforcement learning methods can handle more complex state spaces and make more intelligent decisions but also face challenges such as high training costs and poor model generalization capabilities Luo et al. (2024).

### 3.1.2 Shortcuts

Shortcuts on mobile devices refer to predefined rules or trigger conditions that enable users to execute a series of actions automatically Bridle & McCreath (2006); Guerreiro et al. (2008); Kennedy & Everett (2011). These shortcuts are designed to streamline interaction by reducing repetitive manual input. For instance, the Tasker app on the Android platform[2] and the Shortcuts feature on iOS[3] allow users to automate tasks like turning on Wi-Fi, sending text messages, or launching apps under specific conditions such as time, location, or events. These implementations leverage simple IF-THEN and manually-designed logic but are inherently limited in scope and flexibility.

---

[1]https://www.statista.com.
[2]https://play.google.com.
[3]https://support.apple.com.

### 3.1.3 Robotic Process Automation

Robotic Process Automation(RPA) applications on phone devices aim to simulate human users performing repetitive tasks across applications Agostinelli et al. (2019). Phone RPA tools generate repeatable automation processes by recording user action sequences. These tools are used in enterprise environment to automate tasks such as data entry and information gathering, reducing human errors and improving efficiency, but they **struggle with dynamic interfaces and require frequent script updates** Pramod (2022); Syed et al. (2020).

### 3.2 Challenges of Traditional Methods

Despite the advancements made, traditional phone automation methods faced significant challenges that hindered further development. This subsection analyzes these challenges, including lack of generality and flexibility, high maintenance costs, difficulty in understanding complex user intentions, and insufficient intelligent perception, highlighting the need for new approaches.

### 3.2.1 Limited Generality

Traditional automation methods are often tailored to specific applications and interfaces, lacking adaptability to different apps and dynamic user environment Clarke et al. (2016); Li et al. (2017); Patel & Pasha (2015); Asadullah & Raza (2016). For example, automation scripts designed for a specific app may not function correctly if the app updates its interface or if the user switches to a different app with similar functionality. This inflexibility makes it difficult to extend automation across various usage scenarios without significant manual reconfiguration.

These methods typically follow predefined sequences of actions and cannot adjust their operations based on changing contexts or user preferences. For instance, if a user wants an automation to send a customized message to contacts who have birthdays on a particular day, traditional methods struggle because they cannot dynamically access and interpret data from the contacts app, calendar, and messaging app simultaneously. Similarly, automating tasks that require conditional logic—such as playing different music genres based on the time of day or weather conditions—poses a challenge for traditional automation tools, as they lack the ability to integrate real-time data and make intelligent decisions accordingly Majeed et al. (2020); Liu et al. (2023).

### 3.2.2 High Maintenance Costs

Writing and maintaining automation scripts require professional knowledge and are time-consuming and labor-intensive Kodali et al. (2019); Kodali & Mahesh (2017); Moreira et al. (2023); Lamberton et al. (2017); Meironke & Kuehnel (2022). Taking RPA as an example, as applications continually update and iterate, scripts need frequent modifications. When an application's interface layout changes or functions are updated, RPA scripts originally written for the old version may not work properly, requiring professionals to spend considerable time and effort readjusting and optimizing the scripts Tripathi (2018); Ling et al. (2020); Agostinelli et al. (2022).

The high entry barrier also limits the popularity of some automation features Le et al. (2020); Roffarello et al. (2024). For example, Apple's Shortcuts [4] can combine complex operations, such as starting an Apple Watch fitness workout, recording training data, and sending statistical data to the user's email after the workout. However, setting up such a complex shortcut often requires the user to perform a series of complicated operations on the phone following fixed rules. This is challenging for ordinary users, leading many to abandon usage due to the complexity of manual script writing.

### 3.2.3 Poor Intent Comprehension

Rule-based and script-based systems can only execute predefined tasks or engage in simple natural language interactions Kepuska & Bohouta (2018); Cowan et al. (2017). Simple instructions like "open the browser" can be handled using traditional natural language processing algorithms, but complex instructions like "open the

---

[4]https://support.apple.com.

browser, go to Amazon, and purchase a product" cannot be completed. These traditional systems are based on fixed rules and lack in-depth understanding and parsing capabilities for complex natural language Anicic et al. (2010); Kang et al. (2013); Karanikolas et al. (2023).

They require users to manually write scripts to interact with the phone, greatly limiting the application of intelligent assistants that can understand complex human instructions. For example, when a user wants to check flight information for a specific time and book a ticket, traditional systems cannot accurately understand the user's intent and automatically complete the series of related operations, necessitating manual script writing with multiple steps, which is cumbersome and requires high technical skills.

### 3.2.4   Weak Screen GUI Perception

Different applications present a wide variety of GUI elements, making it challenging for traditional methods like RPA to accurately recognize and interact with diverse controls Fu et al. (2024); Banerjee et al. (2013); Chen et al. (2018); Brich et al. (2017). Traditional automation often relies on fixed sequences of actions targeting specific controls or input fields, exhibiting Weak Screen GUI Perception that limits their ability to adapt to variations in interface layouts and component types. For example, in an e-commerce app, the product details page may include dynamic content like carousels, embedded videos, or interactive size selection menus, which differ significantly from the simpler layout of a search results page. Traditional methods may fail to accurately identify and interact with the "Add to Cart" button or select product options, leading to unsuccessful automation of purchasing tasks.

Moreover, traditional automation struggles with understanding complex screen information such as dynamic content updates, pop-up notifications, or context-sensitive menus that require adaptive interaction strategies. Without the ability to interpret visual cues like icons, images, or contextual hints, these methods cannot handle tasks that involve navigating through multi-layered interfaces or responding to real-time changes. For instance, automating the process of booking a flight may involve selecting dates from a calendar widget, choosing seats from an interactive seat map, or handling security prompts—all of which require sophisticated perception and interpretation of the interface Zhang et al. (2024e).

In phone automation, many apps do not provide open API interfaces, forcing solutions to rely directly on the GUI for triggering actions and retrieving information. Even when tools are used to parse the Android UI Wu et al. (2021), non-standard controls often prevent accurate JSON parsing, further complicating automated testing and interaction. Additionally, because the GUI is a universal and consistent interface across apps regardless of their internal design, it naturally becomes the central focus of phone automation methods.

These limitations significantly impede the widespread application and deep development of traditional phone automation technologies. Without intelligent perception capabilities, automation cannot adapt to the complexities of modern app interfaces, which are increasingly dynamic and rich in interactive elements. This underscores the urgent need for new methods and technologies that can overcome these bottlenecks and achieve more intelligent, flexible, and efficient phone automation.

### 3.3   LLMs Boost Phone Automation

The advent of LLMs has marked a significant shift in the landscape of phone automation, enabling more dynamic, context-aware, and sophisticated interactions with mobile devices. As illustrated in Figure 3, the research on LLM-powered phone GUI agents has progressed through pivotal milestones, where models become increasingly adept at interpreting multimodal data, reasoning about user intents, and autonomously executing complex tasks. This section clarifies how LLMs address traditional limitations and examines why *scaling laws* can further propel large models in phone automation. As will be detailed in § 5 and § 6, LLM-based solutions for phone automation generally follow two routes: (1) *Prompt Engineering*, where *pre-trained* models are guided by carefully devised prompts, and (2) *Training-Based Methods*, where LLMs undergo additional optimization on GUI-focused datasets. The following subsections illustrate how LLMs mitigate the core challenges of traditional phone automation—ranging from *contextual semantic understanding* and *GUI perception* to *reasoning and decision making*—and briefly highlight the role of *scaling laws* in enhancing these capabilities.

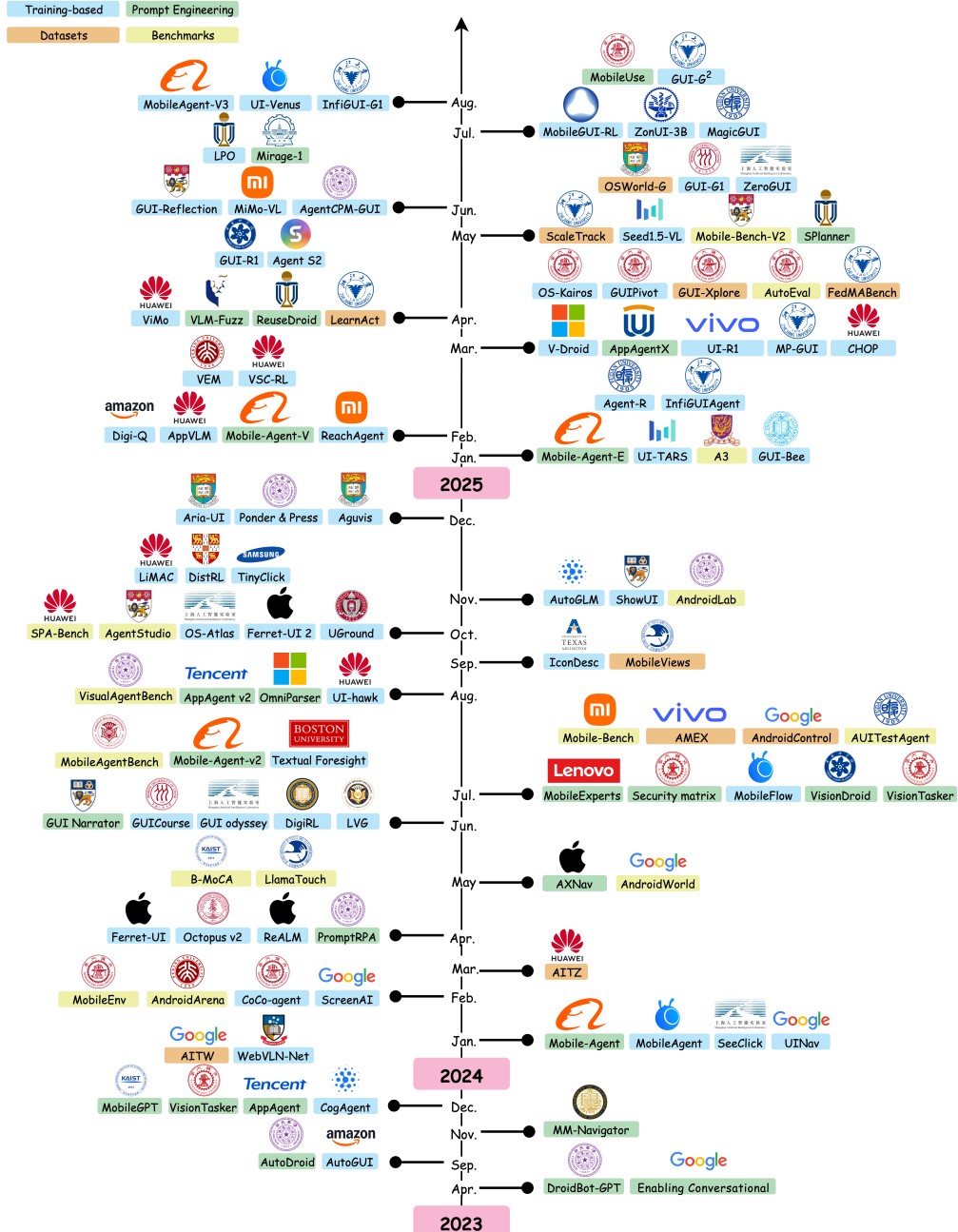

Figure 3: Milestones in the development of LLM-powered phone GUI agents. This figure divides advancements into four primary parts: **Prompt Engineering**, **Training-Based Methods**, **Datasets** and **Benchmarks**. Prompt Engineering leverages pre-trained LLMs by strategically crafting input prompts, as detailed in §5.1, to perform specific tasks without modifying model parameters. In contrast, Training-Based Methods, discussed in §5.2, involve adapting LLMs via supervised fine-tuning or reinforcement learning on GUI-specific data, thereby enhancing their ability to understand and interact with mobile UIs.

**Scaling Laws in LLM-Based Phone Automation.** Scaling laws—originally observed in general-purpose LLMs, where increasing model capacity and training data yields emergent capabilities Brown et al. (2020); Kaplan et al. (2020); Hagendorff (2023)—have similarly begun to manifest in phone GUI automation. As datasets enlarge and encompass more diverse apps, usage scenarios, and user behaviors, recent findings Cheng et al. (2024); Chen et al. (2024d); Lu et al. (2024a); Pawlowski et al. (2024) show consistent gains in

step-by-step automation tasks such as clicking buttons or entering text. This data scaling not only captures broader interface layouts and device contexts but also reveals latent "emergent" competencies, allowing LLMs to handle more abstract, multi-step instructions. Empirical evidence from *in-domain* scenarios Li et al. (2024a) further underscores how expanded coverage of phone apps and user patterns systematically refines automation accuracy. In essence, as model sizes and data complexity grow, phone GUI agents exploit these scaling laws to bridge the gap between user intent and real-world GUI interactions with increasing efficiency and sophistication.

**Contextual Semantic Understanding.** LLMs have transformed natural language processing for phone automation by learning from extensive textual corpora Vaswani (2017); Brown et al. (2020); Radford (2018); Devlin (2018); Wen et al. (2024); Zhang et al. (2023a). This training captures intricate linguistic structures and domain knowledge Karanikolas et al. (2023), allowing agents to parse multi-step commands and generate context-informed responses. MobileAgent Wang et al. (2024b), for example, interprets user directives like scheduling appointments or performing transactions with high precision, harnessing the Transformer architecture Vaswani (2017) for efficient encoding of complex prompts. Consequently, phone GUI agents benefit from stronger natural language grounding, bridging user-intent gaps once prevalent in script-based systems.

**Screen GUI with Multi-Modal Perception.** Screen GUI perception in earlier phone automation systems typically depended on static accessibility trees or rigid GUI element detection, which struggled to adapt to changing app interfaces. Advances in LLMs, supported by large-scale multimodal datasets Zhao et al. (2023); Chang et al. (2024); Minaee et al. (2024), allow models to unify textual and visual signals in a single representation. Systems like UGround Gou et al. (2024), Ferret-UI You et al. (2024), and UI-Hawk Zhang et al. (2024d) excel at grounding natural language descriptions to on-screen elements, dynamically adjusting as interfaces evolve. Moreover, SeeClick Cheng et al. (2024) and ScreenAI Baechler et al. (2024) demonstrate that learning directly from screenshots—rather than purely textual metadata—can further enhance adaptability. By integrating visual cues with user language, LLM-based agents can respond more flexibly to a wide range of UI designs and interaction scenarios.

**Reasoning and Decision Making.** LLMs also enable advanced reasoning and decision-making by combining language, visual context, and historical user interactions. Pre-training on broad corpora equips these models with the capacity for complex reasoning Wang et al. (2023a); Yuan et al. (2024), multi-step planning Song et al. (2023a); Valmeekam et al. (2023), and context-aware adaptation Talukdar & Biswas (2024); Koike et al. (2024). MobileAgent-V2 Wang et al. (2024a), for instance, introduces a specialized planning agent to track task progress while a decision agent optimizes actions. Auto-GUI Zhang & Zhang (2023) applies a multimodal chain-of-action approach that accounts for both previous and forthcoming steps, and SteP Sodhi et al. (2024) uses stacked LLM modules to solve diverse web tasks. Similarly, MobileGPT Lee et al. (2023) leverages an app memory system to minimize repeated mistakes and bolster adaptability. Such architectures demonstrate higher success rates in complex phone operations, reflecting a new level of autonomy in orchestrating tasks that previously demanded handcrafted scripts.

Overall, LLMs are transforming phone automation by reinforcing semantic understanding, expanding multimodal perception, and enabling sophisticated decision-making strategies. The scaling laws observed in datasets like AndroidControl Li et al. (2024a) reinforce the notion that a larger volume and diversity of demonstrations consistently elevate model accuracy. As these techniques mature, LLM-driven phone GUI agents continue to redefine how users interact with mobile devices, ultimately paving the way for a more seamless and user-centric automation experience.

### 3.4 Emerging Commercial Applications

The integration of LLMs has enabled novel commercial applications that leverage phone automation, offering innovative solutions to real-world challenges. This subsection highlights several prominent cases, presented in chronological order based on their release dates, where LLM-based GUI agents are reshaping user experiences, improving efficiency, and providing personalized services.

**Apple Intelligence.** On June 11, 2024, Apple introduced its personal intelligent system, Apple Intelligence[5], seamlessly integrating AI capabilities into iOS, iPadOS, and macOS. It enhances communication, productivity, and focus features through intelligent summarization, priority notifications, and context-aware replies. For instance, Apple Intelligence can summarize long emails, transcribe and interpret call recordings, and generate personalized images or "Genmoji." A key aspect is on-device processing, which ensures user privacy and security. By enabling the system to operate directly on the user's device, Apple Intelligence safeguards personal information while providing an advanced, privacy-preserving phone automation experience.

**vivo PhoneGPT.** On October 10, 2024, vivo unveiled OriginOS 5[6], its newest mobile operating system, featuring an AI agent ability named PhoneGPT. By harnessing large language models, PhoneGPT can understand user instructions, preferences, and on-screen information, autonomously engaging in dialogues and detecting GUI states to operate the smartphone. Notably, it allows users to order coffee or takeout with ease and can even carry out a full phone reservation process at a local restaurant through extended conversations. By integrating the capabilities of large language models with native system states and APIs, PhoneGPT illustrates the great potential of phone GUI agents.

**Honor YOYO Agent.** Released on October 24, 2024, the Honor YOYO Agent[7] exemplifies an phone automation assistant that adapts to user habits and complex instructions. With just one voice or text command, YOYO can automate multi-step processes—such as comparing prices to secure discounts when shopping, automatically filling out forms, ordering beverages aligned with user preferences, or silencing notifications during online meetings. By learning from user behaviors, YOYO reduces the complexity of human-device interaction, offering a more effortless and intelligent phone experience.

**Zhipu.AI AutoGLM.** On October 25, 2024, Zhipu.AI introduced AutoGLM Liu et al. (2024b), an intelligent agent that simulates human operations on smartphones. With simple text or voice commands, AutoGLM can like and comment on social media posts, purchase products, book train tickets, or order takeout. Its capabilities extend beyond mere API calls—AutoGLM can navigate interfaces, interpret visual cues, and execute tasks that mirror human interaction steps. This approach streamlines daily tasks and demonstrates the versatility and practicality of LLM-driven phone automation in commercial applications.

These emerging commercial applications—from Apple's privacy-focused on-device intelligence to vivo's PhoneGPT, Honor's YOYO agent, and Zhipu.AI's AutoGLM—showcase how LLM-based agents are transcending traditional user interfaces. They enable more natural, efficient, and personalized human-device interactions. As models and methods continue to evolve, we can anticipate even more groundbreaking applications, further integrating AI into the fabric of daily life and professional workflows.

## 4 Frameworks and Components of Phone GUI Agents

MLLM-powered phone GUI agents can be designed using different architectural paradigms and components, ranging from straightforward, single-agent systems Wang et al. (2023b); Wen et al. (2023; 2024); Zhang et al. (2023a); Wang et al. (2024b) to more elaborate multi-agent Wang et al. (2024a); Zhang et al. (2024b;f) or multi-stage Zheng et al. (2024a); Gou et al. (2024); Hoscilowicz et al. (2024) approaches. A fundamental scenario involves a *single agent* that operates incrementally, without precomputing an entire action sequence from the outset. Instead, the agent continuously observes the **dynamically changing** mobile environment—where available UI elements, device states, and relevant contextual factors may shift in unpredictable ways—and cannot be exhaustively enumerated in advance. As a result, the agent must adapt its strategy **step-by-step**, making decisions based on the current situation rather than following a fixed plan. This **iterative** decision-making process can be effectively modeled using a **Partially Observable Markov Decision Process (POMDP)**, a well-established framework for handling sequential decision-making under uncertainty Monahan (1982); Spaan (2012). By modeling the task as a POMDP, we capture its dynamic nature, the impossibility of pre-planning all actions, and the necessity of adjusting the agent's approach at each decision point.

---

[5]https://www.apple.com/apple-intelligence/.
[6]https://www.vivo.com.cn/originos
[7]https://www.honor.com/cn/magic-os/.

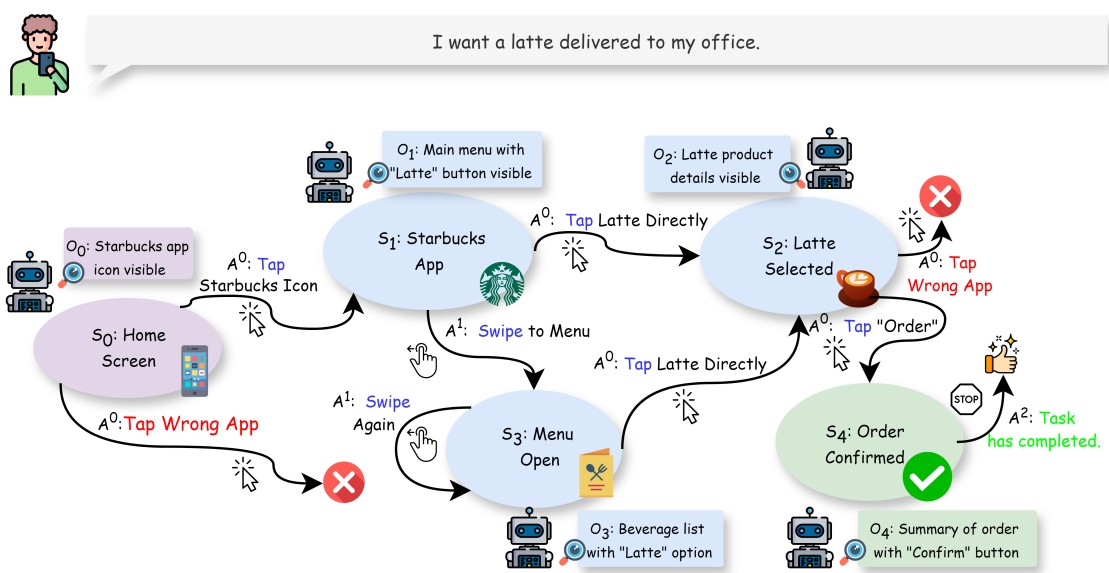

Figure 4: POMDP model for ordering a latte. Each circle represents a state (e.g., Home Screen, App Homepage, Latte Details Page, Customize Order, Order Confirmation, Order Complete). The agent starts at the initial state $S_0$ (Home Screen) and makes decisions at each step (e.g., tapping the Starbucks app icon, selecting the "Latte" button, viewing latte details). Due to partial observability, the agent receives limited information at each decision point (e.g., $O_0$: Starbucks app icon visible, $O_1$: "Latte" button visible, $O_2$: Latte product details visible). Some actions correctly advance towards the goal, while others may cause errors requiring corrections. The final goal is to confirm the order.

As illustrated in Figure 4, consider a simple example: the agent's goal is to order a latte through the Starbucks app. The app's interface may vary depending on network latency, promotions displayed, or the user's last visited screen. The agent cannot simply plan all steps in advance; it must observe the current screen, identify which UI elements are present, and then choose an action (like tapping the Starbucks icon, swiping to a menu, or selecting the latte). After each action, the state changes, and the agent re-evaluates its options. This dynamic, incremental decision-making is precisely why POMDPs are a suitable framework. In the POMDP formulation for phone automation:

**States** ($S$). At each decision point, the agent's perspective is described as a *state*, a comprehensive snapshot of all relevant information that could potentially influence the decision-making process. This state encompasses the current **UI information** (e.g., screenshots, UI trees, OCR-extracted text, icons), the **phone's own status** (network conditions, battery level, location), and the **task context** (the user's goal—"order a latte"—and the agent's progress toward it). The state $S_t$ represents the complete, underlying situation of the environment at time $t$, which may not be directly observable in its entirety.

**Actions** ($A$). Given the state $S_t$ at time $t$, the agent selects from available actions (taps, swipes, typing text, launching apps) that influence the subsequent state. The details of how phone GUI agents make decisions are introduced in § 4.2, and the design of the action space is discussed in § 4.3.

**Transition Dynamics** ($P(s'|s, a)$). When the agent executes an action $a_t$ at time $t$, it leads to a new state $S_{t+1}$. Some transitions may be deterministic (e.g., tapping a known button reliably opens a menu), while others are uncertain (e.g., network delays, unexpected pop-ups). Mathematically, we have the transition probability $P(s'|s, a)$ which describes the likelihood of transitioning from state $S_t$ to state $S_{t+1}$ given action $a_t$.

**Observations** ($O$). The agent receives *observations* $O_t$ at time $t$ which are partial and imperfect reflections of the true state $S_t$. In the phone automation context, these observations could be, for example, a glimpse of the visible UI elements (not the entire UI tree), a brief indication of the network status (such as a signal icon

without detailed connection parameters), or a partial view of the battery level indicator. These observations $O_t$ provide the agent with some, but not all, of the information relevant to the state $S_t$. The agent must infer and make decisions based on these limited observations, attempting to reach the desired goal state despite the partial observability. The details of phone GUI agent perception are discussed in § 4.1.

Under this POMDP-based paradigm, the agent aims to make decisions that lead to the goal state by observing the current state and choosing appropriate actions. It continuously re-evaluates its strategy as conditions evolve, promoting real-time responsiveness and dynamic adaptation. The agent **observes the state $S_t$ at time $t$, chooses an action $a_t$, and then based on the resulting observation $O_{t+1}$ and new state $S_{t+1}$**, refines its strategy.

As illustrated in Figure 5, frameworks of phone GUI agents aim to integrate perception, reasoning, and action capabilities into cohesive agents that can interpret user intentions, understand complex UI states, and execute appropriate operations within mobile environment. By examining these frameworks, we can identify best practices, guide future advancements, and choose the right approach for various applications and contexts.

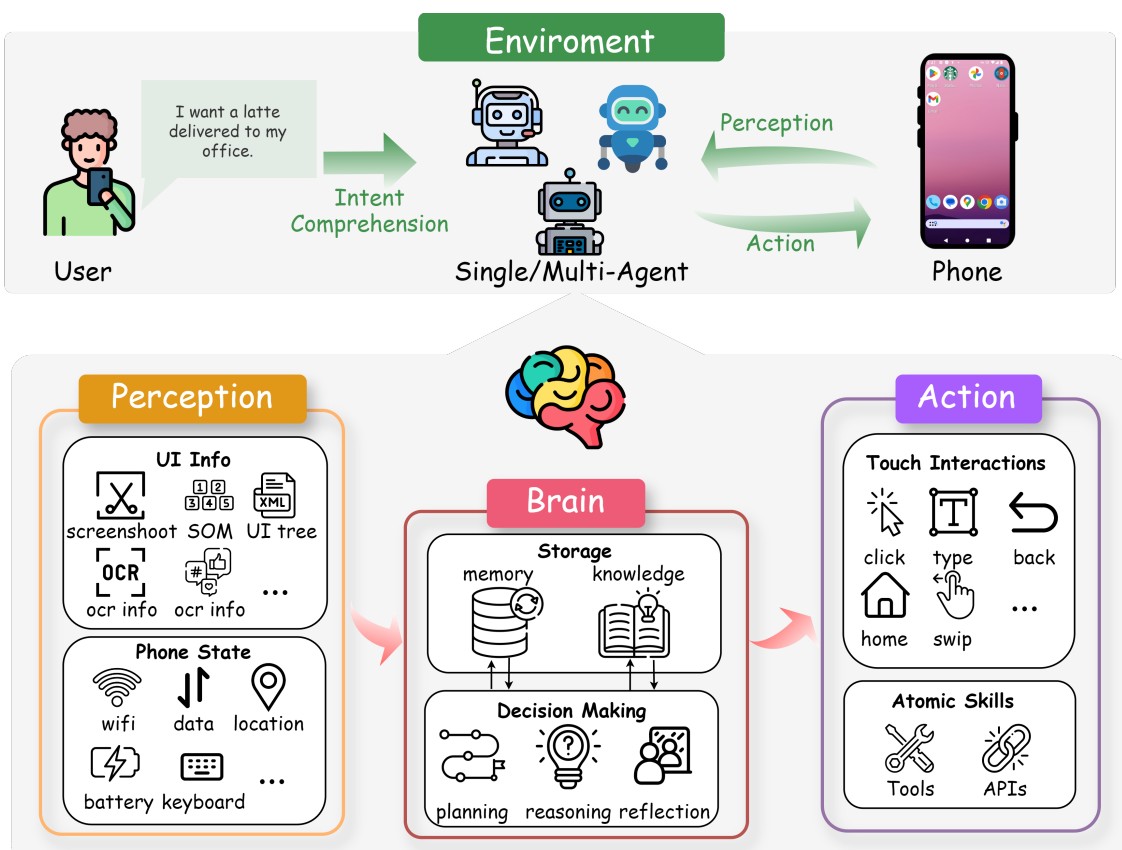

Figure 5: Overview of MLLM-powered phone GUI agent framework. The user's intent, expressed through natural language, is mapped to UI operations. By perceiving UI information and phone state(§4.1) , the agent leverages stored knowledge and memory to plan, reason, and reflect (§4.2) . Finally, it executes actions to fulfill the user's goals(§4.3).

To address limitations in adaptability and scalability, §4.4 introduces multi-agent frameworks, where specialized agents collaborate, enhance efficiency, and handle more diverse tasks in parallel. Finally, §4.5 presents the Plan-Then-Act Framework, which explicitly separates the planning phase from the execution phase. This approach allows agents to refine their conceptual plans before acting, potentially improving both accuracy and robustness.

### 4.1 Perception in Phone GUI Agents

Perception is a fundamental component of the basic framework for MLLM-powered phone GUI agents. It is responsible for capturing and interpreting the state of the mobile environment, enabling the agent to understand the current context and make informed decisions. In the overall pipeline, perception serves as the initial step in the POMDP, providing the necessary input for the reasoning and action modules to operate effectively.

#### 4.1.1 UI Information Perception

UI information is crucial for agents to interact seamlessly with the mobile interface. It can be further categorized into UI tree-based and screenshot-based approaches, supplemented by techniques like Set-of-Marks (SoM) and Icon & OCR enhancements.

**UI tree** is a structured, hierarchical representation of the UI elements on a mobile screen Medhi et al. (2013); Räsänen & Saarinen (2015). Each node in the UI tree corresponds to a UI component, containing attributes such as class type, visibility, and resource identifiers.[8] Early datasets like PixelHelp Li et al. (2020), MoTIF Burns et al. (2021), and UIBert Bai et al. (2021) utilized UI tree data to enable tasks such as mapping natural language instructions to UI actions and performing interactive visual environment interactions. DroidBot-GPT Wen et al. (2023) was the first work to investigate how pre-trained language models can be applied to app automation without modifying the app or the model. DroidBot-GPT uses the UI tree as its primary perception information. The challenge lies in converting the structured UI tree into a format that LLMs can process effectively. DroidBot-GPT addresses this by transforming the UI tree into natural language sentences. Specifically, it extracts all user-visible elements, generates prompts like "A view <name>that can..." for each element, and combines them into a cohesive description of the current UI state. This approach mitigates the issue of excessively long and complex UI trees by presenting the information in a more natural and concise format suitable for LLMs. Subsequent developments, such as Enabling Conversational Interaction Wang et al. (2023b) and AutoDroid Wen et al. (2024), further refined this approach by representing the view hierarchy as HTML. Enabling Conversational Interaction introduces a method to convert the view hierarchy into HTML syntax, mapping Android UI classes to corresponding HTML tags and preserving essential attributes such as class type, text, and resource identifiers. This representation aligns closely with the training data distribution of LLMs, enhancing their ability to perform few-shot learning and improving overall UI understanding. AutoDroid extends this work by developing a GUI parsing module that converts the GUI into a simplified HTML representation using specific HTML tags like <button>, <checkbox>, <scroller>, <input>, and <p>. Additionally, AutoDroid implements automatic scrolling of scrollable components to ensure that comprehensive UI information is available to the LLM, thereby enhancing decision-making accuracy and reducing computational overhead. Furthermore, LLMPA Guan et al. (2023) employs object detection models to comprehend page layouts and optimizes the grouping of UI elements for potential actions. This approach reduces redundant information in the UI tree, thereby enhancing the accuracy and speed of decision making. Similar to this approach, the TOL Agent Fan et al. (2024) introduces a variant of the UI tree, known as the Hierarchical Layout Tree, to represent the hierarchical layout of screen captures. In this tree, nodes represent different levels of regions. This structure, combined with a trained DINO model, aids in generating more accurate screen descriptions for MLLM.

**Screenshots** provide a visual snapshot of the current UI, capturing the appearance and layout of UI elements. Unlike UI trees, which require API access and can become unwieldy with complex hierarchies, screenshots offer a more flexible and often more comprehensive representation of the UI. Additionally, UI trees present challenges such as missing or overlapping controls and the inability to directly reference UI elements programmatically, making screenshots a more practical and user-friendly alternative for quickly assessing and sharing the state of a user interface. Auto-GUI Zhang & Zhang (2023) introduced a multimodal agent that relies on screenshots for GUI control, eliminating the dependency on UI trees. This approach allows the agent to interact with the UI directly through visual perception, enabling more natural and human-like interactions. Auto-GUI employs a chain-of-action technique that uses both previously executed actions and planned future actions to guide decision-making, achieving high action type prediction accuracy

---

[8]https://developer.android.com/reference/android/view/View.

and efficient task execution. Following Auto-GUI, a series of multimodal solutions emerged, including MM-Navigator Yan et al. (2023), CogAgent Hong et al. (2024), AppAgent Zhang et al. (2023a), VisionTasker Song et al. (2024b), MobileGPT Lee et al. (2023), GUI Narrator Wu et al. (2024b), MobileVLM Wu et al. (2024c), AdaptAgent Verma et al. (2024), WebVoyager He et al. (2024) and Steward Tang & Shin (2024). These frameworks leverage screenshots in combination with supplementary information to enhance UI understanding and interaction capabilities.

**Set-of-Mark (SoM)** is a prompting technique used to annotate screenshots with OCR, icon, and UI tree information, thereby enriching the visual data with textual descriptorsYang et al. (2023). For example, MM-Navigator Yan et al. (2023) uses SoM to label UI elements with unique identifiers, allowing the LLM to reference and interact with specific components more effectively. This method has been widely adopted in subsequent works such as AppAgent Zhang et al. (2023a), VisionDroid Liu et al. (2024d), OmniParser Lu et al. (2024b) and VisualWebArena Koh et al. (2024a), which utilize SoM to enhance the agent's ability to interpret and act upon UI elements based on visual, textual, and structural cues.

**Icon & OCR enhancements** provide additional layers of information that complement the visual data, enabling more precise action decisions. For instance, Mobile-Agent-v2 Wang et al. (2024a) integrates OCR and icon data with screenshots to provide a richer context for the LLM, allowing it to interpret and execute more complex instructions that require understanding both text and visual icons. Icon & OCR enhancements are employed in various works, including VisionTasker Song et al. (2024b), MobileGPT Lee et al. (2023), OmniParser Lu et al. (2024b), and WindowsAgentArena Bonatti et al. (2024), to improve the accuracy and reliability of phone GUI agents.

### 4.1.2 Phone State Perception

Phone state information, such as keyboard status and location data, further contextualizes the agent's interactions. For example, Mobile-Agent-v2 Wang et al. (2024a) uses keyboard status to determine when text input is required. Location data, while not currently utilized, represents a potential form of phone state information that could be used to recommend nearby services or navigate to specific addresses. This additional state information enhances the agent's ability to perform context-aware actions, making interactions more intuitive and efficient.

The perception information gathered through UI trees, screenshots, SoM, OCR, and phone state is converted into prompt tokens that the LLM can process. This conversion is crucial for enabling seamless interaction between the perception module and the reasoning and action modules. Detailed methodologies for transforming perception data into prompt formats are discussed in § 5.1.

## 4.2 Brain in Phone GUI Agents

The brain of an LLM-based phone automation agent is its cognitive core, primarily constituted by a LLM. The LLM serves as the agent's reasoning and decision-making center, enabling it to interpret inputs, generate appropriate responses, and execute actions within the mobile environment Ge et al. (2023); Mei et al. (2024). Leveraging the extensive knowledge embedded within LLMs, agents benefit from advanced language understanding, contextual awareness, and the ability to generalize across diverse tasks and scenarios.

### 4.2.1 Storage

Storage encompasses both memory and knowledge, which are critical for maintaining context and informing the agent's decision-making processes.

**Memory** refers to the agent's ability to retain information from past interactions with users and the environment Xi et al. (2023b). This is particularly useful for cross-application operations, where continuity and coherence are essential for completing multi-step tasks. For example, Mobile-Agent-v2 Wang et al. (2024a) integrates a memory unit that records task-related focus content from historical screens. This memory is accessed by the decision-making module when generating operations, ensuring that the agent can reference and update relevant information dynamically. The Self-MAP framework Deng et al. (2024) establishes a memory repository based on the history of conversational interactions. It utilizes a multifaceted matching

approach to retrieve the top-K memory snippets that are semantically relevant to the current dialogue state and have similar trajectories. This assists the agent in effectively utilizing limited context space during multi-turn interactions, thereby enhancing its ability to comprehend and execute user instructions.

**Knowledge** pertains to the agent's understanding of phone automation tasks and the functionalities of various apps. This knowledge can originate from multiple sources:

- **Pre-trained Knowledge.** LLMs are inherently equipped with a vast amount of general knowledge, including common-sense reasoning and familiarity with programming and markup languages such as HTML. This pre-existing knowledge allows the agent to interpret and generate meaningful actions based on the UI representations.

- **Domain-Specific Training.** Some agents enhance their knowledge by training on phone automation-specific datasets. Works such as Auto-GUI Zhang & Zhang (2023), CogAgent Hong et al. (2024), ScreenAI Baechler et al. (2024), CoCo-agent Ma et al. (2024), and Ferret-UI You et al. (2024) have trained LLMs on datasets tailored for mobile UI interactions, thereby improving their capability to understand and manipulate mobile interfaces effectively. For a more detailed discussion of knowledge acquisition through model training, see § 5.2.

- **Knowledge Injection.** Agents can enhance their decision-making by incorporating knowledge derived from exploratory interactions and stored contextual information. This involves utilizing data collected during offline exploration phases or from observed human demonstrations to inform the LLM's reasoning process. For instance, AutoDroid Wen et al. (2024) explores app functionalities and records UI transitions in a UI Transition Graph (UTG) memory, which are then used to generate task-specific prompts for the LLM. Similarly, AppAgent Zhang et al. (2023a) compiles knowledge from autonomous interactions and human demonstrations into structured documents, enabling the LLM to make informed decisions based on comprehensive UI state information and task requirements. AppAgent v2 Li et al. (2024c) introduces a more efficient mechanism for knowledge base construction and updating. It leverages Retrieval-Augmented Generation (RAG) technology to achieve real-time dynamic updates of knowledge base information. This significantly enhances the agent's adaptability in new environments. AppAgentX Jiang et al. (2025) introduces an evolutionary mechanism that enables dynamic learning from past interactions and replaces inefficient low-level operations with high-level actions. Other similar works include AdaptAgent Verma et al. (2024), Mobile-Agent-V Wang et al. (2025a), LearnAct Liu et al. (2025a) and others. Further advancing the concept of online adaptation, Mirage-1 Xie et al. (2025b) addresses the challenge of long-horizon tasks and the domain gap between offline and online environments. It introduces a Hierarchical Multimodal Skills module that abstracts offline trajectories into a structured, hierarchical knowledge base of execution, core, and meta skills. To adapt to dynamic online settings, Mirage-1 employs a Skill-Augmented Monte Carlo Tree Search algorithm. This algorithm leverages the HMS knowledge to guide online exploration, enabling the agent to efficiently acquire and update its skills, thereby bridging the offline-online domain gap.

### 4.2.2 Decision Making

Decision Making is the process by which the agent determines the appropriate actions to perform based on the current perception and stored information Xi et al. (2023b). The LLM processes the input prompts, which include the current UI state, historical interactions from memory, and relevant knowledge, to generate action sequences that accomplish the assigned tasks.

**Planning** involves devising a sequence of actions to achieve a specific task goal Song et al. (2023a); Xi et al. (2023b). Effective planning is essential for decomposing complex tasks into manageable steps and adapting to changes in the environment. For instance, Mobile-Agent-v2 Wang et al. (2024a) incorporates a planning agent that generates task progress based on historical operations, ensuring effective operation generation by the decision agent. Additionally, approaches like Dynamic Planning of Thoughts (D-PoT) have been proposed to dynamically adjust plans based on environmental feedback and action history, significantly improving accuracy and adaptability in task execution Zhang et al. (2024f). Simultaneously, by reducing the number of

calls to LLMs and employing a phased planning strategy, the agent can plan all actions in a given state at once, thereby enhancing planning efficiency Li et al. (2023d).

**Reasoning** enables the agent to interpret and analyze information to make informed decisions Gandhi et al. (2024); Chen et al. (2024g); Plaat et al. (2024). It involves understanding the context, evaluating possible actions, and selecting the most appropriate ones to achieve the desired outcome. By leveraging chain-of-thought(COT) Wei et al. (2022), LLMs enhance their reasoning capabilities, allowing them to think step-by-step and handle intricate decision-making processes. This structured approach facilitates the generation of coherent and logical action sequences, ensuring that the agent can navigate complex UI interactions effectively. The best-first tree search algorithm is utilized in real-world environments to iteratively construct, explore, and prune trajectory graphs, thereby enhancing the reasoning and decision-making capabilities of agents. A value function serves as a reward signal to guide agents in conducting efficient searches Koh et al. (2024b). Additionally, research indicates that LLMs to estimate the latent states of agents, in combination with reasoning methods, can further improve the agents' reasoning performance Bishop et al. (2024).

**Reflection** allows the agent to assess the outcomes of its actions and make necessary adjustments to improve performance Shinn et al. (2024). It involves evaluating whether the executed actions meet the expected results and identifying any discrepancies or errors. For example, Mobile-Agent-v2 Wang et al. (2024a) includes a reflection agent that evaluates whether the decision agent's operations align with the task goals. If discrepancies are detected, the reflection agent generates appropriate remedial measures to correct the course of action. This continuous feedback loop enhances the agent's reliability and ensures that it can recover from unexpected states or errors during task execution. Furthermore, structured self-reflection identifies initial erroneous actions, which prevents agents from repeating the same mistakes. It also draws on reflective memory to avoid known unsuccessful actions Li et al. (2023d). Additionally, regular reflection through automated evaluation methods significantly enhances the performance of agents Pan et al. (2024); Duan et al. (2024).

By integrating robust planning, advanced reasoning, and reflective capabilities, the Decision Making component of the Brain ensures that MLLM-powered phone GUI agents can perform tasks intelligently and adaptively. These mechanisms enable the agents to handle a wide range of scenarios, maintain task continuity, and improve their performance over time through iterative learning and adjustment.

Table 2: Types of actions in phone GUI agents

| Action Type | Description |
| --- | --- |
| **Touch Interactions** | **Tap:** Select a specific UI element.
**Double Tap:** Quickly tap twice to trigger an action.
**Long Press:** Hold a touch for extended interaction, triggering contextual options or menus. |
| **Gesture-Based Actions** | **Swipe:** Move a finger in a direction (left, right, up, down).
**Pinch:** Zoom in/out by bringing fingers together/apart.
**Drag:** Move UI elements to a new location. |
| **Typing and Input** | **Type Text:** Enter text into input fields.
**Select Text:** Highlight text for editing or copying. |
| **System Operations** | **Launch Application:** Open a specific app.
**Change Settings:** Modify system settings (e.g., Wi-Fi, brightness).
**Navigate Menus:** Access app sections or system menus. |
| **Media Control** | **Play/Pause:** Control media playback.
**Adjust Volume:** Increase or decrease device volume. |

### 4.3 Action in Phone GUI Agents

The Action component is a critical part of MLLM-powered phone GUI agents, responsible for executing decisions made by the Brain within the mobile environment. By bridging high-level commands generated by the LLM with low-level device operations, the agent can effectively interact with the phone's UI and system functionalities. Actions encompass a wide variety of operations, ranging from simple interactions like tapping a button to complex tasks such as launching applications or modifying device settings. Execution mechanisms leverage tools like Android's UI Automator Patil et al. (2016), iOS's XCTest Lodi (2021), or popular automation frameworks such as Appium Singh et al. (2014) and Selenium Gundecha (2015); Sinclair to send precise commands to the phone. Through these mechanisms, the agent ensures that decisions are translated into tangible, reliable operations on the device.

The types of actions in phone GUI agents are diverse and can be broadly categorized based on their functionalities. Table 2 summarizes these actions, providing a clear overview of the operations agents can perform.

The above categories reflect the key interactions required for phone automation. Touch interactions form the foundation of UI navigation, while gesture-based actions add flexibility for dynamic control. Typing and input enable text-based operations, whereas system operations and media controls extend the agent's capabilities to broader device functionalities. By combining these actions, phone GUI agents can achieve high accuracy and adaptability in executing user tasks, ensuring a seamless experience even in complex and dynamic environment.

### 4.4 Multi-Agent Framework

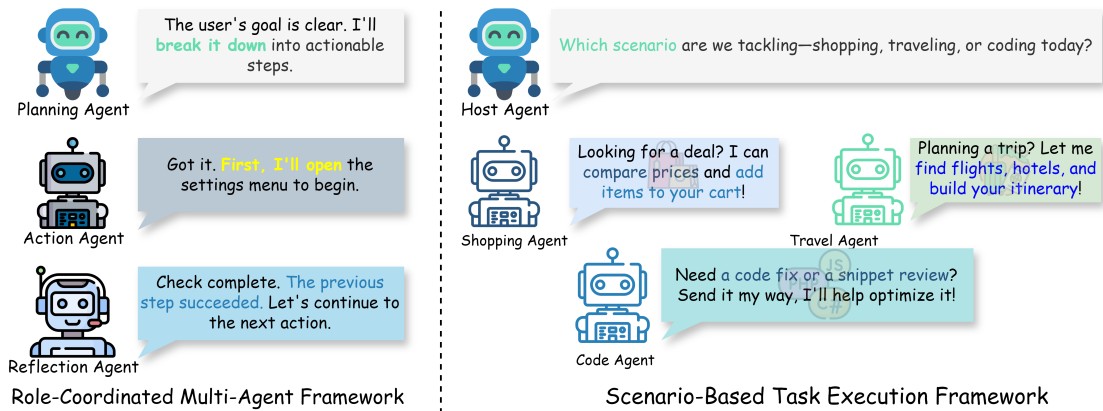

Figure 6: Comparison of the role-coordinated and scenario-based multi-agent frameworks. The Role-Coordinated framework organizes agents based on general functional roles with a fixed workflow, while the Scenario-Based framework dynamically assigns tasks to specialized agents tailored for specific scenarios, allowing for increased flexibility and adaptability in handling diverse tasks.

While single-agent frameworks based on LLMs have achieved significant progress in screen understanding and reasoning, they operate as isolated entitiesTorreno et al. (2017); Dorri et al. (2018); Gong et al. (2023). This isolation limits their flexibility and scalability in complex tasks that may require diverse, coordinated skills and adaptive capabilities. Single-agent systems may struggle with tasks that demand continuous adjustments based on real-time feedback, multi-stage decision-making, or specialized knowledge in different domains. Furthermore, they lack the ability to leverage shared knowledge or collaborate with other agents, reducing their effectiveness in dynamic environment Xi et al. (2023b); Wang et al. (2024a); Tan et al.; Song et al. (2024c).

Multi-agent frameworks address these limitations by facilitating collaboration among multiple agents, each with specialized functions or expertise Chen et al. (2019); Talebirad & Nadiri (2023); Wu et al. (2023); Chen et al. (2023); Li et al. (2023b); Liu et al. (2024e); Li et al. (2024b); Tran et al. (2025). This collaborative

approach enhances task efficiency, adaptability, and scalability, as agents can perform tasks in parallel or coordinate their actions based on their specific capabilities. As illustrated in Figure 6, multi-agent frameworks in phone automation can be categorized into two primary types: the **Role-Coordinated Multi-Agent Framework** and the **Scenario-Based Task Execution Framework**. These frameworks enable more flexible, efficient, and robust solutions in phone automation by either organizing agents based on general functional roles or dynamically assembling specialized agents according to specific task scenarios.

### 4.4.1 Role-Coordinated Multi-Agent

In the Role-Coordinated Multi-Agent Framework, agents are assigned general functional roles such as planning, decision-making, memory management, reflection, or tool invocation. These agents collaborate through a predefined workflow, with each agent focusing on its specific function to collectively achieve the overall task. This approach is particularly beneficial for tasks that require a combination of these general capabilities, allowing each agent to specialize and optimize its role within the workflow.

For example, in MMAC-Copilot Song et al. (2024c), multiple agents with distinct general functions collaborate as an OS copilot. The **Planner** strategically manages and allocates tasks to other agents, optimizing workflow efficiency. Meanwhile, the **Librarian** handles information retrieval and provides foundational knowledge, and the **Programmer** is responsible for coding and executing scripts, directly interacting with the software environment. The **Viewer** interprets complex visual information and translates it into actionable commands, while the **Video Analyst** processes and analyzes video content. Additionally, the **Mentor** offers strategic oversight and troubleshooting support. Each agent contributes its specialized function to the collaborative workflow, thereby enhancing the system's overall capability to handle complex interactions with the operating system.

Similarly, in Mobile-Agent-v2 Wang et al. (2024a), three agents with general roles are utilized: a planning agent, a decision agent, and a reflection agent. The **planning agent** compresses historical actions and state information to provide a concise representation of task progress. The **decision agent** uses this information to navigate the task effectively, while the **reflection agent** monitors the outcomes of actions and corrects any errors, ensuring accurate task completion. This role-based collaboration reduces context length, improves task progression, and enhances focus content retention through a memory unit managed by the decision agent.

In contrast, Mobile-Agent-E Wang et al. (2025d) decomposes tasks into high-level planning and low-level action execution, creating a system with a Manager Agent responsible for high-level planning and four subordinate agents: the Perceptor Agent, Operator Agent, Action Reflector Agent, and Notetaker Agent. The **Perceptor Agent** is responsible for fine-grained visual perception. The **Operator Agent** determines the next specific actions based on task and perception information. The **Action Reflector Agent** checks the screenshots before and after operations to verify if the expected outcomes are achieved and provides feedback to the Manager and Operator Agents. The **Notetaker Agent** extracts task-related information for use in subsequent steps. Additionally, Mobile-Agent-E incorporates a Self-Evolution Module, using two specialized agents, AES and AET, to update long-term memory after each task completion. **AES** summarizes lessons learned, while **AET** records reusable operational sequences, helping the agent in efficiently completing common subtasks and making better decisions in similar future tasks.

CHOP Zhou et al. (2025b) introduces a mobile operating assistant with Constrained High-frequency Optimized subtask Planning. This approach addresses challenges in the subtask level, which links high-level goals with low-level executable actions. CHOP overcomes VLM's deficiency in GUI scenario planning by using human-planned subtasks as basis vectors, significantly improving both effectiveness and efficiency across multiple applications in both English and Chinese contexts. The framework specifically targets two common issues: ineffective subtasks that lower-level agents cannot execute and inefficient subtasks that fail to contribute to higher-level task completion.

The principles of role-coordinated multi-agent systems were extensively explored first in general computer automation, pioneering architectural insights that are now being adapted for mobile agents. In general computer automation, Cradle Tan et al. leverages foundational agents with general roles to achieve versatile computer control. It provides a key example of this division-of-labor approach, where agents specialize in

functions like planning or perception. This architectural blueprint is directly applicable to mobile agents for tackling complex tasks that require a similar separation of concerns. Additionally, studies such as Ask-before-Plan Zhang et al. (2024g), PromptRPA Huang et al. (2024a), LUMOS Yin et al. (2024), and WebPilot Zhang et al. (2024h) also utilize general-purpose role agents to execute tasks and excel in complex tasks like planning. Among these, LUMOS provides high-quality training data and methods for future intelligent agent research. Agent S2 Agashe et al. (2025) presents a compositional generalist-specialist framework for computer use agents that delegates cognitive responsibilities across various models. It introduces a Mixture-of-Grounding technique for precise GUI localization and Proactive Hierarchical Planning that dynamically refines action plans at multiple temporal scales based on evolving observations. MobileUse Li et al. (2025a) introduces a role-coordinated framework centered on error recovery and adaptation. It features an Operator for execution, a Progressor for summarizing task status, Hierarchical Reflectors for multi-level verification, and a Proactive Explorer to gather knowledge in unfamiliar environments. The core of its design is a hierarchical reflection mechanism that checks for errors from the individual step to the overall task level, combined with a Reflection-on-Demand strategy that balances thoroughness with efficiency. Mobile-Agent-v3 Ye et al. (2025a) proposes a framework built upon a foundational agent model, GUI-Owl, which is specifically post-trained for GUI automation. Within the Mobile-Agent-v3 framework, this foundational model is instantiated into four distinct roles: a Manager Agent for high-level strategic planning and dynamic adaptation of subgoals; a Worker Agent that executes the tactical operations on the GUI; a Reflector Agent that provides self-correction by evaluating action outcomes and generating feedback; and a Notetaker Agent that preserves contextual memory by storing critical information from the screen. This modular architecture is designed to enhance long-horizon task automation through coordinated planning, execution, and reflection.

### 4.4.2 Scenario-Based Task Execution

In the Scenario-Based Task Execution Framework, tasks are dynamically assigned to specialized agents based on specific task scenarios or application domains. Each agent is endowed with capabilities tailored to a particular scenario, such as shopping, code editing, or navigation. By assigning tasks to agents specialized in the relevant domain, the system improves task success rates and efficiency.

For instance, MobileExperts Zhang et al. (2024b) forms different expert agents through an **Expert Exploration phase**. In the exploration phase, each agent receives tailored tasks broken down into sub-tasks to streamline the exploration process. Upon completion of a sub-task, the agent extracts three types of memories from its trajectory: interface memories, procedural memories (tools), and insight memories for use in subsequent execution phases. When a **new task** arrives, the system dynamically forms an expert team by **selecting agents whose expertise matches the task requirements**, enabling them to collaboratively execute the task more effectively. Similarly, in the SteP Sodhi et al. (2024) framework, agents are specialized based on **specific web scenarios** such as shopping, GitLab, maps, Reddit, or CMS platforms. While web-focused, this framework is a pioneering example of the scenario-based specialization approach. It validates the concept of dynamically assigning tasks to different "expert" agents, a crucial paradigm for mobile automation where an agent must handle a wide variety of functionally distinct applications. Each scenario agent possesses specific capabilities and knowledge relevant to its domain. When a task is received, it is dynamically assigned to the appropriate scenario agent, which executes the task leveraging its specialized expertise. This approach enhances flexibility and adaptability, allowing the system to handle a wide range of tasks across different domains more efficiently.

Through dynamic task assignment and specialization, the Scenario-Based Task Execution Framework optimizes multi-agent systems to adapt to diverse and evolving contexts, significantly enhancing both the efficiency and effectiveness of task execution. As illustrated in Figure 6, the Role-Coordinated Framework relies on agents with general functional roles collaborating through a fixed workflow, suitable for tasks requiring a combination of general capabilities. In contrast, the Scenario-Based Framework dynamically assigns tasks to specialized agents tailored to specific scenarios, providing a flexible structure that adapts to the varying complexity and requirements of real-world tasks.

Despite the potential of multi-agent frameworks in phone automation, several challenges remain. In the Role-Coordinated Framework, coordinating agents with general functions requires efficient workflow design and may introduce overhead in communication and synchronization. In the Scenario-Based Framework,

maintaining and updating a diverse set of specialized agents can be resource-intensive, and dynamically assigning tasks requires effective task recognition and agent selection mechanisms. Future research could explore hybrid frameworks that combine the strengths of both approaches, leveraging general functional agents while also incorporating specialized scenario agents as needed. Additionally, developing advanced algorithms for agent collaboration, learning, and adaptation can further enhance the intelligence and robustness of multi-agent systems. Integrating external knowledge bases, real-time data sources, and user feedback can also improve agents' decision-making capabilities and adaptability in dynamic environment.

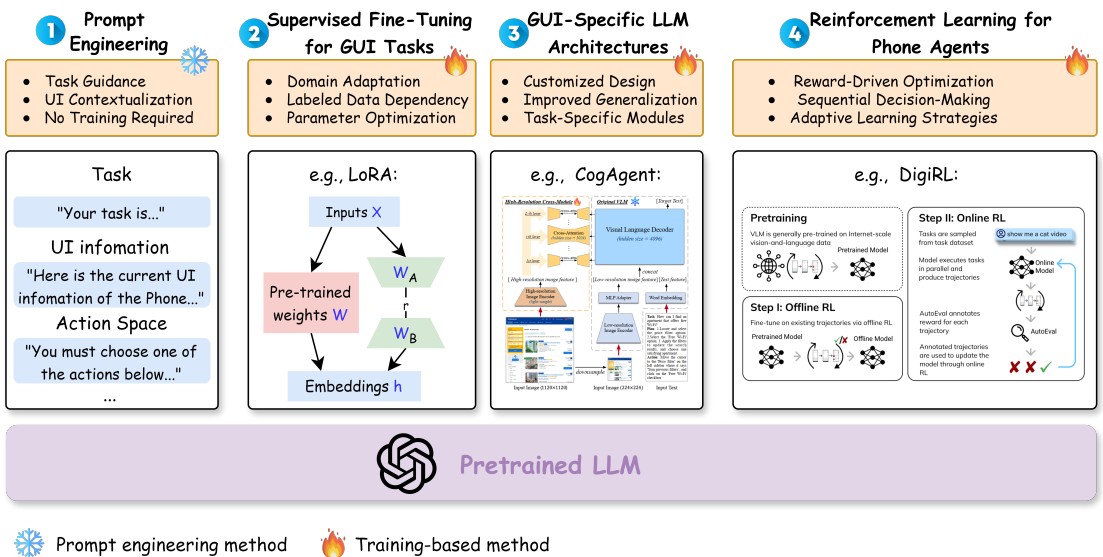

Figure 7: Differences between training-based methods and prompt engineering in phone automation. Training-based methods adapt the model's parameters through additional training, enhancing its ability to perform specific tasks, whereas prompt engineering leverages the existing capabilities of pre-trained models by guiding them with well-designed prompts.

## 4.5 Plan-Then-Act Framework

While single-agent and multi-agent frameworks enhance adaptability and scalability, some tasks benefit from explicitly separating high-level planning from low-level execution. This leads to what we term the *Plan-Then-Act Framework*. In this paradigm, the agent first formulates a conceptual plan—often expressed as human-readable instructions—before grounding and executing these instructions on the device's UI.

The Plan-Then-Act approach addresses a fundamental challenge: although LLMs and multimodal LLMs (MLLMs) excel at interpreting instructions and reasoning about complex tasks, they frequently struggle to precisely map their textual plans to concrete UI actions. By decoupling these stages, the agent can focus on **what should be done (planning)** and then handle **how to do it on the UI (acting)**. Recent works highlight the effectiveness of this approach:

- SeeAct Zheng et al. (2024a) demonstrates that GPT-4V(ision)Achiam et al. (2023) can generate coherent plans for navigating websites. However, bridging the gap between textual plans and underlying UI elements remains challenging. By clearly delineating planning from execution, the system can better refine its plan before finalizing actions.

- UGround Gou et al. (2024) and related efforts You et al. (2024); Zhang et al. (2024d) emphasize advanced visual grounding. Under a Plan-Then-Act framework, the agent first crafts a task solution plan, then relies on robust visual grounding models to locate and manipulate UI components. This modular design enhances performance across diverse GUIs and platforms, as the grounding model can evolve independently of the planning mechanism.

• LiMAC (Lightweight Multi-modal App Control) Christianos et al. (2024) also embodies a Plan-Then-Act spirit. LiMAC's Action Transformer (AcT) determines the required action type (the plan), and a specialized VLM is invoked only for natural language needs. By structuring decision-making and text generation into distinct stages, LiMAC improves responsiveness and reduces compute overhead, ensuring that reasoning and UI interaction are cleanly separated.

• ClickAgent Hoscilowicz et al. (2024) similarly employs a two-phase approach. The MLLM handles reasoning and action planning, while a separate UI location model pinpoints the relevant coordinates on the screen. Here, the MLLM's plan of which element to interact with is formed first, and only afterward is the element's exact location identified and the action executed.

• Ponder & Press Wang et al. (2024j) employs a general MLLM to decompose user instructions into executable actions. It then uses a GUI-specific MLLM to map the target elements in the action descriptions to pixel coordinates, thereby constructing a Plan-Then-Act Framework based solely on visual input. This framework is adaptable across various software environments without relying on supplementary information such as HTML or UI Trees.

• To improve the reliability of the planning phase, SPlanner Mo et al. (2025) introduces a planning module that leverages Extended Finite State Machine (EFSM). By modeling an application's core interaction logic as an EFSM, the agent generates a complete and stable execution path from a start state to a goal. This path is then translated into a natural language plan that guides a separate VLM executor, effectively separating the high-level strategic planning from the low-level visual grounding and execution.

The Plan-Then-Act Framework offers several advantages. Modularity allows improvements in planning without requiring changes to the UI grounding and execution modules, and vice versa. Error Mitigation enables the agent to revise its plan before committing to actions; if textual instructions are ambiguous or infeasible, they can be corrected, reducing wasted actions and improving reliability. Additionally, improved visual grounding models, OCR enhancements, and scenario-specific knowledge can further refine the Plan-Then-Act approach, making agents more adept at handling intricate, real-world tasks. In summary, the Plan-Then-Act Framework represents a natural evolution in designing MLLM-powered phone GUI agents. By separating planning from execution, agents can achieve clearer reasoning, improved grounding, and ultimately more effective and reliable task completion.

## 4.6 Comparative Analysis of Frameworks

Rather than viewing the Single-Agent, Multi-Agent, and Plan-Then-Act paradigms as mutually exclusive competitors, a more insightful analysis examines the trade-offs and future research directions within the core components that constitute any mobile agent: **Perception**, **Brain**, and **Action**. The optimal framework is not a fixed architecture but rather a dynamic assembly of the most suitable components for a given task. Table 3 summarizes this component-wise analysis.

**Perception: The Trade-off Between Richness and Speed.** Current agents leverage a rich set of perceptual inputs, from structured **UI trees** Wen et al. (2023) to visual **screenshots** Zhang & Zhang (2023), often augmented with **SoM** Yang et al. (2023) and **OCR** Wang et al. (2024a). While this fusion provides a comprehensive understanding of the UI, it comes at the cost of high latency. Processing a high-resolution screenshot combined with a verbose UI tree for every single step is slow and computationally expensive. Future work must focus on more efficient multi-modal fusion techniques that can extract the most salient information without sacrificing speed, while also addressing the current inability to perceive dynamic content like animations or embedded videos.

**Brain: The Trilemma of Speed, Memory, and Cost.** The cognitive core of agents has evolved significantly, from simple reactive models to sophisticated reasoning with **CoT** and reflection Wang et al. (2024a). However, this advancement presents a critical trilemma for practical deployment.

• **Acceleration Methods:** A primary challenge is the slow pace of execution. Single-step task execution, which can take several seconds to tens of seconds due to model inference and perception processing, is not viable for real-world user interaction. Future research must prioritize acceleration, exploring techniques

Table 3: Comparative analysis of core components in mobile agent frameworks.

| Component | Current State & Strengths | Limitations & Challenges | Future Directions |
|---|---|---|---|
| **Perception** | Fusion of visual (screenshots) and structural (UI trees) data, enhanced with SoM and OCR for rich context. | • High latency from processing large inputs.
• Brittleness of UI tree parsers.
• Inability to understand dynamic content (videos, animations). | • Efficient multi-modal fusion models.
• Real-time perception for dynamic content.
• Robust parsing resilient to UI changes. |
| **Brain (Cognition)** | Step-by-step reasoning (CoT, CoAT); distributed cognition in multi-agent systems; session-based short-term memory. | • **Speed:** Slow, step-by-step reasoning is impractical for users.
• **Memory:** Lacks long-term memory of user habits.
• **Cost:** High computational/financial cost of powerful models. | • Acceleration via model distillation and caching.
• Persistent long-term memory for personalization.
• Adaptive agent architectures (single vs. multi-agent). |
| **Action** | Discrete and low-level action space (e.g., tap, swipe, type) executed via device controllers like UI Automator. | • Inability to perform complex or continuous gestures.
• Actions are not human-like. | • More expressive and fine-grained action spaces.
• Learning complex action primitives.
• Generating human-like, continuous touch event sequences. |

like model distillation to create smaller, faster specialist models, caching common UI state representations, or developing speculative execution mechanisms.

- **Perfecting Memory for User-Centric Adaptation:** Current agents, such as *Mobile-Agent-v2* Wang et al. (2024a), possess effective short-term memory, retaining context within a single task session. However, they lack persistent, **long-term memory**, which is the bedrock of true personalization. A truly useful agent must be user-centric, remembering individual preferences, learning from past mistakes across sessions, and adapting its behavior to a user's unique habits. The path to such adaptation lies in demonstration-based learning. As shown by *AdaptAgent* Verma et al. (2024), which adapts to new domains from just a few examples, and *LearnAct* Liu et al. (2025a), which extracts knowledge from human demonstrations, the future of agent memory is not just about storing data, but about creating personalized models of user behavior from minimal input.
- **Single vs. Multi-Agent Trade-offs:** Multi-agent frameworks like *MMAC-Copilot* Song et al. (2024c) often achieve a higher task success rate through specialization. However, this advantage comes at the cost of increased latency from inter-agent communication and higher operational costs from multiple concurrent model calls. A promising direction is the development of adaptive or hybrid frameworks that can dynamically scale, using a lightweight single agent for simple tasks and dispatching a team of specialized agents only when complexity demands it.

**Action: The Need for a More Refined Action Space.** The action space in most current agents is functional but rudimentary, typically limited to simple operations like 'tap', 'swipe', and 'type'. While sufficient for many tasks, this limited dexterity prevents agents from performing more complex interactions that are trivial for humans, such as precise drag-and-drop, pinch-to-zoom, or other multi-touch gestures.

The future lies in creating more **expressive and fine-grained action spaces**. This could involve agents learning a library of complex action primitives or even generating low-level, continuous touch event sequences to better mimic human-like interaction and dexterity.

## 5 LLMs for Phone Automation

LLMs Radford (2018); Radford et al. (2019); Brown (2020); Achiam et al. (2023) have emerged as a transformative technology in phone automation, bridging natural language inputs with executable actions. By leveraging their advanced language understanding, reasoning, and generalization capabilities, LLMs enable agents to interpret complex user intents, dynamically interact with diverse mobile applications, and effectively manipulate GUIs.

In this section, we explore two primary approaches to leveraging LLMs for phone automation: **Training-Based Methods** and **Prompt Engineering**. Figure 7 illustrates the differences between these two approaches in the context of phone automation. **Training-Based Methods** involve adapting LLMs specifically for phone automation tasks through techniques like supervised fine-tuning(SFT) Cheng et al. (2024); Chen et al. (2024d); Lu et al. (2024a); Pawlowski et al. (2024) and reinforcement learning Song et al. (2024a); Bai et al. (2024); Wang et al. (2024h). These methods aim to enhance the models' capabilities by training them on GUI-specific data, enabling them to understand and interact with GUIs more effectively. **Prompt Engineering**, on the other hand, focuses on designing input prompts to guide pre-trained LLMs to perform desired tasks without additional training Wei et al. (2022); Yao et al. (2024); Chen et al. (2022). By carefully crafting prompts that include relevant information such as task descriptions, interface states, and action histories, users can influence the model's behavior to achieve specific automation goals Wen et al. (2023); Zhang et al. (2023a); Song et al. (2023b).

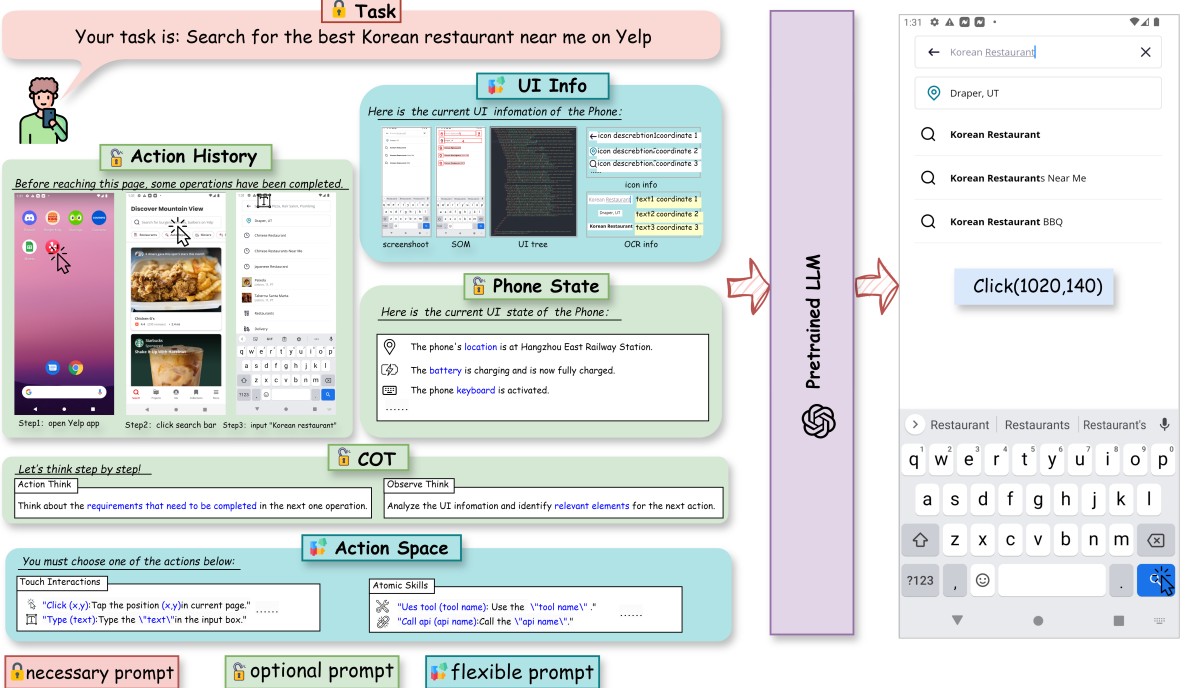

Figure 8: Schematic of prompt engineering for phone automation. The **necessary prompt** is mandatory, initiating the task, e.g., searching for a Korean restaurant. The **optional prompt** are supplementary, enhancing tasks without being mandatory. The **flexible prompt** must include one or more elements from the UI Info, like a screenshot or OCR info, to adapt to task needs.

## 5.1 Prompt Engineering

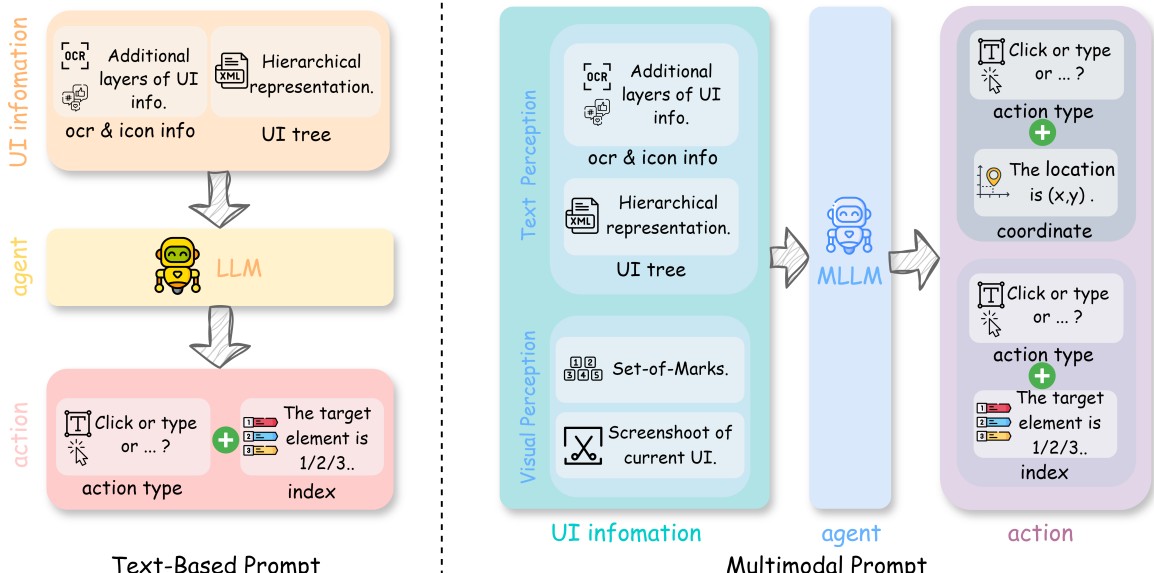

Figure 9: Comparison between text-based prompt and multimodal prompt. In Text-Based Prompt, the LLM processes textual UI information, such as UI tree structures and OCR data, to determine the action type (index). In contrast, Multimodal Prompt integrates screenshot data with supplementary UI information to facilitate decision-making by the agent. The MLLM can then pinpoint the action location using either coordinates or indices.

LLMs like the GPT series Radford (2018); Radford et al. (2019); Brown (2020) have demonstrated remarkable capabilities in understanding and generating human-like text. These models have revolutionized natural language processing by leveraging massive amounts of data to learn complex language patterns and representations.

Prompt engineering is the practice of designing input prompts to effectively guide LLMs to produce desired outputs for specific tasks Wei et al. (2022); Yao et al. (2024); Chen et al. (2022). By carefully crafting the prompts, users can influence the model's behavior without the need for additional training or fine-tuning. This approach allows for leveraging the general capabilities of pre-trained models to perform a wide range of tasks by simply providing appropriate instructions or examples in the prompt.

In the context of phone automation, prompt engineering enables the utilization of general-purpose LLMs to perform automation tasks on mobile devices. Recently, a plethora of works have emerged that apply prompt engineering to achieve phone automation Wen et al. (2023); Yan et al. (2023); Zhang et al. (2023a); Wang et al. (2024b;a); Zhang et al. (2024b); Lu et al. (2024b); Song et al. (2023b); Taeb et al. (2024); Yang et al. (2024c); Liu et al. (2024d); Huang et al. (2024a). These works leverage the strengths of LLMs in natural language understanding and reasoning to interpret user instructions and generate corresponding actions on mobile devices.

The fundamental approach to achieving phone automation through prompt engineering entails the creation of prompts that encapsulate a comprehensive set of information. These prompts should include a detailed task description, such as searching for the best Korean restaurant on Yelp. They also integrate the current UI information of the phone, which may encompass screenshots, SoM, UI tree structures, icon details, and OCR data. Additionally, the prompts should account for the phone's real-time state, including its location, battery level, and keyboard status, as well as any pertinent action history and the range of possible actions (action space). The COT prompt Wei et al. (2022); Zhang et al. (2023c) is also a crucial component, guiding the thought process for the next operation. The LLM then analyzes this rich prompt and determines the subsequent action to execute. This methical process is vividly depicted in Figure 8.

This section explores the application of prompt engineering in phone automation, categorizing related works based on the type of prompts used: **Text-Based Prompt** and **Multimodal Prompt**. As illustrated in Figure 9, the approach to automation significantly diverges between these two prompt types. Table 4 summarizes notable methods, highlighting their main UI information, the type of model used, and other relevant details such as task types and grounding strategies.

Table 4: Summary of prompt engineering methods for phone GUI agents

| Method | Date | Task Type | Model | Screenshot | SoM | UI tree | Icon & OCR | Grounding |
|---|---|---|---|---|---|---|---|---|
| **DroidBot-GPT** Wen et al. (2023) ○ | 2023.04 | General | ChatGPT | ✗ | ✗ | ✔ | ✗ | Index |
| **Enabling conversational** Wang et al. (2023b) | 2023.04 | Screen Under-standing, QA | PaLM | ✗ | ✗ | ✔ | ✗ | Index |
| **AutoDroid** Wen et al. (2024) ○ | 2023.09 | General | GPT-4, GPT-3.5 | ✗ | ✗ | ✔ | ✗ | Index |
| **MM-Navigator** Yan et al. (2023) ○ | 2023.11 | General | GPT-4V | ✔ | ✗ | ✗ | ✔ | Index |
| **VisionTasker** Song et al. (2024b) ○ | 2023.12 | Manual Teaching | GPT-4 | ✗ | ✔ | ✔ | ✔ | Index |
| **AppAgent** Zhang et al. (2023a) ○ | 2023.12 | General | GPT-4 | ✔ | ✔ | ✔ | ✔ | Index |
| **MobileGPT** Lee et al. (2023) ○ | 2023.12 | General | GPT-3.5, GPT-4 | ✗ | ✗ | ✔ | ✗ | Index |
| **Mobile-Agent** Wang et al. (2024b) ○ | 2024.01 | General | GPT-4V | ✔ | ✗ | ✗ | ✔ | Coordinate |
| **AXNav** Taeb et al. (2024) | 2024.05 | Bug Testing | GPT-4 | ✗ | ✗ | ✔ | ✔ | Index |
| **Mobile-Agent-v2** Wang et al. (2024a) ○ | 2024.06 | General | GPT-4V | ✔ | ✗ | ✗ | ✔ | Coordinate |
| **GUI Narrator** Wu et al. (2024b) ○ | 2024.06 | GUI Video Captioning | GPT-4o, QwenVL-7B | ✔ | ✔ | ✗ | ✗ | Index |
| **MobileExpert** Zhang et al. (2024b) | 2024.07 | General | GPT-4V | ✔ | ✗ | ✗ | ✗ | Coordinate |
| **VisionDroid** Liu et al. (2024d) ○ | 2024.07 | Non-Crash Func-tional Bug Detection | GPT-4 | ✔ | ✔ | ✔ | ✗ | Index |
| **AppAgent v2** Li et al. (2024c) | 2024.08 | General | GPT-4 | ✔ | ✔ | ✔ | ✔ | Coordinate, Index |
| **OmniParser** Lu et al. (2024b) | 2024.08 | General | GPT-4V | ✔ | ✔ | ✗ | ✔ | Index |
| **Mobile-Agent-E** Wang et al. (2025d) ○ | 2025.01 | General | GPT-4o, Claude -3.5, Gemini-1.5 | ✔ | ✗ | ✗ | ✔ | Coordinate |
| **Mobile-Agent-V** Wang et al. (2025a) ○ | 2025.02 | General | GPT-4o | ✔ | ✗ | ✗ | ✔ | Coordinate |
| **LearnAct** Liu et al. (2025a) ○ | 2025.02 | General | Gemini-1.5 | ✔ | ✗ | ✗ | ✗ | Coordinate |
| **MobileUse** Li et al. (2025a) ○ | 2025.07 | General | Qwen2.5-VL-72B-Instruct | ✔ | ✗ | ✗ | ✗ | Coordinate |
| **Mobile-Agent-v3** Ye et al. (2025a) ○ | 2025.08 | General | GUI-Owl-32B | ✔ | ✗ | ✗ | ✗ | Coordinate |

### 5.1.1 Text-Based Prompting

In the domain of text-based prompt automation, the primary architecture involves a single text-modal LLM serving as the agent for mobile device automation. This agent operates by interpreting UI information presented in the form of a UI tree. It is important to note that, to date, the approaches discussed have primarily utilized UI tree data and have not extensively incorporated OCR text and icon information. We believe that solely relying on OCR and icon information is insufficient for fully representing screen UI information; instead, as demonstrated in Mobile-agent-v2 Wang et al. (2024a), they are best used as auxiliary information alongside screenshots. These text-based prompt agents make decisions by selecting elements from a list of candidates based on the textual description of the UI elements. For instance, to initiate a search, the LLM would identify and select the search button by its index within the UI tree rather than its screen coordinates, as depicted in Figure 9.

The study by Enabling Conversational Wang et al. (2023b) marked a significant step in this field. It explored the use of task descriptions, action spaces, and UI trees to map instructions to UI actions. However, it focused solely on the execution of individual instructions without delving into sequential decision-making processes. DroidBot-GPT Wen et al. (2023) is a landmark in applying pre-trained language models to app automation. It is the first to explore the use of LLMs for app automation without requiring modifications to the app or the model. DroidBot-GPT perceives UI trees, which are structural representations of the app's UI, and integrates user-provided tasks along with action spaces and output requirements. This allows the model to engage in sequential decision-making and automate tasks effectively. AutoDroid Wen et al. (2024)

takes this concept further. It employs a UI Transition Graph (UTG) generated through random exploration to create an App Memory. This memory, combined with the commonsense knowledge of LLMs, enhances decision-making and significantly advances the capabilities of phone GUI agents. MobileGPT Lee et al. (2023) introduces a hierarchical decision-making process. It simulates human cognitive processes—exploration, selection, derivation, and recall—to augment the efficiency and reliability of LLMs in mobile task automation. Lastly, AXNav Taeb et al. (2024) showcases an innovative application of Prompt Engineering in accessibility testing. AXNav interprets natural language instructions and executes them through an LLM, streamlining the testing process and improving the detection of accessibility issues, thus aiding the manual testing workflows of QA professionals.

Each of these contributions, while unique in their approach, is united by the common thread of Prompt Engineering. They demonstrate the versatility and potential of text-based prompt automation in enhancing the interaction between LLMs and mobile applications.

### 5.1.2 Multimodal Prompting

With the advancement of large pre-trained models, Multimodal Large Language Models (MLLMs) have demonstrated exceptional performance across various domains Achiam et al. (2023); Li et al. (2023c); Ye et al. (2023); Wang et al. (2023d); Bai et al. (2023); Liu et al. (2024a); Wang et al. (2024f); Chen et al. (2024f;e); Koh et al. (2024a); Zheng et al. (2023), significantly contributing to the evolution of phone automation. Unlike text-only models, multimodal models integrate visual and textual information, addressing limitations such as the inability to access UI trees, missing control information, and inadequate global screen representation. By leveraging screenshots for decision-making, multimodal models facilitate a more natural simulation of human interactions with mobile devices, enhancing both accuracy and robustness in automated operations.

The fundamental framework for multimodal phone automation is illustrated in Figure 9. Multimodal prompts integrate visual perception (*e.g., screenshots*) and textual information (*e.g., UI tree, OCR, and icon data*) to guide MLLMs in generating actions. The action outputs can be categorized into two methods: **SoM-Based Indexing Methods** and **Direct Coordinate Output Methods**. These methods define how the agent identifies and interacts with UI elements, either by referencing annotated indices or by pinpointing precise coordinates.

**SoM-Based Indexing Methods.** SoM-based methods involve annotating UI elements with unique identifiers within the screenshot, allowing the MLLM to reference these elements by their indices when generating actions. This approach mitigates the challenges associated with direct coordinate outputs, such as precision and adaptability to dynamic interfaces. MM-Navigator Yan et al. (2023) represents a breakthrough in zero-shot GUI navigation using GPT-4V Achiam et al. (2023). By employing SoM prompting Yang et al. (2023), MM-Navigator annotates screenshots through OCR and icon recognition, assigning unique numeric IDs to actionable widgets. This enables GPT-4V to generate indexed action descriptions rather than precise coordinates, enhancing action execution accuracy. Building upon the SoM-based approach, AppAgent Zhang et al. (2023a) integrates autonomous exploration and human demonstration observation to construct a comprehensive knowledge base. This framework allows the agent to navigate and operate smartphone applications through simplified action spaces, such as tapping and swiping, without requiring backend system access. Tested across 10 different applications and 50 tasks, AppAgent showcases superior adaptability and efficiency in handling diverse high-level tasks, further advancing multimodal phone automation. OmniParser Lu et al. (2024b) enhances the SoM-based method by introducing a robust screen parsing technique. It combines fine-tuned interactive icon detection models and functional captioning models to convert UI screenshots into structured elements with bounding boxes and labels. This comprehensive parsing significantly improves GPT-4V's ability to generate accurately grounded actions, ensuring reliable operation across multiple platforms and applications. GUI Narrator Wu et al. (2024b) utilizes video captioning to guide the VLM, aiding in the deeper understanding of GUI operations. The framework uses the mouse cursor as a visual prompt, highlighting it with a green bounding box to enhance the VLM's interpretative abilities with high-resolution screenshots. By extracting screenshots from before and after GUI actions occur in the video as keyframes, it provides temporal and spatial logic to the action screenshots. These are combined into prompts to further guide the VLM in producing accurate action descriptions, thereby improving its performance.

**Direct Coordinate Output Methods.** Direct coordinate output methods enable MLLMs to determine the exact (x, y) positions of UI elements from screenshots, facilitating precise interactions without relying on indexed references. This approach leverages the advanced visual grounding capabilities of MLLMs to interpret and interact with the UI elements directly. VisionTasker Song et al. (2024b) introduces a two-stage framework that combines vision-based UI understanding with LLM task planning. Utilizing models like YOLOv8 Varghese & Sambath (2024) and PaddleOCR Du et al. (2020), VisionTasker parses screenshots to identify widgets and textual information, transforming them into natural language descriptions. This structured semantic representation allows the LLM to perform step-by-step task planning, enhancing the accuracy and practicality of automated mobile task execution. The Mobile-Agent series Wang et al. (2024b;a) leverages visual perception tools to accurately identify and locate both visual and textual UI elements within app screenshots. Mobile-Agent-v1 utilizes coordinate-based actions, enabling precise interaction with UI elements. Mobile-Agent-v2 extends this by introducing a multi-agent architecture comprising planning, decision, and reflection agents. Mobile-Agent-E Wang et al. (2025d) optimizes the multi-agent architecture by detailing the responsibilities of each agent. It also introduces a long-term memory mechanism through the design of a Self-Evolution Module, which accumulates experience and enables agents to evolve, thereby enhancing adaptability to new tasks. MobileExperts Zhang et al. (2024b) advances the direct coordinate output method by incorporating tool formulation and multi-agent collaboration. This dynamic, tool-enabled agent team employs a dual-layer planning mechanism to efficiently execute multi-step operations while reducing reasoning costs by approximately 22%. By dynamically assembling specialized agents and utilizing reusable code block tools, MobileExperts demonstrates enhanced intelligence and operational efficiency in complex phone automation tasks. Unlike AppAgent, AppAgent v2 Li et al. (2024c) integrates parsers with visual features and employs UI element coordinates along with Index information, creating a more flexible action space. This allows the agent to manage dynamic interfaces and non-standard UI elements more adeptly, thereby enhancing its adaptability to various complex tasks. VisionDroid Liu et al. (2024d) applies MLLMs to automated GUI testing, focusing on detecting non-crash functional bugs through vision-based UI understanding. By aligning textual and visual information, VisionDroid enables the MLLM to comprehend GUI semantics and operational logic, employing step-by-step task planning to enhance bug detection accuracy. Evaluations across multiple datasets and real-world applications highlight VisionDroid's superior performance in identifying and addressing functional bugs.

While multimodal prompt strategies have significantly advanced phone automation by integrating visual and textual data, they still face notable challenges. Approaches that do not utilize SoM maps and instead directly output coordinates rely heavily on the MLLM's ability to accurately ground UI elements for precise manipulation. Although recent innovations Wang et al. (2024a); Zhang et al. (2024b); Liu et al. (2024d) have made progress in addressing the limitations of MLLMs' grounding capabilities, there remains considerable room for improvement. Enhancing the robustness and accuracy of UI grounding is essential to achieve more reliable and scalable phone automation.

## 5.2 Training-Based Models

The subsequent sections delve into these approaches, discussing the development of task-specific model architectures, supervised fine-tuning strategies and reinforcement learning techniques in both general-purpose and Phone UI-specific scenarios.

### 5.2.1 Task-Specific LLM-based Agents

To advance AI agents for phone automation, significant efforts have been made to develop Task Specific Model Architectures that are tailored to understand and interact with GUIs by integrating visual perception with language understanding. These models address unique challenges posed by GUI environment, such as varying screen sizes, complex layouts, and diverse interaction patterns. A summary of notable Task Specific Model Architectures is presented in Table 5, highlighting their main contributions, domains, and other relevant details.

**General-Purpose Models.** The general-purpose GUI-specific LLMs are designed to handle a wide range of tasks across different applications and interfaces. They focus on enhancing direct GUI interaction, high-

Table 5: Summary of task-specific model architectures

| Method | Date | Task Type | Backbone | Size | Contributions |
|---|---|---|---|---|---|
| **Auto-GUI** Zhang & Zhang (2023) ⦾ | 2023.09 | General | N/A | 60M / 200M / 700M | Direct screen interaction; Chain-of-action; Action histories and future plans |
| **CogAgent** Hong et al. (2024) ⦾ | 2023.12 | General | CogVLM | 18B | High-res input ($1120 \times 1120$); Specialized in GUI understanding |
| **WebVLN-Net** Chen et al. (2024c) ⦾ | 2023.12 | Screen Understanding, QA | N/A | N/A | Web navigation with visual and HTML content |
| **ScreenAI** Baechler et al. (2024) ⦾ | 2024.02 | Screen Understanding, QA | N/A | 4.6B | UI and infographic understanding; Flexible patching |
| **CoCo-Agent** Ma et al. (2024) ⦾ | 2024.02 | General | LLaVA (LLaMA-2-chat-7B, CLIP) | N/A | Comprehensive perception; Conditional action prediction; Enhanced automation |
| **Ferret-UI** You et al. (2024) ⦾ | 2024.04 | Screen Understanding, Referring | Ferret | N/A | "Any resolution" tech-niques; Precise referring and grounding |
| **LVG** Qian et al. (2024b) | 2024.06 | Screen Understanding, Grounding | SWIN Transformer, BERT | N/A | Visual UI grounding; Layout-guided contrastive learning |
| **Textual Foresight** Burns et al. (2024) ⦾ | 2024.06 | Screen Understanding, Referring | BLIP-2 | N/A | Predict UI state; UI representation learning |
| **MobileFlow** Nong et al. (2024) | 2024.07 | General | Qwen-VL-Chat | 21B | Hybrid visual encoders; Variable resolutions; Multilingual support |
| **UI-Hawk** Zhang et al. (2024d) | 2024.08 | Screen Understanding, Grounding | N/A | N/A | History-aware encoder; Screen stream processing; FunUI benchmark |
| **Ferret-UI 2** Li et al. (2024e) | 2024.10 | Screen Understanding, Referring | Ferret | N/A | Multi-platform; High-resolution encoding |
| **OS-Atlas** Wu et al. (2024d) ⦾ | 2024.10 | Screen Understanding, Grounding | Qwen2-VL, InternVL-2 | 4B / 7B | Grounding data synthesis; Largest GUI grounding corpus |
| **ShowUI** Lin et al. (2024) ⦾ | 2024.11 | General | Qwen2-VL | 2B | Visual tokens selection; Cross-modal understanding |
| **Aguvis** Xu et al. (2024c) ⦾ | 2024.12 | General | Qwen2-VL | 7B / 72B | Comprehensive data pipeline; Two-stage training;Cross-platform |
| **Aria-UI** Yang et al. (2024b) ⦾ | 2024.12 | Screen Understanding, Grounding | Aria | 3.9B | Diversified dataset pipeline; Multimodal dynamic action history |
| **UI-TARS** Qin et al. (2025) ⦾ | 2025.01 | General | Qwen2-VL | 2B / 7B / 72B | System-2 Reasoning; Online boot-strapping; Reflection tuning |
| **GUI-Bee** Fan et al. (2025) ⦾ | 2025.01 | Screen Understanding, Grounding | SeeClick, UIX-7B, Qwen-GUI | N/A | Model-Environment alignment; Self-exploratory Data |
| **V-Droid** Dai et al. (2025) | 2025.03 | General | Llama-3.1-8B | 8b | Verifier-driven framework |
| **MP-GUI** Wang et al. (2025e) ⦾ | 2025.03 | General | InternVL2-8B | 8B | Screen Understanding, Referring |
| **Seed1.5-VL** Guo & et al. (2025) | 2025.05 | General | N/A | N/A | General-purpose multimodal understanding and reasoning |
| **MiMo-VL** Team (2025) ⦾ | 2025.07 | General | N/A | N/A | On-policy Reinforcement Learning |
| **UI-Venus** Gu et al. (2025) ⦾ | 2025.08 | General | Qwen2.5-VL | 7B/72B | Comprehensive data cleaning protocols, Self-evolving framework |
| **GUI-Owl** Gu et al. (2025) ⦾ | 2025.08 | General | Qwen2.5-VL | 7B/32B | Trajectory-aware Relative Policy Optimization for online RL |

resolution visual recognition, and comprehensive perception to improve the capabilities of AI agents in understanding and navigating complex mobile GUIs. One significant challenge in this domain is enabling agents to interact directly with GUIs without relying on environment parsing or application-specific APIs, which can introduce inefficiencies and error propagation. To tackle this, Auto-GUI Zhang & Zhang (2023) presents a multimodal agent that directly engages with the interface. It introduces a chain-of-action technique that leverages previous action histories and future action plans, enhancing the agent's decision-making process and leading to improved performance in GUI control tasks. High-resolution input is essential for recognizing tiny UI elements and text prevalent in GUIs. CogAgent Hong et al. (2024) addresses this by employing both low-resolution and high-resolution image encoders within its architecture. Supporting input resolutions up to $1120 \times 1120$, CogAgent effectively recognizes small page elements and text. Understanding UIs and infographics requires models to interpret complex visual languages and design principles. ScreenAI Baechler et al. (2024) improves upon existing architectures by introducing a flexible patching strategy and a novel textual representation for UIs. During pre-training, this representation teaches the model to interpret UI elements effectively. Leveraging large language models, ScreenAI automatically generates training data at scale, covering a wide spectrum of tasks in UI and infographic understanding. Enhancing both perception and action response is crucial for comprehensive GUI automation. CoCo-Agent Ma et al. (2024) proposes two novel approaches: comprehensive environment perception (CEP) and conditional action prediction (CAP). CEP enhances GUI perception through multiple aspects, including visual channels (screenshots and detailed layouts) and textual channels (historical actions). CAP decomposes action prediction into determining the action type first, then identifying the action target conditioned on the action type. Addressing the need for effective GUI agents in applications featuring extensive Mandarin content, MobileFlow Nong et al. (2024) introduces a multimodal LLM specifically designed for mobile GUI agents. MobileFlow employs a hybrid visual encoder trained on a vast array of GUI pages, enabling it to extract and comprehend information across diverse interfaces. The model incorporates a Mixture of Experts (MoE) and specialized modality alignment training tailored for GUIs. ShowUI Lin et al. (2024) employs the UI-Guided visual tokens selection method, which randomly selects a subset of tokens from each component during training. This approach retains the original positional information while reducing redundant tokens by 33%, thereby accelerating training speed by 1.4 times. Furthermore, by using interleaved vision-language-action streaming combined with high-quality training data, it significantly improves the training speed and performance of GUI visual agents. Aguvis Xu et al. (2024c) employs a two-stage training method to enhance the generalization and efficiency of GUI agents. It uses single-step task data to train the model's grounding abilities and multi-step task data to develop the model's planning and reasoning capabilities. This approach significantly improves the overall performance of the agents. UI-TARS Qin et al. (2025) employs a more in-depth and structurally robust System-2 reasoning method, combined with online bootstrapping and reflection tuning strategies. This combination effectively assists the model in handling complex tasks in dynamic environments and continuously optimizes overall performance. V-Droid Dai et al. (2025) introduces a novel verifier-driven architecture where the LLM does not generate actions directly but instead scores and selects from a finite set of extracted actions, improving task success rates and significantly reducing latency. Collectively, these general-purpose Task Specific Model Architectures address key challenges in phone automation by enhancing direct GUI interaction, high-resolution visual recognition, comprehensive environment perception, and conditional action prediction. By leveraging multimodal inputs and innovative architectural designs, these models significantly advance the capabilities of AI agents in understanding and navigating complex mobile GUIs, paving the way for more intelligent and autonomous phone automation solutions. Seed1.5-VL Guo & et al. (2025) is developed through extensive training on a massive 3 trillion token dataset to build a more general-purpose and capable vision-language model. This dataset is curated using diversified data synthesis pipelines targeting key capabilities like OCR, visual grounding, and GUI interaction. The model's capabilities are further enhanced through a comprehensive post-training phase that includes both SFT for instruction following and advanced reinforcement learning techniques, including Reinforcement Learning from Human Feedback (RLHF) and Reinforcement Learning with Verifiable Rewards. MiMo-VL Team (2025) demonstrates how a compact VLM can achieve state-of-the-art performance on GUI tasks through a sophisticated training pipeline. The process involves a multi-stage pre-training phase that incorporates diverse data types, including specific GUI interaction data, and a novel post-training phase using Mixed On-policy Reinforcement Learning (MORL). This MORL framework uniquely integrates verifiable rewards for tasks like GUI grounding with RLHF. This hybrid approach enables MiMo-VL, a general-purpose model, to surpass even specialized GUI agents

on challenging grounding and understanding benchmarks, highlighting the power of combining extensive pre-training with advanced, task-aware reinforcement learning. UI-Venus Gu et al. (2025) utilizes Group Relative Policy Optimization (GRPO) to build high-performance phone agents. It introduces a Self-Evolving Trajectory History Alignment & Sparse Action Enhancement framework to improve planning capabilities and better handle rare but pivotal actions. The work also places a strong emphasis on data quality, implementing a rigorous multi-stage pipeline to filter and reconstruct training data. By training models at both 7B and 72B scales, UI-Venus demonstrates state-of-the-art performance on both UI grounding and navigation benchmarks, showcasing the effectiveness of scaling up Reinforcement Fine-Tuning(RFT)-based methods for complex UI tasks. To address challenges in training with long and variable action sequences, GUI-Owl Ye et al. (2025a) introduces a scalable reinforcement learning framework. This framework features Trajectory-aware Relative Policy Optimization, which utilizes trajectory-level rewards to compute step-level advantages and employs a replay buffer to enhance training stability, better aligning the agent's policy with real-world task success.

**Phone UI-Specific Models.** Phone UI-Specific Model Architectures have primarily focused on *screen understanding tasks*, which are essential for enabling AI agents to interact effectively with graphical user interfaces. These tasks can be categorized into three main types: *UI grounding*, *UI referring*, and *screen question answering (QA)*. Figure 10 illustrates the differences between these categories.

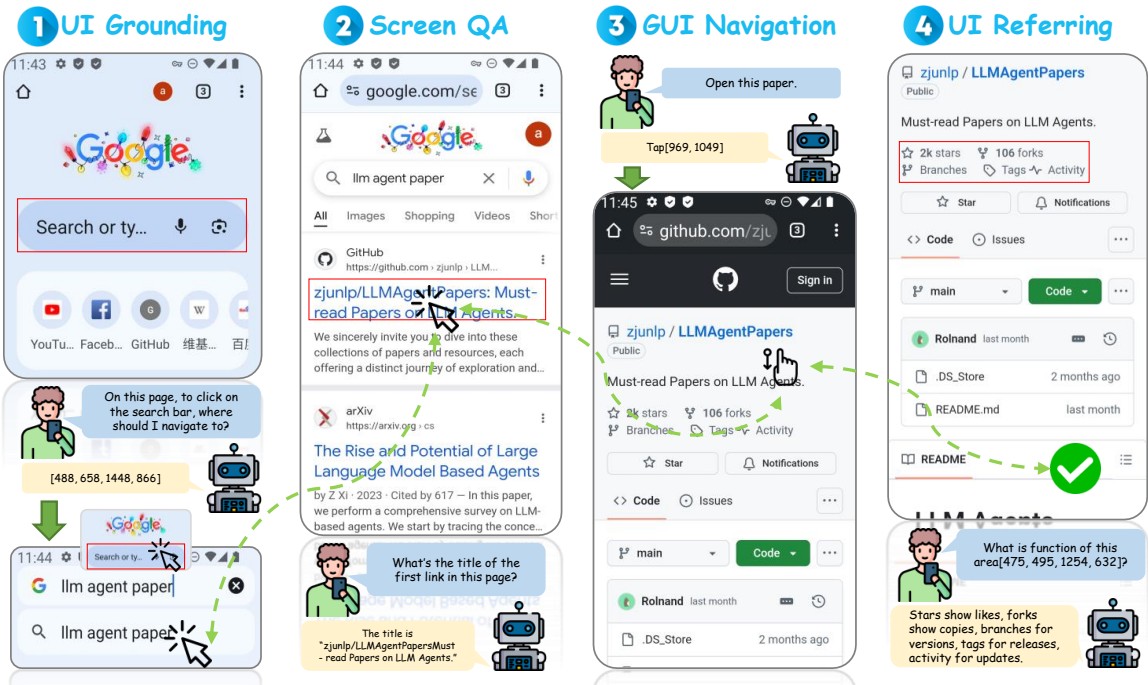

Figure 10: Illustration of screen understanding tasks. (a) *UI Grounding* involves identifying UI elements corresponding to a given description; (b) *UI Referring* focuses on generating descriptions for specified UI elements; (c) *Screen Question Answering* requires answering questions based on the content of the screen.

- **UI Grounding** involves identifying and localizing UI elements on a screen that correspond to a given natural language description. This task is critical for agents to perform precise interactions with GUIs based on user instructions. MUG Li et al. (2022) proposes guiding agent actions through multi-round interactions with users, improving the execution accuracy of UI grounding in complex or ambiguous instruction scenarios. It also leverages user instructions and previous interaction history to predict the next agent action. LVG (Layout-guided Visual Grounding) Qian et al. (2024b) addresses UI grounding by unifying detection and grounding of UI elements within application interfaces. LVG tackles challenges such as *application sensitivity*, where UI elements with similar appearances have different functions across applications, and *context sensitivity*, where the functionality of UI elements depends on their context within the interface. By introducing layout-guided contrastive learning, LVG

learns the semantics of UI objects from their visual organization and spatial relationships, improving grounding accuracy. UI-Hawk Zhang et al. (2024d) enhances UI grounding by incorporating a history-aware visual encoder and an efficient resampler to process screen sequences during GUI navigation. By understanding historical screens, UI-Hawk improves the agent's ability to ground UI elements accurately over time. An automated data curation method generates training data for UI grounding, contributing to the creation of the FunUI benchmark for evaluating screen understanding capabilities. Aria-UI Yang et al. (2024b) leverages strong MLLMs such as GPT-4o to generate diverse and high-quality element instructions for grounding training. It employs a two-stage training method that incorporates action history in textual or interleaved text-image formats, enabling the model to develop both single-step localization capabilities and multi-step context awareness. This approach demonstrates robust performance and generalization ability across various tasks. Similar research includes GUI-Bee Fan et al. (2025), which autonomously explores environments to collect high-quality data, thereby aligning GUI action grounding models with new environments and significantly enhancing model performance. OS-Atlas Wu et al. (2024d) unifies the action space, enabling models to adapt to UI grounding tasks across multiple platforms. Additionally, TAG (Tuning-free Attention-driven Grounding) Xu et al. (2024a) introduces a method that leverages the inherent attention mechanisms of pre-trained MLLMs to accurately identify and locate elements within a GUI without the need for tuning. Validation shows that this method performs comparably to or even surpasses tuned approaches across multiple benchmark datasets, demonstrating exceptional generalization capabilities. This offers a new perspective for the application of MLLMs in UI grounding.

- **UI Referring** focuses on generating natural language descriptions for specified UI elements on a screen. This task enables agents to explain UI components to users or other agents, facilitating better communication and interaction. Ferret-UI You et al. (2024) is a multimodal LLM designed for enhanced understanding of mobile UI screens, emphasizing precise referring and grounding tasks. By incorporating *any resolution* techniques to handle various screen aspect ratios and dividing screens into sub-images for detailed analysis, Ferret-UI generates accurate descriptions of UI elements. Training on a curated dataset of elementary UI tasks, Ferret-UI demonstrates strong performance in UI referring tasks. Leveraging the Ferret-UI framework, Ferret-UI 2 Li et al. (2024e) integrates an adaptive N-grid partitioning mechanism. This system enhances image feature extraction by dynamically resizing grids, thereby improving the model's efficiency and accuracy without sacrificing resolution. Additionally, Ferret-UI 2 demonstrates remarkable cross-platform portability. Textual Foresight Burns et al. (2024) uses user actions as a bridge, requiring the model to predict the global textual description of the next UI state based on the current UI screen and a local action. With limited training data, the Textual Foresight method achieves superior performance compared to similar models, demonstrating exceptional data efficiency. UI-Hawk Zhang et al. (2024d) also contributes to UI referring by defining tasks that require the agent to generate descriptions for UI elements based on their role and context within the interface. By processing screen sequences and understanding the temporal relationships between screens, UI-Hawk improves the agent's ability to refer to UI elements accurately.

- **Screen Question Answering** involves answering questions about the content and functionality of a screen based on visual and textual information. This task requires agents to comprehend complex screen layouts and extract relevant information to provide accurate answers. ScreenAI Baechler et al. (2024) specializes in understanding screen UIs and infographics, leveraging the common visual language and design principles shared between them. By introducing a flexible patching strategy and a novel textual representation for UIs, ScreenAI pre-trains models to interpret UI elements effectively. Using large language models to automatically generate training data, ScreenAI covers tasks such as screen annotation and screen QA. WebVLN Chen et al. (2024c) extends vision-and-language navigation to websites, where agents navigate based on question-based instructions and answer questions using information extracted from target web pages. By integrating visual inputs, linguistic instructions, and web-specific content like HTML, WebVLN enables agents to understand both the visual layout and underlying structure of web pages, enhancing screen QA capabilities. UI-Hawk Zhang et al. (2024d) further enhances screen QA by enabling agents to process screen sequences and answer questions based on historical interactions. By incorporating screen question

answering as one of its fundamental tasks, UI-Hawk improves the agent's ability to comprehend and reason about screen content over time. MP-GUI Wang et al. (2025e) introduces a specialized MLLM for GUI understanding with three dedicated perceivers for graphical, textual, and spatial modalities. Using a fusion gate to adaptively combine these modalities and an automated data collection pipeline to address training data scarcity, MP-GUI achieves strong performance on GUI understanding tasks including screen QA despite limited training data.

These Phone UI-Specific Model Model Architectures demonstrate the importance of focusing on screen understanding tasks to enhance AI agents' interaction with complex user interfaces. By categorizing these tasks into UI grounding, UI referring, and screen question answering, researchers have developed specialized models that address the unique challenges within each category. Integrating innovative techniques such as layout-guided contrastive learning, history-aware visual encoding, and flexible patching strategies has led to significant advancements in agents' abilities to understand, navigate, and interact with GUIs effectively.

### 5.2.2 Supervised Fine-Tuning

Supervised fine-tuning has emerged as a crucial technique for enhancing the capabilities of LLMs in GUI tasks within phone automation. By tailoring models to specific tasks through fine-tuning on curated datasets, researchers have significantly improved models' abilities in GUI grounding, optical character recognition (OCR), cross-application navigation, and efficiency. A summary of notable works in this area is presented in Table 6, highlighting their main contributions, domains, and other relevant details.

Table 6: Summary of supervised fine-tuning methods for phone GUI agents

| Method | Date | Task Type | Backbone | Size | Contributions |
|---|---|---|---|---|---|
| **MobileAgent** Ding (2024) ○ | 2024.01 | General | Qwen | 7B | Standard Operating Procedure; Human-machine interaction |
| **SeeClick** Cheng et al. (2024) ○ | 2024.01 | General | Qwen-VL | 9.6B | GUI grounding pre-training; ScreenSpot benchmark |
| **ReALM** Moniz et al. (2024) | 2024.04 | Reference Resolution | FLAN-T5 | 80M–3B | Formulated reference resolution as language modeling; Improved performance on resolving references |
| **GUICourse** Chen et al. (2024d) ○ | 2024.06 | General | Qwen-VL, Fuyu-8B, MiniCPM-V | N/A | Suite of datasets (GUIEnv, GUIAct, GUIChat); Enhanced OCR and grounding |
| **GUI Odyssey** Lu et al. (2024a) ○ | 2024.06 | General | Qwen-VL | N/A | Cross-app navigation dataset; Agent with history resampling |
| **IconDesc** Haque & Csallner (2024) | 2024.09 | Alt-Text Generation | GPT-3.5 | N/A | Generated alt-text for UI icons using partial UI data; Improved accessibility |
| **TinyClick** Pawlowski et al. (2024) ○ | 2024.10 | General | Florence-2 | 0.27B | Single-turn agent; Multitask training; MLLM-based data augmentation |
| **InfiGUIAgent** Liu et al. (2025c) ○ | 2025.01 | General | Qwen2-VL | 2B | Model-Environment alignment; Self-exploratory Data |
| **Agent-R** Yuan et al. (2025) ○ | 2025.01 | General | LLama-3.1 | 8B | Self-reflection capabilities; Real-time error correction |
| **Chain-of-Memory** Gao et al. (2025) | 2025.06 | General | Qwen2-VL | 7B | Recognize and memorize tasks in a human-like manner |
| **ZonUI-3B** Hsieh et al. (2025) ○ | 2025.07 | General | Qwen2.5-VL | 3B | Multi-resolution training corpus |

Supervised fine-tuning has been effectively applied to develop more versatile and efficient GUI agents by enhancing their fundamental abilities and GUI knowledge. One of the fundamental challenges in developing visual GUI agents is enabling accurate interaction with screen elements based solely on visual inputs, known as GUI grounding. SeeClick Cheng et al. (2024) addresses this challenge by introducing a visual GUI agent

that relies exclusively on screenshots for task automation, circumventing the need for extracted structured data like HTML, which can be lengthy and sometimes inaccessible. Recognizing that GUI grounding is a key hurdle, SeeClick enhances the agent's capability by incorporating GUI grounding pre-training. The authors also introduce ScreenSpot, the first realistic GUI grounding benchmark encompassing mobile, desktop, and web environment. Experimental results demonstrate that improving GUI grounding through supervised fine-tuning directly correlates with enhanced performance in downstream GUI tasks. InfiGUIAgent Liu et al. (2025c) is trained using a supervised fine-tuning method and employs the Reference-Augmented Annotation approach to fully leverage spatial information, establishing bidirectional connections between GUI elements and text descriptions, thereby enhancing the model's understanding of GUI visual language. Additionally, the model incorporates Hierarchical Reasoning and Expectation-Reflection Reasoning capabilities, enabling the agent to perform complex reasoning natively, which improves its grounding ability. Beyond grounding, agents require robust OCR capabilities and comprehensive knowledge of GUI components and interactions to function effectively across diverse applications. GUICourse Chen et al. (2024d) tackles these challenges by presenting a suite of datasets designed to train visual-based GUI agents from general VLMs. The GUIEnv dataset strengthens OCR and grounding abilities by providing 10 million website page-annotation pairs for pre-training and 0.7 million region-text QA pairs for supervised fine-tuning. To enrich the agent's understanding of GUI components and interactions, the GUIAct and GUIChat datasets offer extensive single-step and multi-step action instructions and conversational data with text-rich images and bounding boxes. As users frequently navigate across multiple applications to complete complex tasks, enabling cross-app GUI navigation becomes essential for practical GUI agents.GUI Odyssey Lu et al. (2024a) addresses this need by introducing a comprehensive dataset specifically designed for training and evaluating cross-app navigation agents. The GUI Odyssey dataset comprises 7,735 episodes from six mobile devices, covering six types of cross-app tasks, 201 apps, and 1,399 app combinations. By fine-tuning the Qwen-VL model with a history resampling module on this dataset, they developed OdysseyAgent, a multimodal cross-app navigation agent. Extensive experiments show that OdysseyAgent achieves superior accuracy compared to existing models, significantly improving both in-domain and out-of-domain performance on cross-app navigation tasks. Efficiency and scalability are also critical considerations, especially for deploying GUI agents on devices with limited computational resources. TinyClick Pawlowski et al. (2024) demonstrates that even compact models can achieve strong performance on GUI automation tasks through effective supervised fine-tuning strategies. Utilizing the Vision-Language Model Florence-2-Base, TinyClick focuses on the primary task of identifying the screen coordinates of UI elements corresponding to user commands. By employing multi-task training and Multimodal Large Language Model-based data augmentation, TinyClick significantly improves model performance while maintaining a compact size of 0.27 billion parameters and minimal latency. MobileAgent Ding (2024) combines LoRA and SOP methods to effectively reduce computational overhead through low-rank adaptive supervised fine-tuning, while breaking down complex tasks into subtasks to enhance the model's understanding and execution efficiency. At the same time, this approach does not impose additional burdens on inference speed, significantly improving the model's performance and responsiveness. The performance of agents is often limited by their inability to recover from errors. Agent-R Yuan et al. (2025) identifies the first error step in an erroneous trajectory and combines it with a correct trajectory to create a corrected path, thus enabling real-time error correction. By training on self-generated corrected trajectories and using an iterative supervised fine-tuning approach, Agent-R dynamically identifies and rectifies errors, gradually enhancing decision-making abilities. Moreover, under a multi-task training strategy, its training outcomes improve significantly. This method offers new directions for developing more intelligent and adaptable GUI agents.

Supervised fine-tuning has also been applied to domain-specific tasks to address specialized challenges in particular contexts, such as reference resolution and accessibility. In the context of **Reference Resolution in GUI Contexts**, ReALM Moniz et al. (2024) formulates reference resolution as a language modeling problem, enabling the model to handle various types of references, including on-screen entities, conversational entities, and background entities. By converting reference resolution into a multiple-choice task for the LLM, ReALM significantly improves the model's ability to resolve references in GUI contexts. For **Accessibility and UI Icons Alt-Text Generation**, IconDesc Haque & Csallner (2024) addresses the challenge of generating informative alt-text for mobile UI icons, which is essential for users relying on screen readers. Traditional deep learning approaches require extensive datasets and struggle with the diversity and imbalance of icon types.

IconDesc introduces a novel method using Large Language Models to autonomously generate alt-text with partial UI data, such as class, resource ID, bounds, and contextual information from parent and sibling nodes. By fine-tuning an off-the-shelf LLM on a small dataset of approximately 1.4k icons, IconDesc demonstrates significant improvements in generating relevant alt-text, aiding developers in enhancing UI accessibility during app development. ZonUI-3B Hsieh et al. (2025) demonstrates how a compact 3B model can achieve competitive GUI grounding performance, rivaling much larger models. It introduces a novel two-stage supervised fine-tuning approach, which begins with platform-general pretraining on a diverse corpus and is followed by a resolution-focused specialization stage. This methodology, coupled with efficient data sampling, proves that sophisticated fine-tuning can enable high-performance agents on consumer-grade hardware, addressing the challenge of high computational costs. To address information loss during long-horizon, cross-application tasks, Chain-of-Memory (CoM) Gao et al. (2025) introduces a memory mechanism inspired by human cognition. This paradigm uses Short-Term Memory to track recent task context and Long-Term Memory to retain critical information across applications. The work also presents the GUI Odyssey-CoM dataset, an annotated collection of cross-app trajectories, to fine-tune smaller models, thereby enabling them with sophisticated memory management and reasoning capabilities for complex workflows.

These works collectively demonstrate that supervised fine-tuning is instrumental in advancing GUI agents for phone automation. By addressing specific challenges through targeted datasets and training strategies—whether enhancing GUI grounding, improving OCR and GUI knowledge, enabling cross-app navigation, or optimizing for accessibility—researchers have significantly enhanced the performance and applicability of GUI agents. The advancements summarized in Figure 6 highlight the ongoing efforts and progress in this field, paving the way for more intelligent, versatile, and accessible phone automation solutions capable of handling complex tasks in diverse environment.

### 5.2.3 Reinforcement Learning

Reinforcement Learning (RL) Kaelbling et al. (1996) has emerged as a powerful technique for training agents to interact autonomously with GUIs across various platforms, including phones, web browsers, and desktop environment. Although RL-based approaches for phone GUI agents are relatively few, significant progress has been made in leveraging RL to enhance agent capabilities in dynamic and complex GUI environment. In this section, we discuss RL approaches for GUI agents across different platforms, highlighting their unique challenges, methodologies, and contributions. A summary of notable RL-based methods is presented in Figure 7, which includes specific RL-related features such as the type of RL used (online or offline) and the targeted platform.

**Phone Agents.** Training phone GUI agents using RL presents unique challenges due to the dynamic and complex nature of mobile applications. Agents must adapt to real-world stochasticity and handle the intricacies of interacting with diverse mobile environment. Recent works have addressed these challenges by developing RL frameworks that enable agents to learn from interactions and improve over time. DigiRL Bai et al. (2024) andDistRL Wang et al. (2024h) both tackle the limitations of pre-trained vision-language models (VLMs) in decision-making tasks for device control through GUIs. Recognizing that static demonstrations are insufficient due to the dynamic nature of real-world mobile environment, these works introduce RL approaches to train agents capable of in-the-wild device control. DigiRL proposes an autonomous RL framework that employs a two-stage training process: an initial offline RL phase to initialize the agent using existing data, followed by an offline-to-online RL phase that fine-tunes the model based on its own interactions. By building a scalable Android learning environment with a VLM-based evaluator, DigiRL identifies key design choices for effective RL in mobile GUI domains. The agent learns to handle real-world stochasticity and dynamism, achieving significant improvements over supervised fine-tuning, with a 49.5% absolute increase in success rate on the Android-in-the-Wild dataset. Similarly, DistRL introduces an asynchronous distributed RL framework specifically designed for on-device control agents on mobile devices. To address inefficiencies in online fine-tuning and the challenges posed by dynamic mobile environment, DistRL employs centralized training and decentralized data acquisition. Leveraging an off-policy RL algorithm tailored for distributed and asynchronous data utilization, DistRL improves training efficiency and agent performance by prioritizing significant experiences and encouraging exploration. Experiments show that DistRL achieves a 20% relative improvement in success rate compared to state-of-the-art methods on general Android tasks. Building upon

Table 7: Summary of reinforcement learning methods for phone GUI agents

| Method | Date | Platform | RL Type | Backbone | Size |
|---|---|---|---|---|---|
| **DigiRL** Bai et al. (2024) ⏻ | 2024.06 | Phone | Online RL | AutoUI-Base | 200M |
| **DistRL** Wang et al. (2024h) ⏻ | 2024.10 | Phone | Online RL | T5-based | 1.3B |
| **AutoGLM** Liu et al. (2024b) ⏻ | 2024.11 | Phone, Web | Online RL | GLM-4-9B-Base | 9B |
| **ScreenAgent** Niu et al. (2024) ⏻ | 2024.02 | PC OS | N/A | CogAgent | 18B |
| **ETO** Song et al. (2024a) ⏻ | 2024.03 | Web | Offline-to-Online RL | LLaMA-2-7B-Chat | 7B |
| **AutoWebGLM** Lai et al. (2024) ⏻ | 2024.04 | Web | RL (Curriculum Learning, Boot-strapped RL) | ChatGLM3-6B | 6B |
| **Agent Q** Putta et al. (2024) ⏻ | 2024.08 | Web | Offline RL with MCTS | LLaMA-3-70B | 70B |
| **GLAINTEL** Fereidouni et al. (2024) ⏻ | 2024.11 | Web | RL (Offline-to-Online, Hybrid RL) | Flan-T5 | 0.78B |
| **ReachAgent** Wu et al. (2025d) | 2025.02 | Phone | Hybrid RL | MobileVLM Wu et al. (2024c) | N/A |
| **VEM** Zheng et al. (2025) ⏻ | 2025.02 | Phone | Environment-Free RL | N/A | N/A |
| **Digi-Q** Bai et al. (2025) ⏻ | 2025.02 | Phone | Q-Function Based RL | N/A | N/A |
| **VSC-RL** Wu et al. (2025c) ⏻ | 2025.02 | Phone | Variational Subgoal-Conditioned RL | N/A | N/A |
| **UI-R1** Lu et al. (2025b) ⏻ | 2025.03 | Phone | Rule-Based RL | Qwen2.5-VL | 3B |
| **GUI-G1** Zhou et al. (2025a) ⏻ | 2025.05 | Phone | Rule-Based RL | Qwen2.5-VL | 3B |
| **ZeroGUI** Yang et al. (2025) ⏻ | 2025.05 | Phone | Rule-Based RL | UI-TARS-7B | 7B |
| **AgentCPM-GUI** Zhang et al. (2025) ⏻ | 2025.06 | Phone | Rule-Based RL | MiniCPM-V | 7B |
| **GUI-Reflection** Wu et al. (2025b) ⏻ | 2025.06 | Phone | Rule-Based RL | InternVL3 | 8B |
| **MobileGUI-RL** Shi et al. (2025) | 2025.07 | Phone | Rule-Based RL | Qwen2.5-VL | 7B/32B |
| **MagicGUI** Tang et al. (2025) | 2025.08 | Phone | Rule-Based RL | N/A | N/A |

these advancements, AutoGLM Liu et al. (2024b) extends the application of RL to both phone and web platforms. AutoGLM presents a series of foundation agents based on the ChatGLM model family, aiming to serve as autonomous agents for GUI control. A key insight from this work is the design of an intermediate interface that separates planning and grounding behaviors, allowing for more agile development and enhanced performance. By employing self-evolving online curriculum RL, AutoGLM enables agents to learn from environmental interactions and adapt to dynamic GUI environment. The approach demonstrates impressive success rates on various benchmarks, showcasing the potential of RL in creating versatile GUI agents across platforms.

Recent advances have brought several innovative approaches to reinforcement learning for phone GUI agents. ReachAgent Wu et al. (2025d) decomposes mobile agent tasks into two sub-tasks: page reaching and page operation, utilizing a two-stage fine-tuning strategy. In the first stage, supervised fine-tuning enables the agent to better perform each sub-task. In the second stage, reinforcement learning is applied to further optimize the agent's overall task completion capabilities, thereby enhancing its performance in complex tasks. VEM Zheng et al. (2025) introduces an environment-free RL framework that decouples value estimation from policy optimization using a pretrained Value Environment Model. Unlike traditional RL methods that require costly environment interactions, VEM predicts state-action values directly from offline data, distilling human-like priors about GUI interaction outcomes. This approach avoids compounding errors and enhances resilience to UI changes by focusing on semantic reasoning. Digi-Q Bai et al. (2025) presents an approach to train VLM-based action-value Q-functions for device control. Instead of using on-policy RL with actual environment rollouts, Digi-Q trains the Q-function using offline temporal-difference learning on frozen, intermediate-layer features of a VLM. This approach enhances scalability and reduces computational costs compared to fine-tuning the entire VLM. The trained Q-function then uses a Best-of-N policy extraction operator to imitate the best action without requiring environment interaction. VSC-RL Wu et al. (2025c) addresses the learning inefficiencies in tackling complex sequential decision-making tasks with sparse rewards and long-horizon dependencies. By reformulating vision-language sequential tasks as a variational goal-conditioned RL problem,

VSC-RL optimizes the SubGoal Evidence Lower BOund (SGC-ELBO). This approach maximizes subgoal-conditioned return via RL while minimizing the difference with the reference policy. UI-R1 Lu et al. (2025b) explores how rule-based RL can enhance reasoning capabilities of multimodal large language models for GUI action prediction. Using a small yet high-quality dataset of 136 challenging tasks, UI-R1 introduces a unified rule-based action reward enabling model optimization via GRPO. GUI-G1 Zhou et al. (2025a) provides a critical analysis of the R1-like training paradigm specifically for GUI grounding tasks. The work identifies key challenges that arise from blindly applying general-purpose RL methods, including performance degradation from excessive chain-of-thought reasoning, reward hacking from standard hit-based and IoU-based reward functions, and learning biases within the GRPO algorithm. To address these issues, GUI-G1 proposes a suite of targeted solutions: a "Fast Thinking Template" to encourage direct action generation, a box-size constraint in the reward function to mitigate reward hacking, and a revised GRPO objective that accounts for sample difficulty and removes length bias. This detailed investigation demonstrates that a carefully tailored RL approach, rather than a generic application, is crucial for advancing grounding performance in GUI agents. ZeroGUI Yang et al. (2025) introduces a fully automated online RL framework designed to operate at zero human cost. The framework leverages a VLM for both automatic task generation and reward estimation, completely eliminating the need for human-annotated data or scripted verifiers. ZeroGUI employs a two-stage online RL process, first training on the generated tasks and then performing test-time adaptation, which enables the agent to continuously improve its policy through environmental interaction without any manual supervision. AgentCPM-GUI Zhang et al. (2025) addresses data quality and reasoning generalization challenges, particularly for Chinese GUIs. It introduces a three-stage progressive training pipeline that culminates in RFT. Using GRPO, the RFT stage enhances the agent's reasoning and planning abilities beyond what is achieved through supervised imitation learning alone, demonstrating strong cross-lingual and cross-app generalization. GUI-Reflection Wu et al. (2025b) introduces a comprehensive framework to explicitly instill self-reflection and error-correction capabilities into end-to-end models. The framework employs a multi-stage process that begins with a novel pre-training task suite to cultivate foundational reflection skills, followed by an offline SFT phase that injects automatically generated error scenarios into the training data. Crucially, it culminates in an online learning stage featuring an "iterative online reflection tuning" algorithm. This algorithm enables the agent to interact with the environment, identify its own errors, and learn from them through automatically generated corrective supervision, bridging the gap between mimicking expert demonstrations and developing robust agents that can adaptively recover from real-world failures. MobileGUI-RL Shi et al. (2025) further advances online RL for phone agents by introducing a framework that combines a self-exploratory task generation pipeline with a text-based world model for filtering. It proposes MobGRPO, an adaptation of the GRPO algorithm, which uses a trajectory-aware advantage and a multi-component reward to effectively train agents in a scalable, interactive online environment. MagicGUI Tang et al. (2025) introduces a foundational mobile agent built upon a comprehensive two-stage training procedure. The process begins with Continue Pre-training on a large-scale, high-quality dataset generated by a scalable data pipeline. Subsequently, it employs RFT using a novel Dual Filtering Group Relative Policy Optimization algorithm. This RL stage leverages a spatially enhanced composite reward function to further improve the agent's robustness and decision-making capabilities in diverse and dynamic GUI environments.

**Web Agents.** The web platform has served as a primary testbed for pioneering RL techniques in GUI automation, largely due to its structured nature and the relative ease of data collection. Works like ETO Song et al. (2024a) and Agent Q Putta et al. (2024) were among the first to successfully apply advanced RL methods, such as learning from failure trajectories and offline-to-online fine-tuning, to complex interactive tasks. These techniques, validated on the web, are now being adapted to address the unique challenges of mobile environments, including sparse rewards and high dynamism. Web navigation tasks involve interacting with complex and dynamic web environment, where agents must interpret web content and perform actions to achieve user-specified goals. RL has been employed to train agents that can adapt to these challenges by learning from interactions and improving decision-making capabilities. ETO Song et al. (2024a) (Exploration-based Trajectory Optimization) and Agent Q Putta et al. (2024) both focus on enhancing the performance of LLM-based agents in web environment through RL techniques. ETO introduces a learning method that allows agents to learn from their exploration failures by iteratively collecting failure trajectories and using them to create contrastive trajectory pairs for training. By leveraging contrastive learning methods like Direct Preference Optimization (DPO), ETO enables agents to improve performance through an iterative

cycle of exploration and training. Experiments on tasks such as WebShop demonstrate that ETO consistently outperforms baselines, highlighting the effectiveness of learning from failures. Agent Q combines guided Monte Carlo Tree Search (MCTS) with a self-critique mechanism and iterative fine-tuning using an off-policy variant of DPO. This framework allows LLM agents to learn from both successful and unsuccessful trajectories, improving generalization in complex, multi-step reasoning tasks. Evaluations on the WebShop environment and real-world booking scenarios show that Agent Q significantly improves success rates, outperforming behavior cloning and reinforcement learning fine-tuned baselines. AutoWebGLM Lai et al. (2024) contributes to this domain by developing an LLM-based web navigating agent built upon ChatGLM3-6B. To address the complexity of HTML data and the versatility of web actions, AutoWebGLM introduces an HTML simplification algorithm to represent webpages succinctly. The agent is trained using a hybrid human-AI method to build web browsing data for curriculum training and is further enhanced through reinforcement learning and rejection sampling. AutoWebGLM demonstrates performance superiority on general webpage browsing tasks, achieving practical usability in real-world services. GLAINTEL Fereidouni et al. (2024) effectively utilizes human experience and the adaptive capabilities of reinforcement learning by integrating human demonstrations with reinforcement learning methods. This approach achieves superior performance in complex product search tasks. Collectively, these works demonstrate how RL techniques can be applied to web agents to improve their ability to navigate and interact with complex web environment. By learning from interactions, failures, and leveraging advanced planning methods, these agents exhibit enhanced reasoning and decision-making capabilities.

**PC OS Agents.** PC OS agents were instrumental in demonstrating the feasibility of controlling an entire operating system via raw pixel inputs and low-level keyboard/mouse actions. This pioneering work validates the core vision-based control loop that is fundamental to creating general-purpose mobile agents capable of navigating not just within apps, but the mobile OS itself. ScreenAgent Niu et al. (2024) introduces a vision-based agent that interacts with GUIs using only screenshots and high-level natural language instructions, successfully operating across different operating systems including Windows, Ubuntu, and macOS. By employing a scalable pre-training and fine-tuning pipeline, ScreenAgent learns from a vast dataset of interaction traces, enabling it to generalize across various applications and platforms.

## 5.3 Comparative Analysis of Modeling Approaches

The choice between Prompt Engineering and Training-Based Methods for developing mobile agents is not merely a technical decision but a strategic one, involving fundamental trade-offs between adaptability, performance, and resource investment. Rather than viewing them as opposing methodologies, it is more productive to see them as different stages or philosophies in agent development. A detailed comparison across key dimensions is presented in Table 8.

**Prompt Engineering: The Path of Generality and Low Upfront Cost.** Prompt engineering serves as a powerful and resource-efficient entry point for creating mobile agents. As exemplified by zero-shot agents like *AppAgent* Zhang et al. (2023a), this approach leverages the vast, pre-existing knowledge of massive models like GPT-4V. Its primary strength lies in its flexibility; it requires no task-specific training data or costly fine-tuning, making it ideal for rapid prototyping and for building agents that can generalize across a wide array of unseen applications. However, this "low cost" refers primarily to the development phase; the operational cost of repeatedly calling powerful proprietary APIs for inference can become substantial, especially for complex, multi-step tasks. Furthermore, this approach has a clear performance ceiling. Its effectiveness is fundamentally limited by the inherent capabilities of the general-purpose foundation model and the ingenuity of the prompt design. The model's reasoning is not deeply adapted to the nuances of GUI interaction, making it more prone to errors on complex or unconventional interfaces.

**Training-Based Methods: The Pursuit of Specialization and Peak Performance.** In contrast, training-based methods aim to achieve state-of-the-art performance by deeply adapting a model to the specific domain of mobile GUI automation.

- **Supervised Fine-Tuning (SFT)**, as used in models like *SeeClick* Cheng et al. (2024) and *Aguvis* Xu et al. (2024c), allows an agent to internalize the specific patterns, layouts, and interaction logic of mobile UIs from large datasets. This process embeds specialized knowledge directly into the model's weights,

Table 8: Comparative analysis of modeling approaches for mobile agents.

| Dimension | Prompt Engineering (Zero-Shot/Few-Shot) | Training-Based Methods (SFT & RL) |
|---|---|---|
| **Core Principle** | Leverages a powerful pre-trained model's existing, general-purpose knowledge via meticulously crafted prompts. | Adapts a model's internal parameters to domain-specific data and interactive experience, embedding specialized knowledge. |
| **Data Requirement** | Minimal to none. Relies on examples within the prompt's context window. | <ul><li>**SFT:** Requires large, high-quality labeled datasets.</li><li>**RL:** Requires extensive and costly environmental interaction.</li></ul> |
| **Performance & Ceiling** | Performance is capped by the base model's capabilities and prompt quality. Excellent for establishing a strong baseline. | Can achieve state-of-the-art performance by specializing the model. Possesses a much higher performance ceiling on in-domain tasks. |
| **Development Cost** | Low upfront cost (no training data/computation). Can incur high operational costs from API calls during inference. | High cost for data collection/annotation and computationally intensive training, as demonstrated by works like *SeeClick* Cheng et al. (2024). |
| **Deployment & Efficiency** | Relies on large, cloud-based models; high latency and requires network. Unsuitable for on-device deployment. | Enables creation of smaller, specialized models (e.g., *TinyClick*) suitable for efficient on-device deployment, offering low latency and offline capability. |

enabling it to achieve a higher level of accuracy and reliability on in-domain tasks than is typically possible with prompting alone. The primary drawback is the immense cost associated with creating and annotating large-scale, high-quality datasets, and the risk of the model overfitting to the training data, thereby limiting its adaptability to new applications.

- **Reinforcement Learning (RL)** represents a further step towards creating truly autonomous and adaptive agents. Unlike SFT, which learns from static datasets, RL agents like *DigiRL* Bai et al. (2024) learn through dynamic trial-and-error interaction with the environment. This allows them to develop more robust and sophisticated decision-making policies, especially for tasks with long horizons or complex state dependencies. However, RL introduces its own formidable challenges, including sample inefficiency (requiring millions of interactions), complex reward function design, and the need for stable, scalable, and often parallelized training environments.

**The Path to Lightweight and Efficient On-Device Deployment.** A critical dimension in this trade-off space is the path to practical, on-device deployment. Prompt engineering, which relies on massive, cloud-hosted models, is inherently unsuitable for this goal due to high latency and network dependency. In contrast, training-based methods are essential for creating user-facing agents that are responsive and preserve privacy. This approach allows for the development of smaller, specialized models through techniques like model distillation, quantization, and fine-tuning on targeted data. Works like *TinyClick* Pawlowski et al. (2024) and *Octopus v2* Chen & Li (2024) demonstrate that compact models can be trained to achieve high performance with significantly lower latency, enabling the offline capabilities and responsiveness required for real-world on-device applications.

**Synthesis and Future Outlook.** The distinction between these approaches is blurring, with a clear trend towards hybrid methodologies that harness the strengths of both. A powerful emerging paradigm involves a multi-stage process: begin with a large, general-purpose foundation model (the product of massive pre-training), use SFT on diverse GUI datasets to create a highly capable base agent, and finally, employ RL for further refinement to enhance its decision-making and robustness in specific domains. This synergistic

approach suggests that prompt engineering, SFT, and RL should not be seen as competing alternatives, but as complementary tools in the comprehensive lifecycle of developing next-generation mobile agents.

# 6    Datasets and Benchmarks

The rapid evolution of mobile technology has transformed smartphones into indispensable tools for communication, productivity, and entertainment. This shift has spurred a growing interest in developing intelligent agents capable of automating tasks and enhancing user interactions with mobile devices. These agents rely on a deep understanding of GUIs and the ability to interpret and execute instructions effectively. However, the development of such agents presents significant challenges, including the need for diverse datasets, standardized benchmarks, and robust evaluation methodologies.

Datasets serve as the backbone for training and testing phone GUI agents, offering rich annotations and task diversity to enable these agents to learn and adapt to complex environment. Complementing these datasets, benchmarks provide structured environment and evaluation metrics, allowing researchers to assess agent performance in a consistent and reproducible manner. Together, datasets and benchmarks form the foundation for advancing the capabilities of GUI-based agents.

This section delves into the **key datasets** and **benchmarks** that have shaped the field. Subsection 6.1 reviews notable datasets that provide the training data necessary for enabling agents to perform tasks such as language grounding, UI navigation, and multimodal interaction. Subsection 6.2 discusses benchmarks that facilitate the evaluation of agent performance, focusing on their contributions to reproducibility, generalization, and scalability. Through these resources, researchers and developers gain the tools needed to push the boundaries of intelligent phone automation, moving closer to creating agents that can seamlessly assist users in their daily lives.

Table 9: Summary of datasets for phone GUI agents. "Actions" refers to the number of distinct actions available; "Demos" refers to the number of demonstration sequences; "Apps" refers to the number of applications covered; "Instr." refers to the number of natural language instructions; "Avg. Steps" refers to the average number of steps per task.

| Dataset | Date | Screenshots | UI Trees | Actions | Demos | Apps | Instr. | Avg. Steps | Contributions |
|---|---|---|---|---|---|---|---|---|---|
| **Rico** Deka et al. (2017) ⟳ | 2017.10 | ✔ | ✔ | N/A | 10,811 | 9,772 | N/A | N/A | Large-scale mobile dataset |
| **PixelHelp** Li et al. (2020) ⟳ | 2020.05 | ✔ | ✔ | 4 | 187 | 4 | 187 | 4.2 | Grounding instructions to actions |
| **MoTIF** Burns et al. (2021) ⟳ | 2021.04 | ✔ | ✔ | 6 | 4,707 | 125 | 276 | 4.5 | Interactive visual environment |
| **UIBert** Bai et al. (2021) ⟳ | 2021.07 | ✔ | ✔ | N/A | N/A | N/A | 16,660 | 1 | Pre-training task |
| **Meta-GUI** Sun et al. (2022) ⟳ | 2022.05 | ✗ | ✔ | 7 | 4,684 | 11 | 1,125 | 5.3 | Multi-turn dialogues |
| **UGIF** Venkatesh et al. (2022) ⟳ | 2022.11 | ✔ | ✔ | 8 | 523 | 12 | 523 | 5.3 | Multilingual UI-grounded instructions |
| **AITW** Rawles et al. (2024b) ⟳ | 2023.12 | ✔ | ✗ | 7 | 715,142 | 357 | 30,378 | 6.5 | Large-scale interactions |
| **AITZ** Zhang et al. (2024c) ⟳ | 2024.03 | ✔ | ✗ | 7 | 18,643 | 70 | 2,504 | 7.5 | Chain-of-Action-Thought annotations |
| **GUI Odyssey** Lu et al. (2024a) ⟳ | 2024.06 | ✗ | ✔ | 9 | 7,735 | 201 | 7,735 | 15.4 | Cross-app navigation |
| **AndroidControl** Li et al. (2024a) ⟳ | 2024.07 | ✔ | ✔ | 8 | 15,283 | 833 | 15,283 | 4.8 | UI task scaling law |
| **AMEX** Chai et al. (2024) ⟳ | 2024.07 | ✔ | ✔ | 8 | 2,946 | 110 | 2,946 | 12.8 | Multi-level detailed annotations |
| **MobileViews** Gao et al. (2024) ⟳ | 2024.09 | ✔ | ✔ | N/A | N/A | 21,053 | N/A | N/A | Largest-scale mobile dataset |
| **JEDI** Xie et al. (2025a) | 2025.01 | ✔ | ✔ | N/A | N/A | Desktop | >4,000,000 | 1 | Large-scale data via UI decomposition synthesis |
| **ScaleTrack** Huang et al. (2025) | 2025.02 | ✔ | ✔ | N/A | N/A | N/A | N/A | N/A | Backtracking for GUI planning |

## 6.1    Datasets

The development of phone automation and GUI-based agents has been significantly propelled by the availability of diverse and richly annotated datasets. These datasets provide the foundation for training and evaluating

models that can understand and interact with mobile user interfaces using natural language instructions. In this subsection, we review several key datasets, highlighting their unique contributions and how they collectively advance the field. Table 9 summarizes these datasets, providing an overview of their characteristics.

Rico Deka et al. (2017) is the largest dataset from the early stage of GUI automation development, providing a solid foundation for understanding modern mobile interfaces and developing GUI agents. It includes various types of data, such as UI screenshots, view hierarchies, and UI metadata, offering valuable references for researchers and developers. Based on this, subsequent studies like RICO Semantics Sunkara et al. (2022), GUI-WORLD Chen et al. (2024a), and MobileViews Gao et al. (2024) have emerged, expanding the types and coverage of datasets and driving the growth of GUI agent research. Among them, MobileViews is currently the largest GUI dataset.

Early efforts in dataset creation focused on mapping natural language instructions to UI actions. PixelHelp Li et al. (2020) pioneered this area by introducing a problem of grounding natural language instructions to mobile UI action sequences. It decomposed the task into action phrase extraction and grounding, enabling models to interpret instructions like "Turn on flight mode" and execute corresponding UI actions. Building on this, UGIF Venkatesh et al. (2022) extended the challenge to a multilingual and multimodal setting. UGIF addressed cross-modal and cross-lingual retrieval and grounding, providing a dataset with instructions in English and UI interactions across multiple languages, thus highlighting the complexities of multilingual UI instruction following.

Addressing task feasibility and uncertainty, MoTIF Burns et al. (2021) introduced a dataset that includes natural language commands which may not be satisfiable within the given UI context. By incorporating feasibility annotations and follow-up questions, MoTIF encourages research into how agents can recognize and handle infeasible tasks, enhancing robustness in interactive environment.

For advancing UI understanding through pre-training, UIBert Bai et al. (2021) proposed a Transformer-based model that jointly learns from image and text representations of UIs. By introducing novel pre-training tasks that leverage the correspondence between different UI features, UIBert demonstrated improvements across multiple downstream UI tasks, setting a foundation for models that require a deep understanding of GUI layouts and components.

In the realm of multimodal dialogues and interactions, Meta-GUI Sun et al. (2022) proposed a GUI-based task-oriented dialogue system. This work collected dialogues paired with GUI operation traces, enabling agents to perform tasks through conversational interactions and direct GUI manipulations. It bridges the gap between language understanding and action execution within mobile applications.

Recognizing the need for large-scale datasets to train more generalizable agents, several works introduced extensive datasets capturing a wide range of device interactions. Android In The Wild (AITW) Rawles et al. (2024b) released a dataset containing hundreds of thousands of episodes with human demonstrations of device interactions. It presents challenges where agents must infer actions from visual appearances and handle precise gestures. Building upon AITW, Android In The Zoo (AITZ) Zhang et al. (2024c) provided fine-grained semantic annotations using the Chain-of-Action-Thought (CoAT) paradigm, enhancing agents' ability to reason and make decisions in GUI navigation tasks.

To address the complexities of cross-application navigation, GUI Odyssey Lu et al. (2024a) introduced a dataset specifically designed for training and evaluating agents that navigate across multiple apps. By covering diverse apps, tasks, and devices, GUI Odyssey enables the development of agents capable of handling real-world scenarios that involve integrating multiple applications and transferring context between them.

Understanding how data scale affects agent performance, AndroidControl Li et al. (2024a) studied the impact of training data size on computer control agents. By collecting demonstrations with both high-level and low-level instructions across numerous apps, this work analyzed in-domain and out-of-domain generalization, providing insights into the scalability of fine-tuning approaches for device control agents.

Focusing on detailed annotations to enhance agents' understanding of UI elements, AMEX Chai et al. (2024) introduced a comprehensive dataset with multi-level annotations. It includes GUI interactive element grounding, functionality descriptions, and complex natural language instructions with stepwise GUI-action

chains. AMEX aims to align agents more closely with human users by providing fundamental knowledge and understanding of the mobile GUI environment from multiple levels, thus facilitating the training of agents with a deeper understanding of page layouts and UI element functionalities.

Finally, we should focus on methods for generating, collecting, and annotating high-quality datasets. Dream-Struct Peng et al. (2024) leverages LLMs to generate data design concept descriptions based on target tasks. It then produces HTML code with target labels, embedding semantic tags within. In the post-processing phase, Bing Search API or DALL · E is used to replace placeholder graphic elements, resulting in the final visual content. This research offers a dataset, DreamUI, which includes 9,774 labeled UI interfaces for reference. OS-Genesis Sun et al. (2024) utilizes the method of Reverse Task Synthesis to automatically generate task instructions and corresponding action trajectories from interactions. It then integrates these with a trajectory reward model to produce high-quality and diverse GUI agent data. Learn-by-interact Su et al. (2025) uses LLMs to generate data through interaction with the environment and optimizes this data via *backward construction*. These high-quality data generation techniques reduce the dependency on manually labeled data, facilitating agents' rapid adaptation to new environments and tasks. Ferret-UI 2 Li et al. (2024e) uses the Set-of-Mark (SoM) visual prompt method to tag each UI component with bounding boxes and numerical labels to assist GPT-4o in recognition. Subsequently, GPT-4o generates question-and-answer task data related to UI components, covering multiple aspects of UI comprehension and thus producing high-quality training data. FedMobileAgent Wang et al. (2025b) automatically collects data during users' daily mobile usage and employs locally deployed VLM to annotate user actions, thereby generating a high-quality dataset. Furthermore, even in the absence of explicit ground truth annotations, we can infer user intentions through their interactions within the GUI to generate corresponding UI annotations Berkovitch et al. (2024). This approach opens up new directions for the collection and annotation of GUI data. Addressing the critical bottleneck of data scarcity and quality in GUI grounding, JEDI Xie et al. (2025a) introduces a novel data generation pipeline centered on UI decomposition and synthesis. This approach systematically constructs a massive dataset of over 4 million examples by breaking down GUIs into fundamental elements—icons, components, and layouts—and synthesizing diverse training scenarios. By fine-tuning models on this extensive and structured dataset, JEDI significantly enhances their fine-grained grounding capabilities. This improvement in grounding directly translates to state-of-the-art performance in complex agentic tasks on desktop environments, demonstrating that high-quality, large-scale synthesized data is a key factor in building more capable GUI agents. ScaleTrack Huang et al. (2025) introduces a novel training framework to improve agent learning through data enrichment and a new training paradigm. First, it scales the GUI grounding data by integrating samples generated from various isolated synthesis criteria, such as element referring and context awareness, into a unified training set. More notably, it introduces the concept of backtracking for GUI planning. By collecting data on historical actions and using a hybrid training strategy that requires the agent to predict both the next action and the previous one that led to the current state, this approach helps the agent better learn the intrinsic patterns of task execution.

Collectively, these datasets represent significant strides in advancing phone automation and GUI-based agent research. They address various challenges, from language grounding and task feasibility to large-scale device control and cross-app navigation. By providing rich annotations and diverse scenarios, they enable the training and evaluation of more capable, robust, and generalizable agents, moving closer to the goal of intelligent and autonomous phone automation solutions.

## 6.2 Benchmarks

The development of mobile GUI-based agents is not only reliant on the availability of diverse datasets but is also significantly influenced by the presence of robust benchmarks. These benchmarks offer standardized environment, tasks, and evaluation metrics, which are essential for consistently and reproducibly assessing the performance of agents. They enable researchers to compare different models and approaches under identical conditions, thus facilitating collaborative progress. In this subsection, we will review some of the notable benchmarks that have been introduced to evaluate phone GUI agents, highlighting their unique features and contributions. A summary of these benchmarks is provided in Table 10, which allows for a comparative understanding of their characteristics.

Table 10: Summary of benchmarks for phone GUI agents

| Benchmark | Date | Tasks | Task Completion | Action Quality | Resource Efficiency | Task Under-standing | Format Compliance | Completion Awareness | Reward | Eval Accuracy |
|---|---|---|---|---|---|---|---|---|---|---|
| **MobileEnv** Zhang et al. (2023b) | 2023.05 | 74 | ✔ | ✗ | ✗ | ✗ | ✗ | ✗ | ✔ | ✗ |
| **AutoDroid** Wen et al. (2024) | 2023.09 | N/A | ✔ | ✔ | ✗ | ✗ | ✗ | ✗ | ✗ | ✗ |
| **AndroidArena** Xing et al. (2024) | 2024.02 | N/A | ✔ | ✔ | ✔ | ✔ | ✔ | ✔ | ✔ | ✗ |
| **LlamaTouch** Zhang et al. (2024e) | 2024.04 | 496 | ✔ | ✔ | ✗ | ✔ | ✗ | ✔ | ✗ | ✔ |
| **B-MoCA** Lee et al. (2024b) | 2024.04 | 131 | ✔ | ✗ | ✔ | ✗ | ✗ | ✗ | ✗ | ✗ |
| **AndroidWorld** Rawles et al. (2024a) | 2024.05 | 116 | ✔ | ✗ | ✗ | ✗ | ✗ | ✗ | ✔ | ✗ |
| **MobileAgent Bench** Wang et al. (2024e) | 2024.06 | 100 | ✔ | ✔ | ✔ | ✗ | ✗ | ✗ | ✔ | ✔ |
| **AUITestAgent** Hu et al. (2024) | 2024.07 | N/A | ✔ | ✔ | ✗ | ✔ | ✔ | ✔ | ✔ | ✔ |
| **VisualAgent Bench** Liu et al. (2024c) | 2024.08 | 119 | ✔ | ✗ | ✔ | ✗ | ✗ | ✗ | ✗ | ✗ |
| **AgentStudio** Zheng et al. (2024b) | 2024.10 | 205 | ✔ | ✔ | ✗ | ✔ | ✗ | ✔ | ✔ | ✔ |
| **AndroidLab** Xu et al. (2024b) | 2024.11 | 138 | ✔ | ✔ | ✔ | ✔ | ✗ | ✗ | ✔ | ✔ |
| **A3** Chai et al. (2025) | 2025.01 | 201 | ✔ | ✗ | ✗ | ✗ | ✗ | ✗ | ✔ | ✔ |
| **AutoEval** Sun et al. (2025b) | 2025.03 | 93 | ✔ | ✗ | ✗ | ✗ | ✗ | ✗ | ✔ | ✔ |
| **LearnGUI** Liu et al. (2025a) | 2025.04 | 2,353 | ✔ | ✗ | ✗ | ✗ | ✗ | ✗ | ✔ | ✗ |
| **SPA-BENCH** Chen et al. (2025b) | 2025.05 | 340 | ✔ | ✗ | ✔ | ✗ | ✗ | ✗ | ✔ | ✔ |
| **Mobile-Bench-v2** Xu et al. (2025) | 2025.06 | N/A | ✔ | ✔ | ✗ | ✔ | ✗ | ✗ | ✔ | ✔ |

### 6.2.1 Evaluation Pipelines

Early benchmarks in the field of phone GUI agents focused on creating controlled environment for training and evaluating these agents. MobileEnv Zhang et al. (2023b), for example, introduced a universal platform for the training and evaluation of mobile interactions. It provided an isolated and controllable setting, with support for intermediate instructions and rewards. This emphasis on reliable evaluations and the ability to more naturally reflect real-world usage scenarios was a significant step forward.

To address the challenges presented by the complexities of modern operating systems and their vast action spaces, AndroidArena Xing et al. (2024) was developed. This benchmark was designed to evaluate large language model (LLM) agents within a complex Android environment. It introduced scalable and semi-automated methods for benchmark construction, with a particular focus on cross-application collaboration and user constraints such as security concerns.

Current research primarily focuses on the overall task success rate and often overlooks the evaluation of core capabilities such as GUI grounding of agents in real-world scenarios. AgentStudio Zheng et al. (2024b) provides a comprehensive platform that spans the entire development cycle, from environment setup and data collection to agent evaluation and visualization. AgentStudio also introduces three benchmark datasets: GroundUI, IDMBench, and CriticBench. These datasets are designed to evaluate agents' capabilities in GUI grounding, learning from videos, and success detection, respectively. Additionally, it introduces a benchmark suite comprising 205 real-world tasks to comprehensively evaluate agents' practical capabilities from multiple perspectives.

Recognizing the limitations in scalability and faithfulness of existing evaluation approaches, LlamaTouch Zhang et al. (2024e) presented a novel testbed. This testbed enabled on-device mobile UI task execution and provided a means for faithful and scalable task evaluation. It introduced fine-grained UI component annotation and a

multi-level application state matching algorithm. These features allowed for the accurate detection of critical information in each screen, enhancing the evaluation's accuracy and adaptability to dynamic UI changes.

B-MoCA Lee et al. (2024b) expanded the focus of benchmarking to include mobile device control agents across diverse configurations. By incorporating a randomization feature that could change device configurations such as UI layouts and language settings, B-MoCA was able to more effectively assess agents' generalization performance. It provided a realistic benchmark with 131 practical tasks, highlighting the need for agents to handle a wide range of real-world scenarios.

To provide a dynamic and reproducible environment for autonomous agents, AndroidWorld Rawles et al. (2024a) introduced an Android environment with 116 programmatic tasks across 20 real-world apps. This benchmark emphasized the importance of ground-truth rewards and the ability to dynamically construct tasks that were parameterized and expressed in natural language. This enabled testing on a much larger and more realistic suite of tasks.

For the specific evaluation of mobile LLM agents, MobileAgentBench Wang et al. (2024e) proposed an efficient and user-friendly benchmark. It addressed challenges in scalability and usability by offering 100 tasks across 10 open-source apps. The benchmark also simplified the extension process for developers and ensured that it was fully autonomous and reliable.

In the domain of GUI function testing, AUITestAgent Hu et al. (2024) introduced the first automatic, natural language-driven GUI testing tool for mobile apps. By decoupling interaction and verification into separate modules and employing a multi-dimensional data extraction strategy, it enhanced the automation and accuracy of GUI testing. The practical usability of this tool was demonstrated in real-world deployments.

AndroidLab Xu et al. (2024b) presented a systematic Android agent framework. This framework included an operation environment with different modalities and a reproducible benchmark. Supporting both LLMs and large multimodal models (LMMs), it provided a unified platform for training and evaluating agents. Additionally, it came with an Android Instruction dataset that significantly improved the performance of open-source models.

LearnGUI Liu et al. (2025a) offers a novel approach by introducing the first comprehensive benchmark specifically designed for demonstration-based learning in mobile GUI agents. Rather than pursuing universal generalization through larger datasets, it focuses on improving agent performance in unseen scenarios through human demonstrations. The benchmark comprises 2,252 offline tasks and 101 online tasks with high-quality human demonstrations.

Finally, to evaluate the practical performance of mobile GUI agents in complex real-world environments, VisualAgentBench Liu et al. (2024c) constructs a series of cross-domain tasks. This benchmark examines the agents' abilities in dynamic interaction and decision-making and provides abundant training trajectory data to support further performance improvement via behavior cloning. SPA-BENCH Chen et al. (2025b) presents a comprehensive suite for broad and fair agent comparison to address the narrow scope and manual evaluation requirements of previous benchmarks. It introduces 340 tasks spanning single-app, cross-app, and multilingual scenarios, notably including 58 third-party applications often omitted in prior work. A key contribution is its fully automated evaluation pipeline, which assesses both task completion and resource consumption without human intervention, enabling scalable and reproducible performance analysis across a wide range of integrated agents. A3 (Android Agent Arena) Chai et al. (2025) integrates 201 tasks from 21 widely-used third-party applications, covering common real-world user scenarios. It supports an extended action space compatible with any dataset annotation style. Additionally, the use of business-level LLMs automates task evaluation, reducing the need for manual assessment and enhancing scalability.

AutoEval Sun et al. (2025b) addresses the practicality and scalability challenges in mobile agent evaluation by introducing a framework that requires no manual effort to define task reward signals or implement evaluation codes. It employs a Structured Substate Representation to describe UI state changes during agent execution and utilizes a Judge System that can autonomously evaluate agent performance with over 94% accuracy compared to human verification.

To address the limitations of existing benchmarks, including overly clean environments and single-path evaluations, Mobile-Bench-v2 Xu et al. (2025) introduces a more realistic and comprehensive benchmark. It proposes a novel offline multi-path evaluation method that combines single-path checks with graph-based action search. To better reflect real-world conditions, the benchmark includes two specialized sub-datasets: Mobile-Bench-Noisy, which incorporates ads and pop-ups, and Mobile-Bench-Ambiguous, which facilitates active interactive evaluation by allowing agents to ask for clarification when instructions are vague. This approach pushes the evaluation of agent robustness and adaptability in more challenging and realistic scenarios.

Collectively, these benchmarks have made substantial contributions to the advancement of phone GUI agents. They have achieved this by providing a diverse environment, tasks, and evaluation methodologies. They have addressed various challenges, including scalability, reproducibility, generalization across configurations, and the integration of advanced models like LLMs and LMMs. By facilitating rigorous testing and comparison, they have played a crucial role in driving the development of more capable and robust phone GUI agents.

### 6.2.2 Evaluation Metrics

Evaluation metrics are crucial for measuring the performance of phone GUI agents, providing quantitative indicators of their effectiveness, efficiency, and reliability. This section categorizes and explains the various metrics used across different benchmarks based on their primary functions.

**Task Completion Metrics.** Task Completion Metrics assess how effectively an agent finishes assigned tasks. *Task Completion Rate* indicates the proportion of successfully finished tasks, with AndroidWorld Rawles et al. (2024a) exemplifying its use for real-device assessments. *Sub-Goal Success Rate* further refines this by examining each sub-goal within a larger task, as employed by AndroidLab Xu et al. (2024b), making it particularly relevant for complex tasks that require segmentation. *End-to-end Task Completion Rate*, used by LlamaTouch Zhang et al. (2024e), offers a holistic measure of whether an agent can see an entire multi-step task through to completion without interruption.

**Action Execution Quality Metrics.** These metrics evaluate the agent's precision and correctness when performing specific actions. *Action Accuracy*, adopted by AUITestAgent Hu et al. (2024) and AutoDroid Zhang & Zhang (2023), compares each executed action to the expected one. *Correct Step* measures the fraction of accurate steps in an action sequence, whereas *Correct Trace* quantifies the alignment of the entire action trajectory with the ground truth. *Operation Logic* checks if the agent follows logical procedures to meet task objectives, as AndroidArena Xing et al. (2024) demonstrates. *Reasoning Accuracy*, highlighted in AUITestAgent Hu et al. (2024), gauges how well the agent logically interprets and responds to task requirements.

**Resource Utilization and Efficiency Metrics.** These indicators measure how efficiently an agent handles system resources and minimizes redundant operations. *Resource Consumption*, tracked by AUITestAgent Hu et al. (2024) via Completion Tokens and Prompt Tokens, reveals how much computational cost is incurred. *Step Efficiency*, applied by AUITestAgent and MobileAgentBench Wang et al. (2024e), compares actual steps to an optimal lower bound, while *Reversed Redundancy Ratio*, used by AndroidArena Xing et al. (2024) and AndroidLab Xu et al. (2024b), evaluates unnecessary detours in the action path.

**Task Understanding and Reasoning Metrics.** These metrics concentrate on the agent's comprehension and analytical skills. *Oracle Accuracy* and *Point Accuracy*, used by AUITestAgent Hu et al. (2024), assess how well the agent interprets task instructions and verification points. *Reasoning Accuracy* indicates the correctness of the agent's logical deductions during execution, and *Nuggets Mining*, employed by AndroidArena Xing et al. (2024), measures the ability to extract key contextual information from the UI environment.

**Format and Compliance Metrics.** These metrics verify whether the agent operates within expected format constraints. *Invalid Format* and *Invalid Action*, for example, are tracked in AndroidArena Xing et al. (2024) to confirm that an agent's outputs adhere to predefined structures and remain within permissible action ranges.

**Completion Awareness and Reflection Metrics.** Such metrics evaluate the agent's recognition of task boundaries and its capacity to learn from prior steps. *Awareness of Completion*, explored in AndroidArena Xing

et al. (2024), ensures the agent terminates at the correct time. *Reflexion@K* measures adaptive learning by examining how effectively the agent refines its performance over multiple iterations.

**Evaluation Accuracy and Reliability Metrics.** These indicators measure the consistency and reliability of the evaluation process. *Accuracy*, as used in LlamaTouch Zhang et al. (2024e), validates alignment between the evaluation approach and manual verification, ensuring confidence in performance comparisons across agents.

**Reward and Overall Performance Metrics.** These metrics combine various performance facets into aggregated scores. *Task Reward*, employed by AndroidArena Xing et al. (2024), provides a single effectiveness measure encompassing several factors. *Average Reward*, used in MobileEnv Zhang et al. (2023b), further reflects consistent performance across multiple tasks, indicating the agent's stability and reliability.

These evaluation metrics together provide a comprehensive framework for assessing various dimensions of phone GUI agents. They cover aspects such as effectiveness, efficiency, reliability, and the ability to adapt and learn. By using these metrics, benchmarks can objectively compare the performance of different agents and systematically measure improvements. This enables researchers to identify strengths and weaknesses in different agent designs and make informed decisions about future development directions.

## 6.3 Performance Analysis and Current Limitations

To provide a comprehensive understanding of the current state of mobile GUI agents, we present a quantitative comparison of representative methods across key benchmarks in Table 11. The table encompasses different evaluation paradigms: visual grounding capabilities (ScreenSpot Cheng et al. (2024)), offline task execution (GUI-Odyssey Lu et al. (2024a), AndroidControl Li et al. (2024a)), and online real-world performance (AndroidWorld Rawles et al. (2024a)). We categorize methods by their training paradigms and highlight their backbone MLLMs, where the backbone represents the base model for training (for specialized models) or inference (for frameworks).

**Evaluation Details.** For comprehensive comparison, we adopt standardized evaluation metrics across benchmarks. For ScreenSpot, we calculate the average performance of mobile text and mobile icon grounding components. For GUI-Odyssey, following the OS-Atlas Wu et al. (2024d) evaluation protocol, we report the macro average of single-step success rates across four data split configurations. For AndroidControl, we report the average single-step success rate. For AndroidWorld, we calculate the overall task success rate. The performance data is primarily collected from recent studies, including UI-TARS Qin et al. (2025), GUI-R1 Lu et al. (2025b), and the scores reported by the models in their original papers, ensuring consistency and reliability of the reported metrics.

**Specialized Training Yields Consistent Improvements.** Comparing basic foundation models with GUI-specialized models and targeted frameworks, we observe consistent performance gains across all benchmarks. This demonstrates the necessity of domain-specific adaptation for mobile GUI tasks, whether through fine-tuning, reinforcement learning, or carefully designed prompting strategies.

**Strong Performance in Visual Grounding.** Current models have achieved promising results in mobile UI grounding tasks. On ScreenSpot, the best-performing RL-based method, UI-R1 Lu et al. (2025b), achieves an accuracy of 90.2%, indicating that visual perception and element localization capabilities have reached a relatively mature level.

**Reinforcement Learning Provides Significant Gains.** The dominance of RL is evident across offline evaluation dimensions, where it consistently delivers substantial improvements over supervised fine-tuning approaches. Methods like MagicGUI-RFT Tang et al. (2025) and UI-TARS Qin et al. (2025) achieve state-of-the-art results, reaching up to 93.5% on AndroidControl and 88.6% on GUI-Odyssey. This suggests that constructing high-quality RL datasets and designing effective reward mechanisms remain promising research directions.

**Gap Between Online and Offline Performance.** A notable gap persists between performance on offline datasets and complex online tasks. While RL-based agents dominate offline benchmarks, the highest scores on AndroidWorld are achieved by prompt-based and ensemble methods, with Mobile-Agent-v3 Ye et al.

Table 11: Performance comparison of representative methods across mobile GUI benchmarks. Methods are categorized by their training paradigms and backbone MLLMs. AC refers to AndroidControl and AW refers to AndroidWorld.

| Method | Backbone | ScreenSpot | GUI-Odyssey | AC(Low) | AC(High) | AW |
|---|---|---|---|---|---|---|
| **Basic Method** | | | | | | |
| Qwen2-VL-7B Wang et al. (2024f) | Qwen2-VL-7B | 68.1 | 60.2 | 82.6 | 69.7 | N/A |
| GPT-4o OpenAI (2024) | GPT-4o | 22.6 | 3.3 | 19.4 | 20.8 | 34.5 |
| Claude-3.5-Sonnet Anthropic (2024) | Claude-3.5 | N/A | 3.1 | 19.4 | 12.5 | 27.9 |
| **Prompt Engineering Method** | | | | | | |
| OmniParser Lu et al. (2024b) | GPT-4V | 75.5 | N/A | N/A | N/A | N/A |
| Agent S2 Agashe et al. (2025) | UI-TARS-72B + Claude-3.7 | N/A | N/A | N/A | N/A | 54.3 |
| Mobile-Agent-E Wang et al. (2025d) | GPT-4o + GUI-Owl-7B | N/A | N/A | N/A | N/A | 59.5 |
| Mobile-Agent-E Wang et al. (2025d) | GPT-4o + GUI-Owl-32B | N/A | N/A | N/A | N/A | 62.1 |
| Mobile-Agent-v3 Ye et al. (2025a) | GPT-4o + GUI-Owl-32B | N/A | N/A | N/A | N/A | **73.3** |
| **Supervised Fine-Tuning Method** | | | | | | |
| SeeClick Cheng et al. (2024) | Qwen-VL-9.6B | 65.0 | 53.9 | 75.0 | 59.1 | N/A |
| InfiGuiAgent Liu et al. (2025c) | Qwen2-VL-2B | 81.7 | N/A | N/A | N/A | 9.0 |
| Aria-UI Yang et al. (2024b) | Aria-24-8B | 83.1 | 36.5 | 67.3 | 10.2 | 44.8 |
| AGUVIS Xu et al. (2024c) | Qwen2-VL-7B | 86.7 | N/A | 80.5 | 61.5 | 37.1[9] |
| AGUVIS Xu et al. (2024c) | Qwen2-VL-72B | 89.9 | N/A | 84.4 | 66.4 | 26.1 |
| MagicGUI-CPT Tang et al. (2025) | Qwen-VL | N/A | 73.5 | 86.7 | 73.1 | N/A |
| **Reinforcement Learning Method** | | | | | | |
| UI-R1 Lu et al. (2025b) | Qwen2.5-VL-3B | **90.2** | 32.5 | 66.4 | 45.4 | N/A |
| UI-TARS Qin et al. (2025) | Qwen2-VL-2B | 84.3 | 83.4 | 89.3 | 68.9 | N/A |
| UI-TARS Qin et al. (2025) | Qwen2-VL-7B | 89.9 | 87.0 | 90.8 | 72.5 | 33.0[10] |
| UI-TARS Qin et al. (2025) | Qwen2-VL-72B | 88.7 | **88.6** | 91.3 | 74.7 | 46.6[11] |
| MagicGUI-RFT Tang et al. (2025) | Qwen-VL | N/A | 74.3 | **93.5** | 76.3 | N/A |
| MobileGUI Shi et al. (2025) | Qwen2.5-VL-7B | N/A | N/A | N/A | N/A | 30.0 |
| MobileGUI Shi et al. (2025) | Qwen2.5-VL-32B | N/A | N/A | N/A | N/A | 44.8 |
| UI-Venus Gu et al. (2025) | Qwen2.5-VL-7B | N/A | N/A | N/A | N/A | 49.1 |
| UI-Venus Gu et al. (2025) | Qwen2.5-VL-72B | N/A | N/A | N/A | N/A | 65.9 |
| GUI-Owl Ye et al. (2025a) | Qwen2.5-VL-7B | N/A | N/A | N/A | 72.8 | 66.4 |
| GUI-Owl Ye et al. (2025a) | Qwen2.5-VL-32B | N/A | N/A | N/A | **76.6** | N/A |

(2025a) reaching a 73.3% success rate. This suggests that current RL strategies, while effective for single-step accuracy, may not yet fully capture the long-horizon reasoning required for dynamic online environments. The performance drop for models evaluated on both types of benchmarks, such as UI-TARS (from over 90% on AndroidControl to 46.6% on AndroidWorld), underscores this challenge.

These results reveal that while significant progress has been made in foundational capabilities such as visual grounding, substantial challenges remain in achieving reliable real-world deployment of mobile GUI agents. The performance disparities across different evaluation settings—particularly the gap between offline and online performance—highlight several critical research gaps that require further investigation. We will discuss these challenges and potential solutions in the following section.

### 6.4 Critical Analysis of Datasets and Benchmarks

The datasets and benchmarks discussed have been the bedrock of progress in mobile agent research. Datasets like **Rico** Deka et al. (2017) and **AITW** Rawles et al. (2024b) have provided the massive-scale data needed to train foundational models capable of basic UI understanding and interaction. Similarly, benchmarks such as **AndroidWorld** Rawles et al. (2024a) and **MobileEnv** Zhang et al. (2023b) have created standardized playing fields, allowing for reproducible experiments and fair comparison between different agent architectures. This infrastructure has been indispensable for moving the field from theoretical concepts to tangible prototypes.

However, despite these foundational contributions, a critical analysis reveals several shared limitations that collectively hinder the field's progress toward robust, real-world agents. We will examine the distinct challenges posed by each.

#### 6.4.1 Limitations of Datasets

Despite advances, existing datasets exhibit several common weaknesses that constrain the capabilities of the agents trained on them. Table 12 summarizes these core issues.

Table 12: Critical analysis of limitations in datasets.

| Limitation | Manifestation & Impact | Path Forward |
|---|---|---|
| **Imbalanced Task Coverage** | Over-representation of simple, single-app tasks. This leads to agents that learn atomic skills but lack the strategic, long-horizon planning needed for complex workflows. | Prioritize collection of compositional, cross-app task demonstrations that include conditional logic and error recovery. |
| **Insufficient Annotation Depth** | Suffer from "semantic poverty," providing action trajectories ("what") but not the underlying intent ("why"). This encourages brittle behavioral cloning instead of deep, generalizable understanding. | Develop scalable methods for richer annotations (e.g., CoAT) that capture the agent's reasoning process and high-level goals. |
| **Lack of Real-World Dynamism** | Data is collected in "sterile" environments, free of interruptions like pop-ups or notifications. This results in agents that are not resilient and fail when faced with common dynamic events. | Introduce realistic, stochastic events during data collection; create datasets specifically designed for training error handling and recovery. |

**Imbalanced Task Coverage.** A primary issue is the limited scope and complexity of tasks. Datasets, even large-scale ones like **AITW** Rawles et al. (2024b), are heavily skewed towards simple, short-horizon tasks confined to a single application, such as navigating to a specific page or toggling a setting. While valuable for learning basic interactions, this fails to capture the reality of human smartphone usage. Real-world goals are often long-horizon and compositional, requiring agents to execute complex workflows across multiple applications (e.g., finding a restaurant address in a messaging app, opening it in a maps app, and then booking a ride). Critically underrepresented are tasks that demand conditional logic ("If the flight is delayed, text me"), memory of past interactions, and robust error recovery strategies. This imbalance leads to agents that excel at atomic sub-tasks but are incapable of the strategic planning and reasoning required to achieve complex, real-world objectives.

**Insufficient Annotation Depth.** Many datasets suffer from "semantic poverty." They primarily provide action trajectories—a sequence of taps and swipes—but lack deep annotations explaining the underlying intent or reasoning. A dataset might record a `tap` on a coordinate but not the reason *why* that tap was necessary for the user's goal (e.g., "to select the destination field"). This forces models into a mode of behavioral cloning, where they learn to mimic action sequences rather than understanding the task's semantic structure. This reliance on pattern matching is inherently brittle; a minor UI change can cause the agent to fail, as it hasn't learned the generalizable intent behind the action. While efforts like **AITZ** Zhang et al. (2024c) with its Chain-of-Action-Thought (CoAT) paradigm are a significant step toward richer annotations, creating such data is expensive and time-consuming, limiting its widespread adoption in the largest datasets.

**Lack of Real-World Dynamism.** Datasets are typically collected in "sterile," idealized environments, which results in agents that are not resilient to the unpredictable nature of real-world mobile interactions. These datasets rarely include common dynamic events such as incoming notifications, pop-up advertisements, system permission requests, login requirements, or app updates that alter the UI layout. Agents trained exclusively on this clean data are effectively "overfitting" to static UI representations. When deployed "in the wild," they are brittle and often fail catastrophically at the first sign of an unexpected interruption, as they have not been trained to ignore distractions, handle errors gracefully, or adapt to UI changes. This creates a significant and often disappointing gap between an agent's performance in a controlled evaluation and its utility in the hands of a real user.

### 6.4.2 Limitations of Benchmarks

Similarly, current benchmarks face critical limitations that affect the evaluation of agent capabilities. These are summarized in Table 13.

Table 13: Critical analysis of limitations in benchmarks.

| Limitation | Manifestation & Impact | Path Forward |
|---|---|---|
| **Narrow Evaluation Dimensions** | Over-reliance on binary task success rates. This optimizes for task completion at any cost, neglecting crucial metrics like efficiency, robustness, or human-like interaction. | Develop multi-faceted evaluation suites that holistically score agents on efficiency (e.g., step count), robustness, and error recovery capabilities. |
| **Constrained Task Realism** | Tasks are well-defined and unambiguous, unlike real human instructions. This trains agents to solve clear "puzzles" but not to handle the ambiguity of human language. | Design benchmarks with underspecified, context-dependent, and conversational instructions that better reflect real-world usage patterns. |
| **Lack of Robustness Testing** | Environments are static and predictable. This produces "brittle" agents that perform well in controlled tests but fail in dynamic, real-world scenarios. | Systematically introduce perturbations (e.g., UI changes, pop-ups, network latency) into benchmarks to explicitly measure and reward agent resilience. |
| **Unreliable Automated Evaluation** | • Rule-based methods are brittle and not scalable.
• LLM-as-a-judge approaches suffer from high cost and potential hallucinations. | • Develop hybrid evaluation systems.
• Create specialized, cost-effective judge models.
• Explore unsupervised reward modeling. |

**Narrow Evaluation Dimensions.** A primary limitation is the narrow scope of evaluation dimensions. As shown in Table 10, the vast majority of benchmarks prioritize the binary *task success rate* as the ultimate measure of performance. While this metric is fundamental, its dominance overshadows other critical aspects of agent quality. For instance, operational efficiency is often overlooked. An agent might complete a task by taking a convoluted path with numerous redundant steps, which would be unacceptable to a human user. Although some benchmarks like **MobileAgentBench** Wang et al. (2024e) and **AndroidArena** Xing et al. (2024) have begun to incorporate efficiency metrics like step count or redundancy ratios, these are often

treated as secondary analyses rather than core performance indicators. More importantly, crucial capabilities like robustness and error recovery are severely under-tested. A notable exception is **B-MoCA** Lee et al. (2024b), which introduces randomization in UI layouts and language settings to test generalization. However, this practice is not widespread, leading to agents that are proficient in controlled, static environments but brittle in the face of real-world unpredictability.

**Constrained Realism of Task Scenarios.** Furthermore, the tasks within most benchmarks lack ecological validity. They are typically atomic, well-defined, and unambiguous (e.g., "Find a flight from SFO to LAX on July 1st"). This fails to reflect the nature of real human instructions, which are often ambiguous ("Find me a cheap flight to LA sometime next month"), incomplete, context-dependent ("Book my usual ride home"), or conversational, requiring clarifying questions. Even benchmarks with programmatic task generation, such as **AndroidWorld** Rawles et al. (2024a), tend to create variations by altering parameters within structured templates rather than capturing the rich, underspecified nature of human goals. This discrepancy creates a generation of agents optimized for solving puzzles with clear rules, rather than serving as genuinely helpful assistants that can navigate the complexities and implicit intents of human communication. Consequently, a high score on a current benchmark does not reliably translate to high utility and user satisfaction in a real-world deployment.

**Lack of Robustness Testing.** Compounding these issues, most benchmark environments are static and predictable, failing to test for agent resilience. The real world is dynamic; UIs change, apps crash, and networks fail. Yet, very few benchmarks systematically evaluate an agent's ability to adapt to such perturbations. A notable exception is **B-MoCA** Lee et al. (2024b), which introduces randomization in UI layouts and language settings to test generalization. However, this practice is not widespread, leading to agents that are proficient in controlled, static environments but brittle in the face of real-world unpredictability.

**Challenges in Automated Evaluation.** Creating scalable and truly reliable automated evaluation systems remains a significant bottleneck. As illustrated in Figure 11, current approaches can be broadly categorized into two paradigms: rule-based evaluation and LLM-as-a-judge. **Rule-based evaluation**, employed by benchmarks like AndroidWorld Rawles et al. (2024a), LlamaTouch Zhang et al. (2024e), and AndroidLab Xu et al. (2024b), relies on programmatic logic to assess task success. This approach is heavily dependent on manual labor from domain experts and direct access to the app's internal data, such as databases or UI trees, which severely limits its scalability. For example, AndroidWorld Rawles et al. (2024a) required experts to manually script success and failure rules for each of its 116 tasks, and its verification process relies on reading an application's internal database to check for expected states. Similarly, LlamaTouch Zhang et al. (2024e)ouch requires humans to annotate the key UI states that define task completion, and it evaluates success by

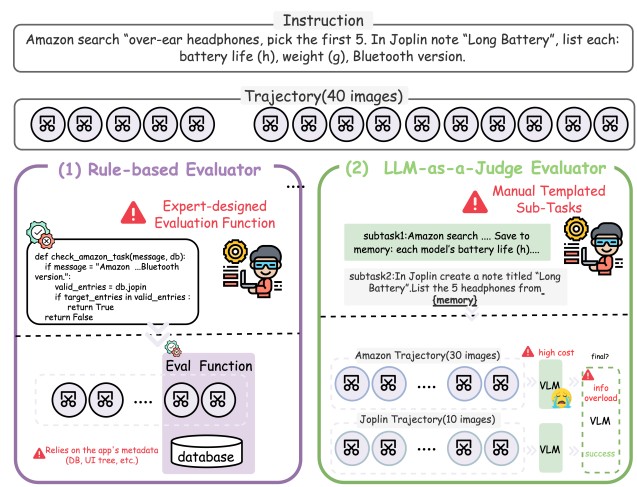

Figure 11: A Comparison of Automated Evaluation Paradigms: Rule-Based vs. LLM-as-a-Judge approaches for assessing mobile agent performance.

inspecting the UI tree of every step in an agent's trajectory. AndroidLab Xu et al. (2024b) follows a similar pattern, depending on expert annotations. The primary drawback of this paradigm is its poor scalability and brittleness; creating rules for new tasks is resource-intensive, and the evaluation logic can easily break with minor app updates. In contrast, the **LLM-as-a-judge** paradigm, adopted by benchmarks such as A3 Chai et al. (2025) and SPA-Bench Chen et al. (2024b), leverages powerful MLLMs to assess agent performance. While this approach reduces some manual effort, it still partially relies on human input and introduces new challenges, including high costs associated with using commercial MLLMs and the pervasive issue of model hallucination. For instance, SPA-Bench Chen et al. (2024b) combines manual annotation of critical states or sub-tasks with a GPT-4o-based judge. However, when evaluating complex tasks, the long trajectory of

screenshots fed to the model can lead to severe hallucinations, compromising the accuracy of the evaluation. A3 Chai et al. (2025) faces the same limitations, struggling with the trade-off between automation, cost, and the reliability issues caused by model hallucinations. A promising future direction for automated evaluation is to address these critical challenges by focusing on three key areas: reducing evaluation cost, mitigating evaluation hallucination, and minimizing manual intervention. Developing lightweight, open-source judge models specifically fine-tuned for GUI evaluation could lower the dependency on expensive commercial APIs. To combat hallucinations, hybrid systems that combine rule-based checks for factual verification with LLM judgments for more subjective assessments could be explored. Finally, advancing techniques for unsupervised or semi-supervised reward modeling could help in automatically identifying key task states, thus reducing the need for laborious manual annotation and making the entire evaluation pipeline more scalable and efficient.

Addressing these gaps in datasets and benchmarks is essential for driving the development of next-generation phone GUI agents that are not only successful in controlled environments but also robust, efficient, and reliable in the real world.

# 7 Challenges and Future Directions

Integrating LLMs into phone automation has propelled significant advancements but also introduced numerous challenges. The preceding analyses of frameworks (§4.6), models (§5.3), and the critical limitations of datasets and benchmarks (§6.4), further reinforced by quantitative results from our performance analysis (§6.3), reveal a set of core tensions that the field must resolve. Table 14 reframes these issues, moving from a simple list to a structured overview of the key trade-offs and research frontiers. This section elaborates on these points, outlining a path toward more capable, reliable, and user-centric agents.

**Frameworks and Components: The Performance Trilemma.** As analyzed in §4.6, the architectural design of agents faces a fundamental trilemma between **speed, memory, and cost**. Current perception pipelines, fusing rich data from screenshots and UI trees, introduce significant latency, and future work must also address **real-time perception** of dynamic content like videos. The brain's step-by-step reasoning is too slow for interactive use, creating a need for **acceleration methods**. Furthermore, agents possess effective short-term memory but lack the **persistent, long-term memory** required for true user-centric personalization. Finally, the agent's **action space**, typically limited to simple taps and swipes, needs to become more expressive to handle complex gestures. The high computational and financial **cost** of powerful models, especially in multi-agent setups, necessitates research into adaptive frameworks that can dynamically scale resources based on task complexity.

**Modeling Approaches: Bridging the Generalization-Specialization Gap.** The choice of modeling approach, as discussed in §5.3, presents a trade-off between the generality of prompt engineering and the specialization of training-based methods. Prompting offers excellent adaptability to new apps but is constrained by the performance ceiling of the base model and high operational costs. Conversely, training-based methods (SFT, RL) can achieve state-of-the-art performance and enable on-device deployment, but require costly data collection and struggle to generalize to unseen applications. The future direction lies in **hybrid methodologies**, as corroborated by performance analysis in §6.3 showing RL-based methods consistently outperform SFT-only approaches. A powerful paradigm is emerging: using SFT for a capable base, followed by RL to refine decision-making. Key research frontiers for these models include improving **high-resolution visual grounding** and enhancing **long-horizon task planning**.

**The Ecosystem of Data and Evaluation: Addressing the Capability Gap.** The quantitative results from §6.3 reveal a critical capability gap: while agents have achieved strong performance in foundational offline tasks like **visual grounding**, their success rates remain low on complex, multi-step **online tasks** such as those in AndroidWorld. This discrepancy highlights that the entire field is bottlenecked by the limitations of current datasets and benchmarks, as detailed in §6.4. Datasets often lack the task complexity, semantic depth, and real-world dynamism required to train for long-horizon reasoning. This forces agents to become proficient at single-step perception but leaves them unprepared for realistic workflows. Consequently, future work must focus on creating datasets with compositional, cross-app tasks and developing multi-faceted benchmarks that test for efficiency, robustness, and the ability to handle ambiguity, thereby bridging the gap between isolated skills and true task completion.

Table 14: Core trade-offs and future directions for LLM-powered phone GUI agents.

| State of the Art & Existing Techniques | Remaining Challenges | Future Directions |
|---|---|---|
| **Frameworks & Components:** Modular design (**Perception, Brain, Action**) and diverse architectures (Single/Multi-agent, Plan-then-Act). | **Performance Trilemma:** Balancing speed, memory, and cost. | • Develop adaptive architectures that scale from single to multi-agent.
• Build persistent long-term memory for true user-centric personalization.
• Achieve real-time perception for dynamic content (e.g., videos).
• Design more expressive, fine-grained action spaces. |
| **Modeling Approaches:** Two main paths: **Prompt Engineering** for high generality and **Training-Based Methods** (SFT, RL) for high specialization, with performance analysis showing RL methods yield significant gains. | **Generalization-Specialization Gap:** Bridging the high adaptability of prompting with the deep, specialized performance of training. | • Refine hybrid training pipelines (SFT + RL).
• Improve high-resolution visual grounding and long-horizon task planning. |
| **Datasets & Evaluation:** Foundational datasets and benchmarks have led to mature capabilities in specific areas like **visual grounding**. | **The Capability Gap:** Strong performance on single-step offline tasks does not translate to complex, long-horizon online tasks (e.g., low success rates on *AndroidWorld*). Precise grounding also remains a challenge. | • Collect datasets of compositional, cross-app tasks with richer intent annotations (e.g., CoAT).
• Build multi-faceted benchmarks that evaluate efficiency and robustness against systematic perturbations. |
| **On-Device Deployment:** Initial successes with model compression (*Octopus v2*, *TinyClick*) demonstrate feasibility of low-latency, private on-device agents. | **Practicality Trilemma:** Balancing model size, inference time, and on-device performance. | • Optimize the trade-off between model size and inference speed to ensure robust, real-world performance on resource-constrained devices. |
| **Security Threat Identification:** Research has identified sophisticated attack vectors (environmental injection, data poisoning) and preliminary defenses. | **End-to-End Trustworthiness:** Lack of comprehensive defenses and bias mitigation. | • Develop end-to-end security frameworks with real-time threat detection.
• Utilize privacy-preserving methods like Federated Learning and systematically test against safety benchmarks. |

**On-Device Deployment: The Practicality Trilemma.** While prompt engineering approaches are tied to the cloud, training-based methods have paved the way for lightweight, on-device deployment. Works like *Octopus v2* Chen & Li (2024) and *TinyClick* Pawlowski et al. (2024) have proven the feasibility of creating small, efficient models that offer low latency and enhanced privacy. However, a significant gap remains between these proofs-of-concept and a truly practical on-device agent. The next challenge is to ensure these lightweight models are not just fast, but also **robust**. They must maintain high performance when faced with the dynamic, unpredictable nature of real-world mobile use, including UI changes, system interruptions, and variable network conditions, all while navigating the critical trade-off between model size and inference time on resource-constrained devices.

**Ensuring Reliability, Security, and Ethical Alignment.** As agents gain access to sensitive data and perform critical tasks, their reliability, security, and ethical alignment are paramount. While traditional

systems may be susceptible to adversarial attacks, data breaches, and unintended actions Wu et al. (2024a), emerging threat paradigms are becoming more covert and sophisticated. For instance, multi-modal LLM-powered mobile agents are highly vulnerable to Active Environmental Injection Attacks (AEIA), where attackers manipulate environmental elements like notifications to mislead agents, achieving attack success rates up to 93% in benchmark tests Chen et al. (2025c). Such attacks exploit flaws in the agent's contextual reasoning rather than just its perceptual layer.

Furthermore, threats have migrated from the runtime environment into the model supply chain. Backdoor attacks, such as VisualTrap Yang et al. (2024a); Wang et al. (2024i); Ye et al. (2025b), implant an invisible visual trigger by injecting a small amount of "poisoned" data into the vision foundation model during pre-training. When an agent encounters this trigger on any interface, its behavior is hijacked to perform malicious actions even when following benign instructions, and this vulnerability persists after fine-tuning on clean data. Beyond external attacks, the inherent biases and unintended harmful behaviors of agents also pose a significant risk, directly addressing ethical concerns. Research shows that LLM agents can amplify human cognitive biases present in training data (e.g., omission bias) Sun et al. (2025a) or conform to the inherent biases of their base model, leading to behavior misaligned with user intent or societal norms Gan et al. (2024); Leng & Yuan (2024). Table 15 provides a taxonomy of these emerging security threats, categorizing them by paradigm, attack vector, and target component.

To address these challenges, the community is exploring solutions from multiple dimensions. On one hand, robust security protocols, error-handling techniques, and privacy-preserving methods are needed to maintain user trust Ma et al. (2024); Bai et al. (2024). On the other hand, to solve privacy issues in distributed settings, Federated Learning offers a promising paradigm. By training models locally on user devices and sharing only aggregated model updates instead of raw data, federated learning can fundamentally protect user privacy while leveraging diverse, real-world data Wang et al. (2025b). Works like FedMABench Wang et al. (2025c) provide comprehensive benchmarks for evaluating distributed mobile agents. Moreover, standardized safety benchmarks like MobileSafetyBench Lee et al. (2024a) are crucial for systematically evaluating agent robustness against harmful instructions or indirect prompt injections. Continuous monitoring and validation processes can detect vulnerabilities and mitigate risks in real-time Lee et al. (2023). Ensuring that agents behave predictably, respect user privacy, and maintain consistent performance under challenging conditions will be crucial for widespread adoption and long-term sustainability.

Table 15: Taxonomy of Security Threats to GUI Agents.

| Method | Attack Paradigm | Attack Vector | Target Component | Stealthiness | Requires Model Access |
|---|---|---|---|---|---|
| CLIP Attack Wu et al. (2024a) | Adversarial Input | Visual Perturbation | Perception | High | ✔ |
| Adversarial Grounding Zhao et al. (2025) | Adversarial Input | Visual Perturbation | Visual Grounding | High | ✔ |
| AEIA-MN Chen et al. (2025c) | Environmental Injection | Content Manipulation | Reasoning | High | ✗ |
| FPI Chen et al. (2025a) | Environmental Injection | Content Manipulation | Reasoning | High | ✗ |
| EVA Lu et al. (2025a) | Environmental Injection | Content Manipulation | Reasoning | High | ✗ |
| Hijacking JARVIS Liu et al. (2025b) | Environmental Injection | Content Manipulation | Reasoning | Variable | ✗ |
| VisualTrap Ye et al. (2025b) | Backdoor Attack | Data Poisoning | Visual Grounding | High | ✔(Training Phase) |

Addressing these challenges requires a concerted, multi-faceted effort. Progress in agent capabilities is not just about scaling models, but about building a robust ecosystem around them: creating deeper datasets, designing more holistic evaluations, developing adaptive frameworks that balance performance with efficiency, and embedding security and ethics from the ground up. By tackling these core issues, the next generation of LLM-powered phone GUI agents can evolve from promising prototypes into indispensable tools that are efficient, trustworthy, and seamlessly integrated into users' daily lives.

# 8 Conclusion

In this paper, we have presented a comprehensive survey of recent developments in LLM-driven phone automation technologies, illustrating how large language models can catalyze a paradigm shift from static script-based approaches to dynamic, intelligent systems capable of perceiving, reasoning about, and operating

on mobile GUIs. We examined a variety of frameworks, including single-agent architectures, multi-agent collaborations, and plan-then-act pipelines, demonstrating how each approach addresses specific challenges in task complexity, adaptability, and scalability. In parallel, we analyzed both prompt engineering and training-based techniques (such as supervised fine-tuning and reinforcement learning), underscoring their roles in bridging user intent and device action.

Beyond clarifying these technical foundations, we also spotlighted emerging research directions and provided a critical appraisal of persistent obstacles. These include ensuring robust dataset coverage, optimizing LLM deployments under resource constraints, meeting real-world demand for user-centric personalization, and maintaining security and reliability in sensitive applications. We further emphasized the need for standardized benchmarks, proposing consistent metrics and evaluation protocols to fairly compare and advance competing designs.

Looking ahead, ongoing refinements in model architectures, on-device inference strategies, and multimodal data integration point to an exciting expansion of what LLM-based phone GUI agents can achieve. We anticipate that future endeavors will see the convergence of broader AI paradigms—such as embodied AI and AGI—into phone automation, thereby enabling agents to handle increasingly complex tasks with minimal human oversight. Overall, this survey not only unifies existing strands of research but also offers a roadmap for leveraging the full potential of large language models in phone GUI automation, guiding researchers toward robust, user-friendly, and secure solutions that can adapt to the evolving needs of mobile ecosystems.

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
