# OpenReview forum: "LLM-Powered GUI Agents in Phone Automation: Surveying Progress and Prospects"
_TMLR — Accepted by TMLR_

### Review · Reviewer_JVjB · 2025-07-13

**Summary Of Contributions:**

This paper provides a survey of LLM-powered GUI agents for phone automation. It proposes a taxonomy of agent frameworks, modeling methods, datasets, and benchmarks. It provides a background and evolution of phone GUI automation and its applications. The survey also outlines key challenges and future directions.

**Audience:**

Yes

**Claims And Evidence:**

Yes

**Requested Changes:**

- Given the existing surveys on using LLM for GUI control and surveys on efficient models on phones,  more clearly highlighting the key contributions/differences (in the introduction) of this survey compared to others would be beneficial.

- Adding more quantitative evaluation comparisons would be helpful for the audience.

- Consider adding the distributed LLM perspective.

**Strengths And Weaknesses:**

Strength:
- Timely and Relevant Topic: The paper addresses an important and fast-growing intersection of LLMs and mobile GUI automation.
- Comprehensiveness: Covers a wide breadth of frameworks, models, datasets, and commercial systems with detailed referencing to recent literature.
- Clear Taxonomy: The classification is well-organized, intuitive, and captures key axes of design and development in the field.

Weekness:
- Evaluation comparison Gaps: The paper lacks a quantitative or comparative synthesis of model performance across benchmarks. Visuals/tables summarizing such comparisons would strengthen the utility of the survey.
- Quantitative understanding of the challenges are lacking: it is unclear how to quantify the challenges for eventual deployment of LLM for phone GUI control.
- Limited discussion on the distributed setup and privacy: Very limited discussion on how to support distributed inference approaches for running LLM models across cloud and on the phone, or across multiple devices, and the corresponding privacy and security concern.

---

> ### Author Response · Authors · 2025-09-02
> **Response to Reviewer JVjB Part 1**
>
> Thank you for your constructive review and for recognizing the timeliness and comprehensiveness of our survey. We address your concerns below.
>
> *W1, W2 / RC2 (Evaluation comparison Gaps / Quantitative understanding of the challenges are lacking / Adding more quantitative evaluation comparisons): The paper lacks a quantitative or comparative synthesis of model performance across benchmarks.*
>
> **Response.** Thank you for your valuable feedback on quantitative evaluation comparison. We have addressed your concern by adding **Table 11: Performance comparison of representative methods across mobile GUI benchmarks.** to the revised manuscript, which provides a comprehensive performance comparison of representative methods on key mobile GUI benchmarks.
>
> * **Revisions to the Datasets Section:** We have added a new subsection **Section 6.3** titled "Performance Analysis and Current Limitations" to the datasets section, which systematically analyzes the quantitative performance of existing methods. This analysis covers visual grounding capabilities (`ScreenSpot`), offline task execution (`GUI-Odyssey`, `AndroidControl`), and online real-world performance (`AndroidWorld`).
> * **Quantitative Understanding of Challenges:** Based on these quantitative results, we have identified several key performance bottlenecks:
>   1. Although models perform well on visual grounding tasks (top methods achieving over 90.2% accuracy), they are less effective in complex online tasks (often below 50% on AndroidWorld).
>   2. Reinforcement learning methods significantly outperform supervised learning methods across all dimensions, indicating the great potential of this research direction.
>   3. Error propagation in multi-step reasoning tasks severely impacts overall performance.
> * **Corresponding Revisions to the Challenges Section:** We have restructured the challenges section to align with these quantitative findings, with a particular emphasis on key challenges such as generalization, data acquisition efficiency, and robustness in multi-step scenarios.
>
> **Reference**
>
> [1] SeeClick: Harnessing gui grounding for advanced visual gui agents.
>
> [2] GUI Odyssey: A Comprehensive Dataset for Cross-App GUI Navigation on Mobile Devices.
>
> [3] On the Effects of Data Scale on Computer Control Agents.
>
> [4] AndroidWorld: A dynamic benchmarking environment for autonomous agents.

---

> ### Author Response · Authors · 2025-09-02
> **Response to Reviewer JVjB Part 2**
>
> *W3 / RC3 (Limited discussion on the distributed setup and privacy / Consider adding the distributed LLM perspective): Very limited discussion on how to support distributed inference approaches for running LLM models across cloud and on the phone, or across multiple devices, and the corresponding privacy and security concern.*
>
> **Response.** We thank the reviewer for astutely pointing out the lack of discussion on distributed settings and privacy protection. We acknowledge this as a critical research direction and have added significant content in the revised version. The changes are as follows:
>
> 1. In the revised **Section 7**, within the subsection now titled **"Ensuring Reliability, Security, and Ethical Alignment,"** we have added a dedicated part to discuss distributed systems and privacy protection.
> 2. We have explicitly introduced **Federated Learning** as a key privacy-preserving paradigm. We explain how its core mechanism, training models locally on user devices and only sharing aggregated model parameters, can leverage real-world data from heterogeneous environments to train and optimize agents without accessing raw sensitive data.
> 3. We cite related work such as **`FedMBench`** to show that there are already specialized benchmarks in this area to evaluate and promote the development of distributed mobile agents, thus closely linking this cutting-edge direction with the core content of our survey.
>
> **Reference**
>
> [1] FedMABench: Benchmarking Mobile Agents on Decentralized Heterogeneous User Data.
>
> By introducing Federated Learning, we have integrated the perspective of distributed LLMs into our security and privacy discussion as you guided. We now explicitly discuss how to achieve distributed training and deployment of models while protecting user privacy, which provides an important technical path for the future development of mobile GUI agents. We believe that with these changes, our paper now provides a more detailed, cutting-edge, and structured discussion of the challenges faced by GUI agents, especially in terms of security, ethics, and distributed privacy.

---

> ### Author Response · Authors · 2025-09-02
> **Response to Reviewer JVjB Part 3**
>
> *RC1: More clearly highlighting the key contributions/differences of this survey compared to others would be beneficial.*
>
> **Response.** Thank you for this critical feedback. We would like to clarify that our original manuscript already detailed our survey’s five core contributions in the introduction. However, we agree that a more direct and comprehensive comparison with existing work is needed to better highlight our distinctiveness. To address this, we have undertaken a significant structural revision by introducing a dedicated **new section, "Section 2: Related Work."** This section now serves as the central hub for systematically differentiating our work.
>
> Within this new section, we not only consolidate all comparisons and introduce a redesigned, more nuanced comparative table (Table 1), but more importantly, we articulate our unique contribution through **three distinct pillars** that are deeply rooted in a **Mobile-Specific Analysis**, a perspective absent in prior work. As detailed in **Section 2.2**, these pillars are:
>
> 1. **A Unified Methodological Framework Tailored for Mobile:** Unlike platform-agnostic surveys, we propose a comprehensive taxonomy organized into *Frameworks* (Section 4), *Models* (Section 5), and *Resources* (Section 6), with each pillar analyzed through a uniquely mobile-centric lens. For **Frameworks**, we ground our analysis in the specific I/O modalities of smartphones, detailing mobile-specific perceptual information (e.g., view hierarchies, Section 4.2) and the distinct touch-based action space (Section 4.3). For **Models**, we assess approaches based on their suitability for the severe constraints of **on-device deployment**. For **Resources**, we provide a targeted review of datasets and benchmarks built specifically for **phone GUI automation** (Section 6). This multi-faceted, mobile-specific methodological breakdown is a core contribution that other surveys do not offer.
> 2. **An In-depth Analysis of the Mobile Automation Trajectory:** Our survey provides a crucial historical narrative focused specifically on phone automation. We first dissect the pre-LLM era, identifying four central, long-standing challenges that hindered traditional methods like RPA and script-based testing: **Limited Generality**, **High Maintenance Costs**, **Poor Intent Comprehension**, and **Weak Screen GUI Perception** (Section 3). We then demonstrate how LLMs directly address these mobile-specific bottlenecks, establishing *why* they represent a paradigm shift *for mobile automation*. Specifically, we analyze how LLMs provide sophisticated **Contextual Semantic Understanding** to overcome intent ambiguity, leverage **Multi-Modal Perception** to interpret complex GUIs, and employ advanced **Reasoning and Decision-Making** to handle dynamic, cross-app workflows. This historical and problem-solution analysis provides a clear trajectory of the field's development.
> 3. **A Forward-Looking Perspective on Phone-Centric Challenges:** Our analysis of future directions (Section 7) is also grounded in the mobile ecosystem. We prioritize challenges that are most acute on phones, such as the technical hurdles of **on-device deployment**, the heightened need for **privacy and security** on personal devices, and the complexity of **cross-app workflow automation**.
>
> We believe that by creating this dedicated chapter and structuring our distinctiveness around these three deeply mobile-centric pillars, we now offer a much stronger and more persuasive argument for the unique value our survey provides.

---

### Review · Reviewer_DAiC · 2025-07-19

**Summary Of Contributions:**

This paper provides a timely and focused survey on the integration of large language models (LLMs) into phone GUI automation. It fills a clear gap in the literature by examining how LLM-based agents have transformed mobile interface tasks, which previous surveys on GUI automation did not specifically address. In particular, the authors systematically review recent advances and propose a structured taxonomy of LLM-powered phone agents. They outline key frameworks (e.g. single-agent vs. multi-agent vs. plan-then-act) and modeling approaches (prompt-based vs. training-based), along with the necessary datasets and benchmarks that have emerged for this domain. The survey also offers an insightful analysis of why LLMs improve phone automation – highlighting how advanced language understanding, multimodal perception, and reasoning capabilities allow these agents to better bridge user intent and GUI actions. Additionally, the paper introduces the latest developments (including new datasets and evaluation benchmarks) and identifies open challenges such as data diversity, on-device efficiency limitations, user-centric adaptation needs, and security/privacy concerns. Overall, by synthesizing recent work and offering future directions, the paper serves as a valuable reference for researchers and practitioners interested in scalable, intelligent phone GUI agents.

**Audience:**

Yes

**Broader Impact Concerns:**

The authors effectively identify relevant concerns around privacy, security, and user adaptation. To enhance completeness, briefly mentioning the potential risk of biases or unintended harmful behaviors in deployed LLM systems would strengthen the ethical discussion.

**Claims And Evidence:**

Yes

**Requested Changes:**

Clarify Distinctiveness:
Clearly articulate in the introduction or a dedicated subsection precisely how and why a survey specifically dedicated to phone GUI automation with LLMs is fundamentally different and practically more valuable than previous broader surveys on multimodal or general GUI agents.

Deepen LLM-Specific Insights:
Include a dedicated analytical discussion on unique strengths, limitations, and open problems specifically related to the use of LLMs, highlighting capabilities such as in-context learning, scalability, adaptability to unseen tasks, or emergent behaviors specifically beneficial to phone automation scenarios.

**Strengths And Weaknesses:**

Strengths:

The paper presents a timely and comprehensive survey on using large language models (LLMs) for phone GUI automation. It offers a structured taxonomy, categorizes existing frameworks, models, datasets, and benchmarks, and clearly identifies key challenges and future research directions. The authors convincingly illustrate how LLMs address traditional limitations in automation through improved semantic understanding, multimodal perception, and reasoning capabilities.

Weaknesses:

Limited Distinctiveness:
While the paper provides a specialized focus on phone GUI automation using LLMs, its fundamental frameworks and methodological discussions largely overlap with existing broader surveys (e.g., multimodal GUI agents, foundation model-based agents). The survey's distinct contribution—namely, its explicit emphasis on phone-specific tasks using LLMs—remains somewhat incremental and could be more convincingly highlighted.

Insufficient Depth on LLM-Centric Analysis:
The survey occasionally appears as a comprehensive listing of related works rather than providing a deeper critical synthesis around the unique advantages or limitations of LLM-based approaches. More explicit discussion is needed regarding how LLM-specific capabilities (such as emergent behaviors, multimodal reasoning, or in-context learning) significantly surpass traditional methods or other foundation model strategies.

---

> ### Author Response · Authors · 2025-09-02
> **Response to Reviewer DAiC Part 1**
>
> We sincerely appreciate your valuable feedback and recognition of our work's contributions. We address your concerns below.
>
> *W1 / RC1 (Limited Distinctiveness / Clarify Distinctiveness): The survey's distinct contribution remains somewhat incremental and could be more convincingly highlighted compared to broader surveys.*
>
> **Response.** Thank you for this critical feedback. We would like to clarify that our original manuscript already detailed our survey’s five core contributions in the introduction. However, we agree that a more direct and comprehensive comparison with existing work is needed to better highlight our distinctiveness. To address this, we have undertaken a significant structural revision by introducing a dedicated **new section, "Section 2: Related Work."** This section now serves as the central hub for systematically differentiating our work.
>
> Within this new section, we not only consolidate all comparisons and introduce a redesigned, more nuanced comparative table (Table 1), but more importantly, we articulate our unique contribution through **three distinct pillars** that are deeply rooted in a **Mobile-Specific Analysis**, a perspective absent in prior work. As detailed in **Section 2.2**, these pillars are:
>
> 1. **A Unified Methodological Framework Tailored for Mobile:** Unlike platform-agnostic surveys, we propose a comprehensive taxonomy organized into *Frameworks* (Section 4), *Models* (Section 5), and *Resources* (Section 6), with each pillar analyzed through a uniquely mobile-centric lens. For **Frameworks**, we ground our analysis in the specific I/O modalities of smartphones, detailing mobile-specific perceptual information (e.g., view hierarchies, Section 4.2) and the distinct touch-based action space (Section 4.3). For **Models**, we assess approaches based on their suitability for the severe constraints of **on-device deployment**. For **Resources**, we provide a targeted review of datasets and benchmarks built specifically for **phone GUI automation** (Section 6). This multi-faceted, mobile-specific methodological breakdown is a core contribution that other surveys do not offer.
> 2. **An In-depth Analysis of the Mobile Automation Trajectory:** Our survey provides a crucial historical narrative focused specifically on phone automation. We first dissect the pre-LLM era, identifying four central, long-standing challenges that hindered traditional methods like RPA and script-based testing: **Limited Generality**, **High Maintenance Costs**, **Poor Intent Comprehension**, and **Weak Screen GUI Perception** (Section 3). We then demonstrate how LLMs directly address these mobile-specific bottlenecks, establishing *why* they represent a paradigm shift *for mobile automation*. Specifically, we analyze how LLMs provide sophisticated **Contextual Semantic Understanding** to overcome intent ambiguity, leverage **Multi-Modal Perception** to interpret complex GUIs, and employ advanced **Reasoning and Decision-Making** to handle dynamic, cross-app workflows. This historical and problem-solution analysis provides a clear trajectory of the field's development.
> 3. **A Forward-Looking Perspective on Phone-Centric Challenges:** Our analysis of future directions (Section 7) is also grounded in the mobile ecosystem. We prioritize challenges that are most acute on phones, such as the technical hurdles of **on-device deployment**, the heightened need for **privacy and security** on personal devices, and the complexity of **cross-app workflow automation**.
>
> We believe that by creating this dedicated chapter and structuring our distinctiveness around these three deeply mobile-centric pillars, we now offer a much stronger and more persuasive argument for the unique value our survey provides.

---

> ### Author Response · Authors · 2025-09-02
> **Response to Reviewer DAiC Part 2**
>
> *W2 / RC2 (Insufficient Depth on LLM-Centric Analysis / Deepen LLM-Specific Insights): The survey appears as a listing of works rather than a deeper critical synthesis around the unique advantages or limitations of LLM-based approaches.*
>
> **Response.** Thank you for this insightful feedback. We agree that a deep, LLM-centric critical synthesis is crucial. We would like to clarify that our original manuscript was structured from the outset to provide this analysis around the unique strengths, limitations, and open problems of LLMs. Specifically:
>
> * **Section 3.2 ("Challenges of Traditional Methods")** was dedicated to a critical analysis that demonstrates not just *what* LLMs can do, but *why* traditional approaches fundamentally fall short due to limitations in generality, maintenance, intent comprehension, and perception.
> * **Section 3.3 ("LLMs Boost Phone Automation")** was dedicated to analyzing the unique strengths of LLMs, detailing four key capabilities that create qualitative advantages: **Scaling Laws, Contextual Semantic Understanding, Multi-Modal Perception, and Reasoning and Decision Making**.
> * **Section 7 ("Challenges and Future Directions")** has, from the beginning, systematically addressed the unique limitations and open problems for LLM-powered agents, such as dataset bottlenecks, on-device deployment difficulties, and security and reliability.
>
> We acknowledge that the analysis across these sections could be more prominent and systematically integrated. To that end, we have undertaken several key revisions to deepen the critical synthesis:
>
> * **Added Direct Comparative Analysis Sections:** We introduced new, dedicated subsections for critical analysis of **frameworks (Section 4.6)**, **modeling approaches (Section 5.3)**, and **datasets and benchmarks (Section 6.4)**, consolidating and sharpening points from the original draft into focused discussions.
> * **Provided Quantitative Evidence and Analysis:** We have added a new subsection titled **"Performance Analysis and Current Limitations" (Section 6.3)** within the **Datasets and Benchmarks chapter (Section 6)**. In this section, we introduced **Table 11** to provide a comprehensive quantitative comparison of representative methods across key mobile GUI benchmarks (covering visual grounding, offline task execution, and online real-world performance). Based on this data, we systematically analyze the performance bottlenecks of existing methods (e.g., strong visual grounding but poor performance on complex online tasks) and reveal the advantages of different technical routes (e.g., reinforcement learning over supervised fine-tuning). This provides a data-driven critical synthesis of the limitations and potential of LLMs in the mobile automation domain.
> * **Restructured and Deepened the Challenges Section:** We **restructured the original Section 7**, adding **Table 14** to directly visualize the "existing techniques vs. challenges" and reorganizing the subsections for a clearer narrative. This also allowed us to deepen the discussion on LLM-specific limitations, such as the vulnerability to AEIA attacks.
>
> We believe these comprehensive revisions provide the deep, critical synthesis you suggested, weaving our LLM-centric analysis more cohesively throughout the manuscript and specifically evaluating the unique contributions and limitations of these approaches.

---

> ### Author Response · Authors · 2025-09-02
> **Response to Reviewer DAiC Part 3**
>
> *Broader Impact Concerns: Briefly mentioning the potential risk of biases or unintended harmful behaviors in deployed LLM systems would strengthen the ethical discussion.*
>
> **Response.** We are very grateful for this constructive suggestion. We fully agree that a discussion of potential biases and unintended harmful behaviors is essential to strengthening the ethical dimension of our paper. In response, we have made the following revisions:
>
> 1. We have renamed the relevant subsection in Section 7 to **"Ensuring Reliability, Security, and Ethical Alignment"** to better reflect our expanded discussion and have conducted an in-depth survey of the latest literature in GUI agent security.
> 2. In this section, we have added new content that systematically explores the risks of **biases and potentially harmful behaviors** in LLM agents. We cite recent research to discuss how agents might amplify inherent human cognitive biases from the training data (e.g., "omission bias") or adhere to the intrinsic biases of their base models, leading to behaviors misaligned with user intent or social norms.
> 3. We also systematically discuss several emerging and more covert **types of attacks**, including Active Environmental Injection Attacks (AEIA), backdoor attacks (such as VisualTrap), and various forms of agent hijacking and indirect prompt injection.
> 4. Finally, we have introduced standardized security assessment benchmarks like **`MobileSafetyBench`** to emphasize that the academic community is striving to measure and mitigate these ethical risks through systematic methods.
>
> **Reference**
>
> [1] VisualTrap: A Stealthy Backdoor Attack on GUI Agents via Visual Grounding Manipulation.
>
> [2] MobileSafetyBench: Evaluating Safety of Autonomous Agents in Mobile Device Control.
>
> We believe these revisions not only directly address your concerns but also significantly enhance the depth and completeness of the ethical discussion in our paper.

---

### Review · Reviewer_u3op · 2025-08-17

**Summary Of Contributions:**

The paper makes several contributions: it provides a dedicated and systematic survey on LLM-powered phone GUI automation. It introduces a comprehensive taxonomy that organizes the field across agent frameworks, modeling paradigms, and evaluation resources. By tracing the evolution from traditional script- and RPA-based automation to LLM-driven systems, the paper clarifies the developmental trajectory of phone automation and highlights why LLMs transform the landscape. It further consolidates recent resources, including datasets, benchmarks, and commercial deployments, offering a valuable reference for both researchers and practitioners. Finally, it identifies open challenges such as dataset diversity, deployment efficiency, user adaptation, and security, and outlines future research directions.


While it has some merit as a survey paper, this work does not introduce any novel techniques, and as the reviewer is not an expert in evaluating pure survey papers, the significance of this survey in advancing the field is not entirely clear, even though the problem itself is interesting

**Audience:**

No

**Broader Impact Concerns:**

While the survey itself is descriptive and does not propose new algorithms, the broader impact raises several concerns. First, automation at the phone level involves sensitive personal data. Second, reliance on LLM-driven agents may exacerbate accessibility divides. Third, there is the risk of over-automation—leading to safety and trust issues. Finally, the survey highlights potential risks of malicious use, such as automated fraud or unauthorized account manipulation through GUI agents.

**Claims And Evidence:**

No

**Requested Changes:**

See weakness.

From the survey, many of the enabling techniques already exist in research prototypes and some commercial deployments, but the paper is clear that real-world phone automation with LLM agents is still far from solved. Can the authors make a table that summarize “existing techniques vs. challenges”?

**Strengths And Weaknesses:**

Strengths: see Summary Of Contributions

Weaknesses:

1. The paper sometimes reads as a catalog of existing works rather than a critical synthesis. For example, the taxonomy could be accompanied by more comparative analysis (e.g., trade-offs between prompt vs. training-based methods, benchmarks’ coverage gaps).

2. The title and abstract emphasize phone automation, but parts of the discussion (e.g., Anthropic’s Computer Use, PC/web agents) drift toward general GUI agents. The scope could be tightened or reframed more explicitly as cross-device GUI automation with emphasis on phones.

3. Although datasets and benchmarks are listed, the survey could provide more in-depth analysis of their strengths/weaknesses.

---

> ### Author Response · Authors · 2025-09-02
> **Response to Reviewer u3op Part 1**
>
> We sincerely appreciate your thoughtful review and valuable suggestions. We address your concerns below and summarize the corresponding improvements.
>
> *W1: The paper sometimes reads as a catalog of existing works rather than a critical synthesis. For example, the taxonomy could be accompanied by more comparative analysis (e.g., trade-offs between prompt vs. training-based methods, benchmarks’ coverage gaps).*
> **Response.** Thank you for this valuable feedback. We fully agree that an excellent survey must provide insightful critical synthesis, not just a catalog of existing works. We would first like to clarify that our original manuscript was already built upon an analytical framework, which we have now significantly deepened and expanded based on your insightful suggestions.
>
> **Our initial draft already provided multi-faceted analysis and synthesis beyond a simple catalog:**
>
> 1. **Establishing a Problem-Solution Narrative (Section 3):** We did not simply list techniques. Instead, in Section 3.2, we systematically diagnosed the four core challenges of traditional, pre-LLM automation methods (e.g., poor intent comprehension, high maintenance costs). Then, in Section 3.3, we demonstrated precisely how the key capabilities of LLMs (e.g., contextual semantic understanding, multi-modal perception) directly address these long-standing industry bottlenecks.
> 2. **Providing a Theoretical Framework (Section 4):** At the beginning of Section 4, we modeled the agent's decision-making process as a Partially Observable Markov Decision Process (POMDP). This provides a unified, theoretical lens for the entire field, allowing for an understanding and comparison of the underlying logic of different agent architectures that goes beyond a mere description of their implementations.
> 3. **Analyzing Architectural Paradigms (Section 5):** Within our taxonomy, we compared different architectures like Single-Agent, Multi-Agent, and Plan-Then-Act, discussing their intrinsic strengths, limitations, and potential challenges when handling tasks of varying complexity.
>
> **Building upon this analytical foundation, and to further enhance the manuscript's critical depth as you recommended, we have undertaken the following key revisions:**
>
> 1. **Added In-depth Comparative Analysis of Frameworks (Section 4.6):** We introduced a new subsection that deeply explores the design trade-offs across the three core components of any agent: **Perception**, **Brain**, and **Action**. For instance, we analyze the conflict between informational richness and speed in perception, and the "trilemma" of balancing speed, memory, and cost in the agent's cognitive core.
> 2. **Added In-depth Comparative Analysis of Modeling Approaches (Section 5.3):** To directly address your example, we added a dedicated subsection with a new table that provides a comprehensive head-to-head comparison of **Prompt Engineering versus Training-Based Methods**. This analysis covers multiple dimensions, including data dependency, performance ceilings, development costs, and the potential for on-device deployment.
> 3. **Added Quantitative Performance Analysis and Limitations (Section 6.3):** To shift from qualitative descriptions to data-driven insights, we introduced new content in Section 6.3, including a central **performance comparison table (Table 11)**. Based on this quantitative data, we systematically analyze the performance bottlenecks of current leading methods (e.g., models excelling at visual grounding but struggling with complex online tasks).
> 4. **Added a Critical Analysis of Datasets and Benchmarks (Section 6.4):** To address your point on "benchmarks' coverage gaps," we added Section 6.4, a new subsection dedicated to a critical examination of the common limitations of existing resources, such as imbalanced task coverage, insufficient annotation depth, and narrow evaluation dimensions.
> 5. **Restructured the Challenges and Future Directions Section (Section 7):** We have completely restructured Section 7 and introduced a **new summary table (Table 14)** that explicitly maps "Existing Techniques" against "Future Challenges" for each key technical dimension. The entire chapter's narrative is now organized around this table to provide readers with a clearer, more structured critical synthesis of the field's current state and future research gaps.
>
> We are confident that by integrating our original analytical framework with these new, multi-dimensional comparative and critical analyses, our manuscript has been elevated from a comprehensive survey to a more insightful and critical synthesis. We thank you again for your constructive feedback, which has significantly improved the quality of our work.

---

> ### Author Response · Authors · 2025-09-02
> **Response to Reviewer u3op Part 1**
>
> We sincerely appreciate your thoughtful review and valuable suggestions. We address your concerns below and summarize the corresponding improvements.
>
> *W1: The paper sometimes reads as a catalog of existing works rather than a critical synthesis. For example, the taxonomy could be accompanied by more comparative analysis (e.g., trade-offs between prompt vs. training-based methods, benchmarks’ coverage gaps).*
> **Response.** Thank you for this valuable feedback. We fully agree that an excellent survey must provide insightful critical synthesis, not just a catalog of existing works. We would first like to clarify that our original manuscript was already built upon an analytical framework, which we have now significantly deepened and expanded based on your insightful suggestions.
>
> **Our initial draft already provided multi-faceted analysis and synthesis beyond a simple catalog:**
>
> 1. **Establishing a Problem-Solution Narrative (Section 3):** We did not simply list techniques. Instead, in Section 3.2, we systematically diagnosed the four core challenges of traditional, pre-LLM automation methods (e.g., poor intent comprehension, high maintenance costs). Then, in Section 3.3, we demonstrated precisely how the key capabilities of LLMs (e.g., contextual semantic understanding, multi-modal perception) directly address these long-standing industry bottlenecks.
> 2. **Providing a Theoretical Framework (Section 4):** At the beginning of Section 4, we modeled the agent's decision-making process as a Partially Observable Markov Decision Process (POMDP). This provides a unified, theoretical lens for the entire field, allowing for an understanding and comparison of the underlying logic of different agent architectures that goes beyond a mere description of their implementations.
> 3. **Analyzing Architectural Paradigms (Section 5):** Within our taxonomy, we compared different architectures like Single-Agent, Multi-Agent, and Plan-Then-Act, discussing their intrinsic strengths, limitations, and potential challenges when handling tasks of varying complexity.
>
> **Building upon this analytical foundation, and to further enhance the manuscript's critical depth as you recommended, we have undertaken the following key revisions:**
>
> 1. **Added In-depth Comparative Analysis of Frameworks (Section 4.6):** We introduced a new subsection that deeply explores the design trade-offs across the three core components of any agent: **Perception**, **Brain**, and **Action**. For instance, we analyze the conflict between informational richness and speed in perception, and the "trilemma" of balancing speed, memory, and cost in the agent's cognitive core.
> 2. **Added In-depth Comparative Analysis of Modeling Approaches (Section 5.3):** To directly address your example, we added a dedicated subsection with a new table that provides a comprehensive head-to-head comparison of **Prompt Engineering versus Training-Based Methods**. This analysis covers multiple dimensions, including data dependency, performance ceilings, development costs, and the potential for on-device deployment.
> 3. **Added Quantitative Performance Analysis and Limitations (Section 6.3):** To shift from qualitative descriptions to data-driven insights, we introduced new content in Section 6.3, including a central **performance comparison table (Table 11)**. Based on this quantitative data, we systematically analyze the performance bottlenecks of current leading methods (e.g., models excelling at visual grounding but struggling with complex online tasks).
> 4. **Added a Critical Analysis of Datasets and Benchmarks (Section 6.4):** To address your point on "benchmarks' coverage gaps," we added Section 6.4, a new subsection dedicated to a critical examination of the common limitations of existing resources, such as imbalanced task coverage, insufficient annotation depth, and narrow evaluation dimensions.
> 5. **Restructured the Challenges and Future Directions Section (Section 7):** We have completely restructured Section 7 and introduced a **new summary table (Table 14)** that explicitly maps "Existing Techniques" against "Future Challenges" for each key technical dimension. The entire chapter's narrative is now organized around this table to provide readers with a clearer, more structured critical synthesis of the field's current state and future research gaps.
>
> We are confident that by integrating our original analytical framework with these new, multi-dimensional comparative and critical analyses, our manuscript has been elevated from a comprehensive survey to a more insightful and critical synthesis. We thank you again for your constructive feedback, which has significantly improved the quality of our work.

---

> ### Author Response · Authors · 2025-09-02
> **Response to Reviewer u3op Part 1**
>
> We sincerely appreciate your thoughtful review and valuable suggestions. We address your concerns below and summarize the corresponding improvements.
>
> *W1: The paper sometimes reads as a catalog of existing works rather than a critical synthesis. For example, the taxonomy could be accompanied by more comparative analysis (e.g., trade-offs between prompt vs. training-based methods, benchmarks’ coverage gaps).*
> **Response.** Thank you for this valuable feedback. We fully agree that an excellent survey must provide insightful critical synthesis, not just a catalog of existing works. We would first like to clarify that our original manuscript was already built upon an analytical framework, which we have now significantly deepened and expanded based on your insightful suggestions.
>
> **Our initial draft already provided multi-faceted analysis and synthesis beyond a simple catalog:**
>
> 1. **Establishing a Problem-Solution Narrative (Section 3):** We did not simply list techniques. Instead, in Section 3.2, we systematically diagnosed the four core challenges of traditional, pre-LLM automation methods (e.g., poor intent comprehension, high maintenance costs). Then, in Section 3.3, we demonstrated precisely how the key capabilities of LLMs (e.g., contextual semantic understanding, multi-modal perception) directly address these long-standing industry bottlenecks.
> 2. **Providing a Theoretical Framework (Section 4):** At the beginning of Section 4, we modeled the agent's decision-making process as a Partially Observable Markov Decision Process (POMDP). This provides a unified, theoretical lens for the entire field, allowing for an understanding and comparison of the underlying logic of different agent architectures that goes beyond a mere description of their implementations.
> 3. **Analyzing Architectural Paradigms (Section 5):** Within our taxonomy, we compared different architectures like Single-Agent, Multi-Agent, and Plan-Then-Act, discussing their intrinsic strengths, limitations, and potential challenges when handling tasks of varying complexity.
>
> **Building upon this analytical foundation, and to further enhance the manuscript's critical depth as you recommended, we have undertaken the following key revisions:**
>
> 1. **Added In-depth Comparative Analysis of Frameworks (Section 4.6):** We introduced a new subsection that deeply explores the design trade-offs across the three core components of any agent: **Perception**, **Brain**, and **Action**. For instance, we analyze the conflict between informational richness and speed in perception, and the "trilemma" of balancing speed, memory, and cost in the agent's cognitive core.
> 2. **Added In-depth Comparative Analysis of Modeling Approaches (Section 5.3):** To directly address your example, we added a dedicated subsection with a new table that provides a comprehensive head-to-head comparison of **Prompt Engineering versus Training-Based Methods**. This analysis covers multiple dimensions, including data dependency, performance ceilings, development costs, and the potential for on-device deployment.
> 3. **Added Quantitative Performance Analysis and Limitations (Section 6.3):** To shift from qualitative descriptions to data-driven insights, we introduced new content in Section 6.3, including a central **performance comparison table (Table 11)**. Based on this quantitative data, we systematically analyze the performance bottlenecks of current leading methods (e.g., models excelling at visual grounding but struggling with complex online tasks).
> 4. **Added a Critical Analysis of Datasets and Benchmarks (Section 6.4):** To address your point on "benchmarks' coverage gaps," we added Section 6.4, a new subsection dedicated to a critical examination of the common limitations of existing resources, such as imbalanced task coverage, insufficient annotation depth, and narrow evaluation dimensions.
> 5. **Restructured the Challenges and Future Directions Section (Section 7):** We have completely restructured Section 7 and introduced a **new summary table (Table 14)** that explicitly maps "Existing Techniques" against "Future Challenges" for each key technical dimension. The entire chapter's narrative is now organized around this table to provide readers with a clearer, more structured critical synthesis of the field's current state and future research gaps.
>
> We are confident that by integrating our original analytical framework with these new, multi-dimensional comparative and critical analyses, our manuscript has been elevated from a comprehensive survey to a more insightful and critical synthesis. We thank you again for your constructive feedback, which has significantly improved the quality of our work.

---

> ### Author Response · Authors · 2025-09-02
> **Response to Reviewer u3op Part 2**
>
> *W2: The title and abstract emphasize phone automation, but parts of the discussion (e.g., Anthropic’s Computer Use, PC/web agents) drift toward general GUI agents. The scope could be tightened or reframed more explicitly as cross-device GUI automation with emphasis on phones.*
>
> **Response.** We thank the reviewer again for this insightful point, which has prompted us to more clearly articulate the logic and scope of our survey. We agree that the initial draft did not adequately explain why we included some agents that are not strictly mobile-specific. Following your suggestion to tighten the paper's scope, we haveremoved the discussion of Anthropic's Computer Use from the former Section 3.4.We would like to add that **phone automation does not exist in isolation, but is part of the broader evolution of the GUI agent ecosystem**. For the other cross-platform works that we have retained, our core motivation is that they provide an indispensable theoretical foundation and practical reference for mobile agent research, as detailed below:
>
> * **To provide a reference for framework design (Section 4.4):** Frameworks such as `Cradle` for computer control and the web-based `SteP` showcase advanced multi-agent collaboration patterns—namely, "role-coordination" and "scenario-based task execution," respectively. These architectures offer valuable blueprints and inspiration for designing multi-agent systems on mobile platforms capable of handling complex cross-app tasks.
> * **To showcase cutting-edge reinforcement learning techniques (Section 5.2.3):** Mobile RL agents face challenges like dynamic environments and sparse rewards. Web agents, such as `ETO` and `Agent Q`, are more advanced in exploring cutting-edge RL techniques like "learning from failures" and "exploration-based trajectory optimization." Discussing them provides crucial technical insights and a frame of reference for solving similar problems on mobile.
>
> To reflect the logic for retaining these examples more clearly in the paper, we have adopted the reviewer's suggestion. We have revised the **introduction (Section 1)** to clarify our scope and added **transitional sentences** in the relevant sections of the main body (e.g., Sections 4.4 and 5.2.3) to articulate the direct relevance of these cross-platform examples to our core topic of phone automation. We believe that with these changes, the paper's argumentation is more rigorous and its content more comprehensive.
>
> **Reference**
>
> [1] Cradle: Empowering Foundation Agents towards General Computer Control.
>
> [2] SteP: Stacked llm policies for web actions.
>
> [3] Trial and error: Exploration-based trajectory optimization for llm agents.
>
> [4] Agent Q: Advanced reasoning and learning for autonomous ai agents.

---

> ### Author Response · Authors · 2025-09-02
> **Response to Reviewer u3op Part 3**
>
> *W3: Although datasets and benchmarks are listed, the survey could provide more in-depth analysis of their strengths/weaknesses.*
>
> **Response.** Thank you for this suggestion. To provide a more in-depth analysis of datasets and benchmarks, we have made revisions from both quantitative and qualitative perspectives.
>
> For the **quantitative analysis**, we added **Section 6.3 ("Performance Analysis and Current Limitations")**, which includes a new **performance comparison table (Table 11)**. This table uses data to offer a critique of current methods, highlighting limitations such as the significant performance gap between offline benchmarks and online, real-world tasks.
>
> For the **qualitative analysis**, we introduced **Section 6.4 ("Critical Analysis of Datasets and Benchmarks")**. This subsection systematically examines the common weaknesses of existing **datasets** (e.g., imbalanced task coverage, insufficient annotation depth, and a lack of real-world dynamism) and the limitations of **benchmarks** (e.g., narrow evaluation dimensions and challenges in automated evaluation).
>
> We believe these additions provide the in-depth analysis of strengths and weaknesses that you suggested.
>
> *RC1: Can the authors make a table that summarize “existing techniques vs. challenges”?*
>
> **Response.** Thank you for this excellent suggestion. We have fully adopted your proposal and have taken it as an opportunity to systematically restructure the latter part of our manuscript to present a clearer and more insightful "existing techniques vs. challenges" landscape.
>
> Our core revisions are as follows:
>
> 1. **Added Core Trade-off Analysis Tables:** As you suggested, we have added several summary tables. In particular, **the table introduced in Section 4.6 (Table 14)** systematically summarizes the trade-offs between the "current state & strengths" (i.e., existing techniques) and the "limitations & challenges"s.
> 2. **Restructured the Challenges and Future Directions Section (Section 7):** To align with the new analytical tables and a deeper critical perspective, we have restructured the entire narrative and content of **Section 7 (Challenges and Future Directions)**. The discussion is now more focused and systematically explores key challenges such as data, deployment, and security, providing readers with a more organized overview of the field's landscape.
>
> Furthermore, it is worth noting that this new synthesis of "techniques vs. challenges" is built upon the foundation of several in-depth analyses we conducted in response to other reviewer comments. As detailed in our response to W1, we have added deep comparisons of **agent frameworks (Sec 4.6)** and **modeling approaches (Sec 5.3)**, as well as a **quantitative performance analysis (Sec 6.3)** and a **critique of existing benchmarks (Sec 6.4)**. This new critical content provides a solid foundation for the restructured Section 7, collectively forming a more complete and insightful analytical framework.
>
> We believe these systematic revisions not only directly and comprehensively address your suggestion but also significantly enhance the structural clarity and analytical depth of the latter half of our paper.
>
> *Broader Impact Concerns:*
>
> **Response.** Thank you for raising these important ethical considerations. Our original manuscript already included a subsection in **Section 7** dedicated to "Ensuring Reliability and Security." To strengthen this discussion as you suggested, we have expanded this section significantly:
>
> * We renamed the subsection to **"Ensuring Reliability, Security, and Ethical Alignment"** to better reflect its expanded scope.
> * Within this section, we have added new discussions that explicitly address the risks of **malicious use** (e.g., automated fraud via sophisticated attacks like Active Environmental Injection Attacks) and other unintended harmful behaviors. We also introduce security benchmarks like `MobileSafetyBench` to highlight systematic mitigation efforts.
>
> **Reference**
>
> [1] MobileSafetyBench: Evaluating Safety of Autonomous Agents in Mobile Device Control.

---

> > ### Comment · Reviewer_u3op · 2025-10-18
> > **Acknowledgement of Authors' Response**
> >
> > I thank the authors for their detailed and highly responsive revisions. They have thoroughly addressed all the concerns I raised in my initial review. The paper has been significantly improved.  I have no further major concerns.

---

### Review · Reviewer_sUUv · 2025-09-29

**Summary Of Contributions:**

The paper conducts a thorough survey on the LLM-based GUI agent development of phone automation. It covers a broad range of topics, from the traditional phone automation system to the current LLM-based GUI agent, and it also points out the current system's limitations and several future research directions.

**Audience:**

Yes

**Broader Impact Concerns:**

None.

**Claims And Evidence:**

Yes

**Requested Changes:**

Please refer to the weaknesses. I would like to shorten the paper to make the focus narrower to the LLM-based agent method. Although it is good to cover all the related works, making the survey too long will make the paper too broad and dull.

**Strengths And Weaknesses:**

Pros:
1. The paper covers a lot of topics and areas in phone automation. Although I am not an expert on phone automation, the survey looks comprehensive and can be beneficial for the community.
2. The paper includes some analysis on the current system limitations and points out several potential directions, which is helpful and insightful.

Cons:
1. The paper covers too many areas, which, in my opinion, are not very close to the topic of LLM-based GUI agents for phone automation. The development of phone automation can be mentioned, but it shouldn't be discussed too much, as well as several traditional phone automations, such as model-based, learning-based, etc. I suggest incorporating them into a short paragraph or section.
2. The paper needs to address more on the specificity of the phone automation agent rather than the general agent.

---

### Author Response · Authors · 2025-09-02
**Authors' Response to Reviewers**

We sincerely thank all the reviewers for their detailed and constructive feedback. We are encouraged that all reviewers found our survey to be timely, comprehensive, and relevant. We have carefully considered all comments and have revised the manuscript accordingly. Below, we address each reviewer's comments point by point. **All changes in the revised manuscript are** **highlighted in purple**. Furthermore, to ensure our survey remains at the cutting edge, we have actively tracked and incorporated **the latest relevant work published during the review period**; these additions are **highlighted in orange**, demonstrating our commitment to providing the community with a definitive and timely reference for this rapidly evolving field.

---

### Decision · Action_Editor_57s3 · 2025-10-15

**Recommendation:** Accept with minor revision

**Additional Comments:**

Please conduct a final proofreading pass to ensure overall consistency in terminology, clarity of language, and flow throughout the manuscript.

**Audience:**

Yes

**Audience Explanation:**

The paper addresses a highly relevant and rapidly evolving field at the intersection of large language models and mobile automation. Given the pervasive use of smartphones and the increasing capabilities of LLMs, the findings of this survey are undoubtedly of interest to a significant portion of TMLR's audience, including researchers and practitioners working on AI agents, mobile computing, human-computer interaction, and privacy-preserving machine learning. All reviewers (sUUv, JVjB, and DAiC) explicitly stated "Yes" for this criterion.

**Claims And Evidence:**

Yes

**Claims Explanation:**

The paper presents a comprehensive survey of LLM-powered GUI agents for phone automation. The authors have diligently addressed the concerns raised by the reviewers regarding the depth of analysis, distinctiveness from broader surveys, and the inclusion of quantitative comparisons. The detailed responses by the authors, including the addition of new sections, comparative tables (e.g., Table 1, Table 11, Table 14), and expanded discussions on LLM-centric insights, distributed systems, privacy, and ethical alignment, demonstrate that the claims are well-supported. Reviewers sUUv, JVjB, and DAiC all indicated "Yes" for this criterion, even prior to the authors' revisions.